# H3K36 Methylation as a Guardian of Epigenome Integrity

Reinnier Padilla [1,2,3], Gerry A. Shipman [1,2,3], Cynthia Horth [1,2], Michel Gravel[1,2], Eric Bareke [1,2] & Jacek Majewski [1,2] ✉

H3K36 methylation is a key epigenetic mark with critical roles in development and disease. Here, we systematically dissect its functions using CRISPR-engineered mouse mesenchymal stem cells lacking combinations of the five H3K36 methyltransferases, culminating in quintuple knockout cells devoid of H3K36me2/3. We show that H3K36me2 influences enhancer activity, supports the expression of their target genes, and safeguards active genes from encroachment of the repressive marks, H3K27me2/3. In addition, we find that the loss of H3K36me triggers redistribution of large heterochromatic H3K9me3 domains into euchromatin, in part mediated by SUV39H1, leading to global epigenomic remodelling, constitutive heterochromatin erosion, and a collapse of 3D genome organization. Parallel analyses in human HNSCC cells overexpressing the H3K36M oncohistone reveal conserved disruptions to the epigenome and chromatin architecture. Together, these results establish H3K36 methylation as a pivotal regulator of chromatin state and genomic structure.

DNA is wrapped around histones to form nucleosomes, the structural and functional unit of chromatin. Chromatin can be densely packed as heterochromatin or loosely packed as euchromatin. The extent of compaction, and thereby access to DNA, is in part regulated by chemical modifications to the histone tails, known as histone post-translational modifications (PTMs). These histone PTMs form a highly interdependent network, where euchromatic modifications facilitate access to DNA and heterochromatic modifications restrict access to DNA[1–3]. The different types of modifications at histone residues induce conformational changes that influence adjacent or interacting residues.

Methylation of histones plays pivotal roles in modulating transcriptional processes, primarily by influencing the accessibility of chromatin and serving as a substrate for a multitude of effector proteins to exert their functions. The lysine residues on the N-terminal tails of histones can receive up to three methylation marks, and the degree of methylation at specific residues is often associated with distinct transcriptional states and genomic features[4,5]. While the methylation states of certain residues coincide on the same histone tail

and support similar processes, the methylation states of other residues are antagonistic, often due to steric bulk, and support opposing processes. In addition, the collective effects of histone methylation significantly influence the organization and 3D architecture of interphase chromatin[6–11]. Over the past decade, methylation of histone 3 lysine 36 (H3K36me) has increasingly emerged as a critical component of the epigenome. When aberrantly regulated, either through mutations affecting the genes of its writers or of histone 3 (H3), developmental disorders or cancer ensues, underscoring its functional significance[12–21]. H3K36me exists in mono- (H3K36me1), di-(H3K36me2), or tri-methylated (H3K36me3) states, and there are five well-established enzymes that catalyze H3K36 methylation: SETD2, NSD1, NSD2, NSD3, and ASH1L. Briefly, SETD2 tethers to the elongating RNAPII complex within actively transcribed genes to deposit nearly all H3K36me3[22,23]. In contrast, the NSD family of lysine 36 methyltransferases (K36MTs) have established roles in the deposition of the lower methylation states, H3K36me1/2. NSD1 and NSD2 have been found to be responsible for the majority of global H3K36me2, in both IGRs and genes, and their individual contributions appear to vary by

[1]Department of Human Genetics, McGill University, Montreal, Quebec, Canada. [2]Victor Phillip Dahdaleh Institute of Genomic Medicine, Montreal, Quebec, Canada. [3]These authors contributed equally: Reinnier Padilla, Gerry A. Shipman. ✉e-mail: jacek.majewski@mcgill.ca

cell type and developmental stage[3,14,18,24–26]. In comparison, we have previously found that NSD3 has a more distinct role in the deposition of H3K36me2, primarily exerting its catalytic activity at active cis-regulatory elements (CREs)−most notably at enhancers and promoter flanking regions[26]. Finally, ASH1L is the most specific and least prolific of the K36MTs, primarily depositing H3K36me2 at specific developmentally important genes[26].

Since H3K36me is generally associated with transcriptionally active, euchromatic regions of the genome, it is presumed to play a role in maintaining and fine-tuning transcription. It is one of the modifications broadly flanking CREs, whose regulatory states are frequently inferred by the enrichment profiles and co-localization of several histone modifications[27]. Notably, H3K4 methylation (H3K4me) and H3K27 acetylation (H3K27ac) are amongst the most well characterized modifications that distinguish active CREs. Functionally, H3K4me1 serves to fine tune enhancer activity by acting as a substrate for the binding of chromatin remodelers and other protein effectors[28], while H3K27ac has a more direct effect on chromatin by neutralizing the positively charged histone tails, thereby weakening their interaction with DNA and establishing an environment permissive for transcription[29]. The co-presence of H3K4me1 and H3K27ac is frequently used to identify active enhancers[27,30]. In contrast, focal peaks of H3K4me3 and H3K27ac are hallmark chromatin signatures of active promoters[27,31]. H3K36me2 has been found to exist in broad domains at both active enhancers and promoters in most assayed cell types[3,18,24–26]. Loss of broad H3K36me2 domains, resulting from either pathological mutations or genetic ablation of its writer enzymes, has been shown to impact the distribution of H3K4me1/3 and H3K27ac, and is associated with altered gene expression[18,25,32]. While the general relationship between H3K36me2, CREs and gene expression has been documented, it has always been studied in the context of very broad, megabase scale domains, and the precise role of H3K36me2 in maintaining the activity of CREs and its influence on the deposition of other active histone marks in a localized setting has yet to be carefully examined.

Our understanding of the relationship between H3K36me and heterochromatin is also far from complete. On one hand, there exists a well-documented antagonism between H3K36me2/3 and the silencing marks H3K27me2/3. The PRC2 complex, which deposits all H3K27me, is allosterically inhibited from nucleosomes marked with H3K36me2/3[33,34]. As a result, particularly H3K27me3 is excluded from H3K36me2/3 domains. While H3K27me2/3 are considered to be repressive marks, they are generally found in gene-rich regions and their presence is associated with facultative heterochromatin[35]. In comparison, H3K9me3 is distinctive of constitutive heterochromatin, and broad H3K9me3 domains generally span gene-poor regions containing tandem repeat elements, such as those found at telomeres and pericentromeric regions[6,36,37]. While broad genomic domains of H3K9me3 and H3K36me2 tend to be mutually exclusive, the relationship between H3K36me and H3K9me is currently unclear, largely because evidence for mechanistic links between the two marks is lacking.

One of the challenges associated with investigating the functional role of H3K36me is the broad nature of its distribution, which is exacerbated by the number of enzymes capable of depositing these modifications. We have previously used CRISPR-Cas9 gene editing to establish clonal cell lines harboring individual or multiple knockouts (KOs) of the K36MTs, for the purpose of deconstructing their individual and combined contributions to the H3K36me landscape in C3H10T1/2 mouse mesenchymal stem cells (mMSCs)[26]. Here, we employ these cell lines to investigate the effects of H3K36me in more specific contexts: we examine focal peaks of NSD3- and ASH1L-mediated H3K36me2 present at active CREs, and their subsequent erasure, to assess their functional influence on the local epigenome and transcriptome. We provide further mechanistic insights into the ability of H3K36me2/3 to exclude H3K27me2/3 from actively transcribed genes, and the downstream impact on other relevant epigenetic modifications in these regions. Unexpectedly, we uncover a relationship between H3K36me, H3K9me3 and the 3D architecture of the genome that appears to be conserved in both mouse and human cells. Overall, the progressive loss of H3K36me resulting from the sequential deletion of the K36MTs provides a unique opportunity to assess the role of H3K36me in maintaining the active state of CREs, transcription and genomic compartmentalization.

## Results

### NSD3-mediated H3K36me2 maintains the active state of enhancers and their target genes

We have previously found that knocking out NSD1/2 (DKO) in mMSCs results in a near total depletion of broad intergenic H3K36me2[38] (Fig. 1a). Further KO of SETD2 in the triple KO cells (TKO) results in the loss of nearly all genic H3K36me2 and H3K36me3[26] (Fig. 1a). The remaining focal H3K36me2 regions, deposited by NSD3, are primarily targeted to active CREs[26]. Upon subsequent KO of NSD3 in the NSD1/2/3-SETD2 quadruple KO cells (QKO), these broad peaks of H3K36me2 disappear (Fig. 1a). To further explore the role of H3K36me2 in these confined regions, we examined promoters and enhancers that remain accessible, as defined by the presence of an ATAC-seq peak in both the TKO and QKO conditions. Having previously established biological replicates for each cell line, here we used ChIP-seq (H3K36me2, H3K27ac, H3K4me3) and CUT&RUN (H3K4me1, H3K27me3) to profile the relevant active and repressive histone modifications (Supplementary Fig. 1a), ATAC-seq for chromatin accessibility, whole-genome bisulfite sequencing (WGBS) for DNA methylation (DNAme), and RNA-seq for transcriptional state to document the downstream changes following the depletion of H3K36me2 at CREs and their associated genes.

At active enhancers, the depletion of H3K36me2 is accompanied by slight decreases in chromatin accessibility (Fig. 1b). Concurrently, we also observe a decrease in the levels of other known active enhancer modifications, specifically H3K4me1/3 and H3K27ac (Fig. 1b). In contrast, the levels of H3K27me3 increase at enhancers that are depleted of H3K36me2 (Fig. 1b), which was expected given the known antagonistic relationship between these marks[33,34]. Previous studies have shown that H3K36me2/3 serve as templates for the deposition of de novo DNAme by DNMT3A/B, respectively[38,39]. Therefore, we expected that in the absence of H3K36me2/3, enhancers may experience substantial reductions in CpG methylation. Using WGBS, however, we find that DNAme is largely unchanged (Fig. 1b). We also interrogated strong enhancers, as defined by dense clusters of individual enhancers (Fig. 1a; Supplementary Fig. 1b), which are known to be essential for the maintenance and regulation of genes crucial to both normal development and cancer[40]. In TKO cells, most strong enhancers retain H3K36me2, which is subsequently depleted in the QKO condition, demonstrating that NSD3 is recruited to these regions (Fig. 1a, c). Here, we found similar yet more pronounced trends in the decrease of activating and increase of silencing modifications, further demonstrating that H3K36me2 is essential to the maintenance of enhancer activity (Fig. 1c).

Interestingly, the changes occurring at the promoters of NSD3 target genes appear to be considerably more attenuated, as compared to those at enhancers. The overall levels of NSD3-deposited H3K36me2 in TKO cells are lower at accessible promoters than at accessible enhancers (Fig. 1b–d). Thus, while the loss of H3K36me2 is nearly complete at all CREs, the relative decrease is smaller at promoters. Consequently, at promoters we observe no change to chromatin accessibility and H3K27ac, while the decrease of H3K4me1 and increase of H3K27me3 is lower than at enhancers (Fig. 1d). We do, however, observe a comparable twofold decrease in both promoter- and enhancer-marked H3K4me3 (Fig. 1d). Overall, these results provide further evidence that the presence of NSD3-associated H3K36me2 is particularly influential at enhancers.

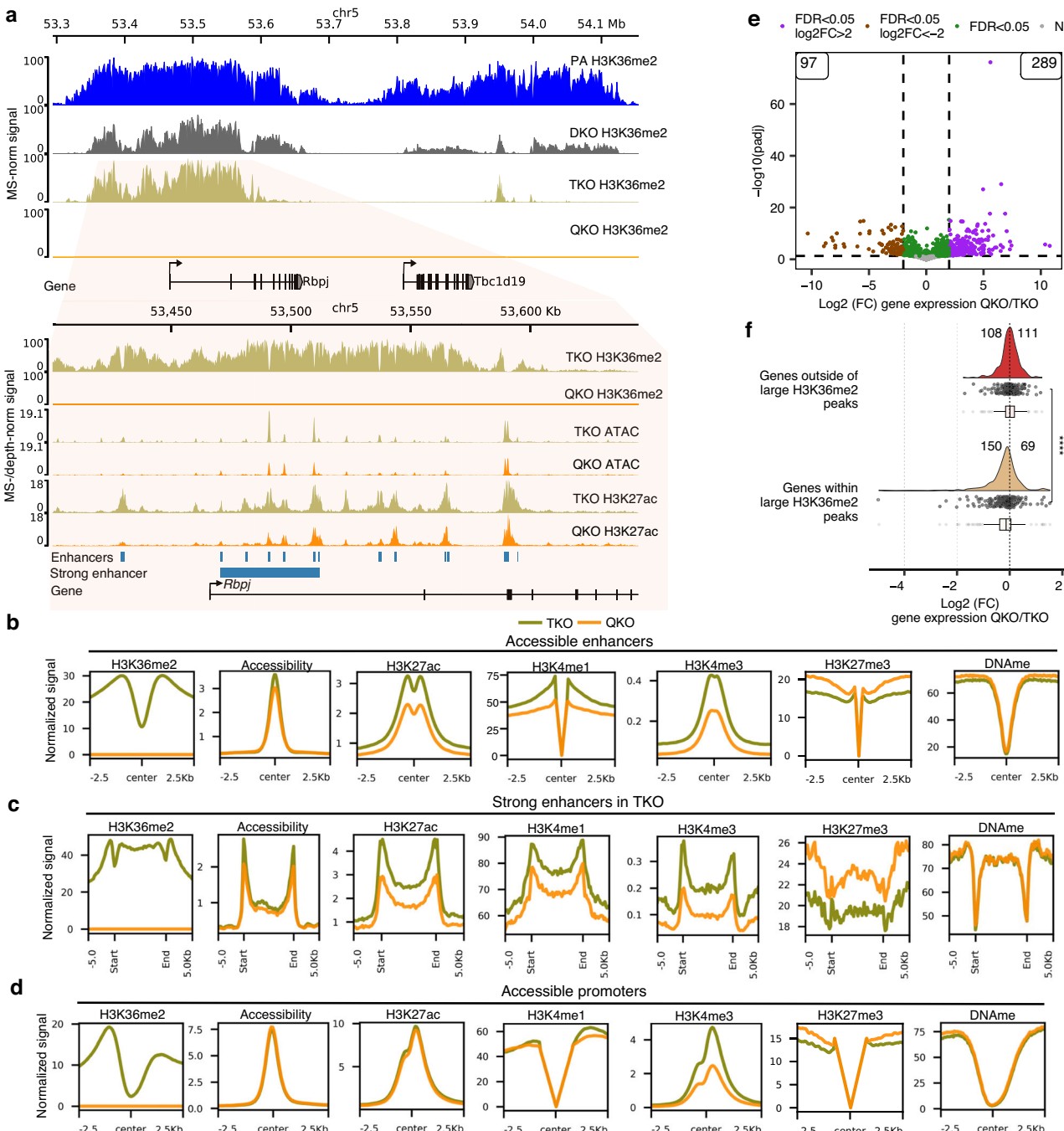

**Fig. 1 | Loss of NSD3-dependent H3K36me2 leads to reduced enhancer activity and downregulation of targeted genes. a** Representative genome browser tracks showing progressive loss of H3K36me2 in multiple-KO conditions: NSD1/2 double-KO (DKO); NSD1/2-SETD2 triple-KO (TKO); NSD1/2/3-SETD2 quadruple-KO (QKO). In TKO, the remaining H3K36me2 peaks mark several active and strong enhancers, which are ablated in QKO and lead to reduced activity of enhancers, indicated by reduced ATAC-Seq and H3K27ac. **b** Aggregate plots comparing TKO to QKO, indicating reduced chromatin accessibility, decreased signals of active marks (H3K27ac, H3K4me1/3) and invasion of H3K27me3 at accessible enhancers ($n = 22,706$). **c** Aggregate plots centered on strong enhancers ($n = 697$), showing further reduced activity for active marks and more pronounced invasion of H3K27me3. **d** Aggregate plots centered on accessible promoters showing no changes in accessibility, DNAme or H3K27ac, whereas H3K4me1/3 decreases and H3K27me3 increases ($n = 9599$). **e** Volcano plot of differentially expressed genes, showing more genes become upregulated (289) than downregulated (97) comparing QKO to TKO. **f** Violin plots of expression-matched genes within and outside

of large H3K36me2 peaks (see **a** for example peak), demonstrating more genes becoming downregulated (150) than upregulated (69). An equal number of genes outside of these large H3K36me2 peaks were randomized and selected as control. Statistical significance was tested using a two-sided Wilcoxon rank-sum test. **** represents $p$-value = 5.8e-07. In the box plots, boxes span the first quartile and third quartile, median is indicated with a center line and whiskers extend to 1.5 times the interquartile range. For **a**–**d**, biological replicates ($n = 3$) were merged. ATAC-seq tracks were depth-normalized. For all other tracks, except for DNAme (which indicates the percent methylation at CpG sites), normalization factors were computed by multiplying genome-wide modification percentages (averaged per condition) from mass-spectrometry (MS) by the total number of bins and dividing by the total signal for a given coverage track. This normalization factor was multiplied to the depth-normalized signal to generate MS-normalized signals. MS-normalized signals represent the mean local frequency of the relevant modification. Source data are provided as a Source Data file.

We next examined the effects on gene expression following the depletion of CRE-associated H3K36me2. Genome-wide, we found no general trend towards downregulation, with more than twice as many genes going up in expression (289) than down (97) (Fig. 1e). However, many of these changes may be secondary, downstream effects from global H3K36me2 depletion and/or ablation of NSD3. In view of the more pronounced epigenetic changes occurring at enhancers, we hypothesized that the most direct effect on gene expression should be observed at genes whose expression is dependent on H3K36me2-marked enhancers. We identified a set of predicted H3K36me2-dependent genes based on their overlap with large H3K36me2 peaks that contained at least one annotated enhancer (from Ensembl and FANTOM databases). We also created a complimentary, randomized control set of H3K36me2-independent genes. The two sets were further matched for baseline gene expression levels by selecting only transcripts within the 9-12 FPKM range (Supplementary Fig. 1c). We found that following the loss of NSD3-dependent H3K36me2, genes associated with large H3K36me2 peaks become significantly more downregulated (150 down versus 69 up) compared to the control set of genes (108 down versus 111 up) (Fig. 1f). Our findings indicate that the depletion of NSD3-dependent H3K36me2 impacts gene expression primarily by reducing enhancer activation of target genes rather than directly influencing promoter activity.

We have previously examined the effects of the individual KO of NSD3 on the global H3K36me2 landscape, and we did not observe a significant depletion in the global abundance of H3K36me2, possibly because of compensation by the other K36MTs[26]. To investigate whether the effects on CREs are a result of the H3K36me2 catalytic product or driven by NSD3 itself, we analyzed the same set of enhancer-dependent NSD3 target genes identified in the TKO cells in the context of comparing parental to NSD3-KO cells. Although globally, we find that many more genes significantly increase in expression (172) than decrease (8) (Supplementary Fig. 1d), within the set of predicted H3K36me2-dependent genes derived from the TKO cells, we found a tendency towards downregulation (122) versus upregulation (76) (Supplementary Fig. 1e). This suggests that the compensation of H3K36me2 levels by the other K36MTs is not complete. Upon closer inspection, we observe a decrease of H3K36me2 at the promoters of this set of genes in NSD3-KO cells (Supplementary Fig. 1f) and an even greater decrease at the nearest enhancers of these promoters (Supplementary Fig. 1g). Hence, even in parental cells, where NSD3 is not the dominant K36MT, its absence is reflected in a slight, localized reduction of H3K36me2 and downregulation of enhancer-dependent genes.

Overall, our results support that while NSD3-mediated H3K36me2 is targeted to both enhancers and promoters, it is most influential at enhancers with regards to maintaining their active chromatin states, restricting the invasion of H3K27me3, and has discernible effects on maintaining the expression of their target genes.

### Functional characterization of focal H3K36me2 at ASH1L target genes

In the QKO cells, we had previously identified 119 remaining H3K36me2 peaks, with the majority of these peaks straddling the transcription start sites (TSS) ($n = 60$) or located within the gene bodies ($n = 24$) of a subset of developmentally important genes[26]. Those peaks disappeared in the NSD1/2/3-SETD2-ASH1L-QuiKO (QuiKO) condition, following the ablation of ASH1L (Fig. 2a). Similar to the comparison between TKO and QKO cells, this provides a unique opportunity to study the functional consequences of H3K36me2 depletion in a very specific, as opposed to a global setting. We find that the majority of ASH1L target genes (62/84, $p = 0.0015$, Wilcoxon test) decrease in expression following the loss of ASH1L (Fig. 2b, c). Furthermore, the change in expression is proportional to the size of the original H3K36me2 peaks: genes with the largest H3K36me2 peaks in

QKO cells have the largest fold changes in gene expression (Fig. 2d). In contrast, the same set of genes did not decrease in expression when comparing TKO to QKO cells, where H3K36me2 is unaffected at these promoters (Fig. 2b, c). Finally, it is possible that ASH1L itself may act as a transcriptional modulator, and may have an effect independent of its catalytic activity - as has been recently suggested in the case of NSD1[25]. To distinguish whether the reduced expression of the 84 ASH1L-target genes may be due to the loss of ASH1L itself, or its catalytic product, H3K36me2, we looked at differences in expression between parental mMSCs and the individual ASH1L-KO cells. In the ASH1L-KO condition, the levels of H3K36me2 remain largely unchanged at the promoters of these genes, most likely as a result of compensation by the other K36MTs, which exhibit less specificity in their deposition of H3K36me2 (Supplementary Fig. 2a). Interestingly, we found no significant changes to the expression of ASH1L target genes, suggesting that the presence of H3K36me2, rather than the presence of ASH1L has a role in maintaining their expression (Fig. 2c).

Concurrently, we investigated changes to other epigenetic modifications at the remaining CREs marked with residual H3K36me2. Genome-wide, we found that chromatin accessibility increases in QuiKO cells compared to QKO, however, the CREs marked with residual H3K36me2 gain considerably less (Fig. 2e, f). This likely reflects the net negative effect of losing H3K36me2 at these specific loci in comparison to other genomic regions. Nevertheless, the depletion of ASH1L-mediated H3K36me2 is accompanied by a further decrease of H3K27ac and a corresponding increase in H3K27me3 at these CREs (Fig. 2e, f). Similar to the loss of NSD3 in QKO cells, we observe that the loss of H3K27ac is greater at enhancers than promoters (Fig. 2e, f), further supporting that the presence of H3K36me2 at CREs may be more influential at enhancers than promoters. Altogether, these analyses demonstrate that H3K36me2 contributes to maintaining the balance between activating and silencing epigenetic modifications at CREs, and this has consequences for the expression of target genes.

### The effect of H3K36 methylation on antagonizing H3K27me within genes

One of the presumed functions of H3K36me3 that is deposited within actively transcribed genes is to prevent the deposition of the silencing modifications at H3K27[33,34]. This effect may further be strengthened by the presence of H3K36me1/2. It has been shown that methylation at H3K36 antagonizes PRC2 and hinders the deposition of methylation at H3K27[33,34,41]. This is particularly true for the highest methylation levels, resulting in mutual exclusivity of H3K36me3 and H3K27me3 on the same histone tail[34,41]. Accordingly, H3K27me3 is excluded from actively transcribed genes, H3K27me2 is generally depleted, and only H3K27me1 invades gene bodies, where its distribution is similar to that of H3K36me3 (Fig. 3a). Using our panel of CRISPR-engineered mMSCs, we investigated the effects of H3K36me depletion on the presence of H3K27me in gene bodies. We used quantitative mass spectrometry (MS) to quantify the changes to the absolute abundance of each of the respective methylation marks, and ChIP-seq/CUT&RUN to profile changes to the global distributions of H3K27me states. Unexpectedly, in the SETD2 mutant which lacks nearly all H3K36me3 (Supplementary Fig. 3a), the increase of H3K27me3 in gene bodies is minimal, and primarily discernible within lowly transcribed genes (Fig. 3a; bottom row, middle box). However, the absence of H3K36me3 has a more pronounced effect on H3K27me2, allowing its deposition within gene bodies largely independent of transcriptional activity, and is accompanied by a corresponding reduction in H3K27me1 (Fig. 3a; middle row). Accordingly, these genic changes are reflected in the overall genome-wide abundance: H3K27me2 slightly increases, and this is accompanied by a proportional decrease of H3K27me1, while changes to H3K27me3 are minimal (Fig. 3b). These observations may be explained by the fact that the sole loss of SETD2 in the presence of the other intact K36MTs primarily depletes H3K36me3, and results in an

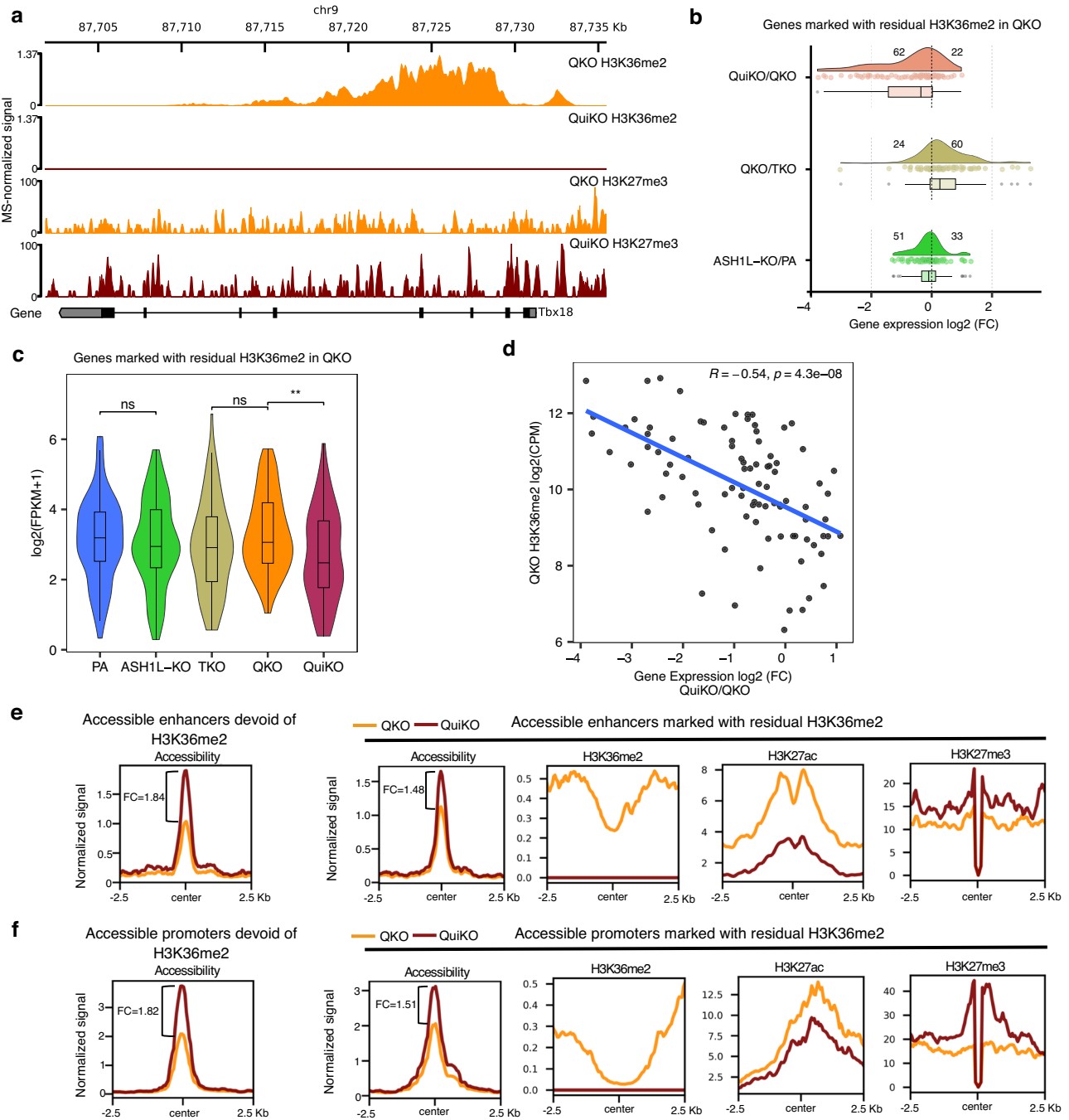

**Fig. 2 | Loss of ASH1L-mediated H3K36me2 results in the downregulation of genes marked by H3K36me2, decreased H3K27ac and increased H3K27me3 at cis-regulatory elements. a** Genome-browser tracks showing one of 84 genes that is marked by residual H3K36me2 in QKO cells, which is lost following ASH1L-KO in QuiKO cells. **b** Log2 fold changes of gene expression for the 84 genes marked by residual H3K36me2 in QKO, showing that more genes become downregulated (62) than upregulated (22) following loss of ASH1L-mediated H3K36me2. Within this gene set, 51 genes are downregulated while 33 are upregulated when comparing ASH1L-KO to the parental cells. This trend of downregulation is not observed when comparing QKO to TKO for this gene set, where 60 genes are upregulated and only 24 are downregulated. **c** Violin plots showing that the 84 genes in **b** only significantly decrease in expression when comparing QKO to QuiKO. ** p-value = 0.0017 from a two-sided Wilcoxon rank-sum test. **d** Scatterplot illustrating a significant negative correlation between the size of the H3K36me2 peak and gene expression log2 fold changes following ASH1L-dependent H3K36me2 loss in the

QKO cell line. Reported values are Pearson's correlation coefficient (R) and the associated p-value. Each point represents a gene. **e, f** Aggregate plots showing H3K36me2 and H3K27ac decreases whereas H3K27me3 increases at accessible enhancers (n = 72) and promoters (n = 55) marked with residual H3K36me2. Genome-wide, chromatin accessibility increases in QuiKO cells compared to QKO, although the CREs marked with residual H3K36me2 gain considerably less. Greater decreases are found at enhancers than promoters marked with residual H3K36me2. For **f**, since an intersection of ATAC-seq peaks between QKO and QuiKO was used to identify accessible promoters, only 55 out of the 60 H3K36me2-flanked promoters had an ATAC-seq peak detected at their promoter. Normalized signals were either depth-normalized (ATAC-seq) or MS-normalized (all other tracks), as previously described. In the box plots for **b** and **c**, boxes span the first quartile and third quartile, the median is indicated with a center line, and whiskers extend to a maximum of 1.5 times the interquartile range. Source data are provided as a Source Data file.

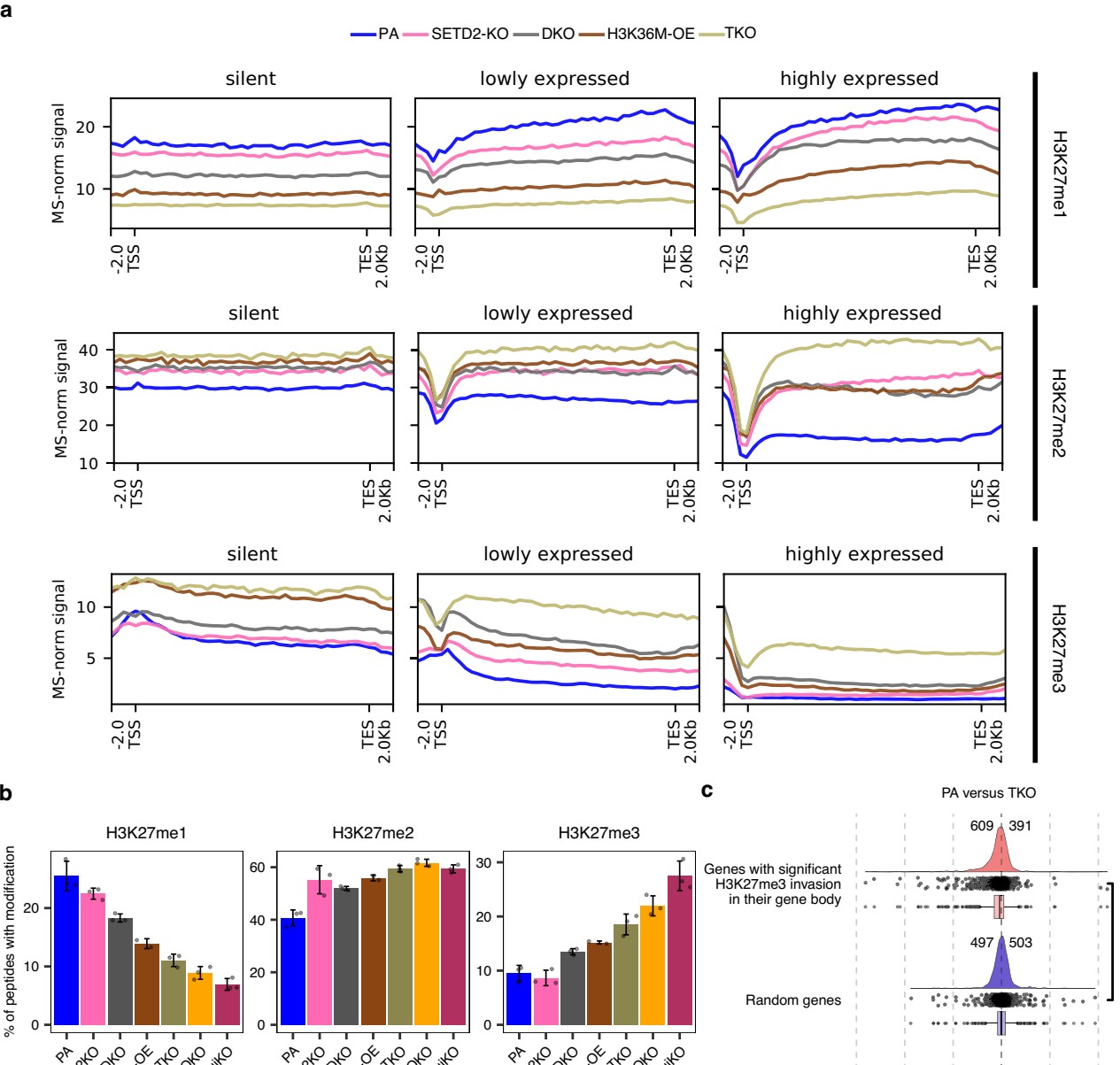

**Fig. 3 | Invasion of H3K27me3 into gene bodies following depletion of H3K36me. a** Metagene plots depicting the invasion of H3K27me within gene bodies. Protein-coding genes with gene lengths falling between the 25th and 75th percentiles and with FPKM values exceeding 0.1 were included. After filtering and stratifying into three quantiles, the bottom 2000 were classified as lowly expressed and the top 2000 as highly expressed genes. 2000 genes with zero expression were designated as silent genes. ChIP-seq H3K27me1/2/3 signals were MS-normalized as previously described. **b** Barplots of genome-wide prevalence of modifications based on mass spectrometry, showing progressive depletion of H3K27me1, and progressive increases in H3K27me2 and H3K27me3 levels in multi-KO conditions reaching approximately 1.5-fold and more than 2.5-fold, respectively. Error bars display the standard deviation around the mean. ($n$ = 3 per condition). **c** Log2 fold changes of gene expression comparing PA to TKO, depicting 1.5-fold more genes being downregulated (609) than upregulated (391) following significant H3K27me3 invasion (adjusted $p$-value < 0.05) into their gene bodies compared to a randomized, control set of genes, which have similar numbers of genes up- (503) and down-regulated (497). Statistical significance was tested using a two-sided Wilcoxon rank-sum test. **** represents $p$-value = 4.9e-13. In the box plots, boxes span the lower (first quartile) and upper quartiles (third quartile), median is indicated with a center line, and whiskers extend to a maximum of 1.5 times the interquartile range. Source data are provided as a Source Data file.

increase of genic H3K36me2 due to fewer residues being upgraded to the higher methylation state[26] (Supplementary Fig. 3a). Since the presence of H3K36me2 also hinders the activity of PRC2 to deposit higher orders of H3K27me[33], these results suggest that the presence of H3K36me3 inhibits the deposition of both H3K27me2/3, while the presence of H3K36me2 may primarily inhibit the deposition of H3K27me3.

Therefore, we next focused specifically on the ability of H3K36me2 to protect gene bodies from methylation at H3K27. In NSD1/2-DKO cells, the levels of H3K36me2 are significantly reduced, with a moderate reduction of H3K36me3 (Supplementary Fig. 3a). This is accompanied by a significant increase in the global levels of H3K27me2/3 (Fig. 3b). Interestingly, the partial reduction of H3K36me2 has a more pronounced effect on genic H3K27me2/3 than

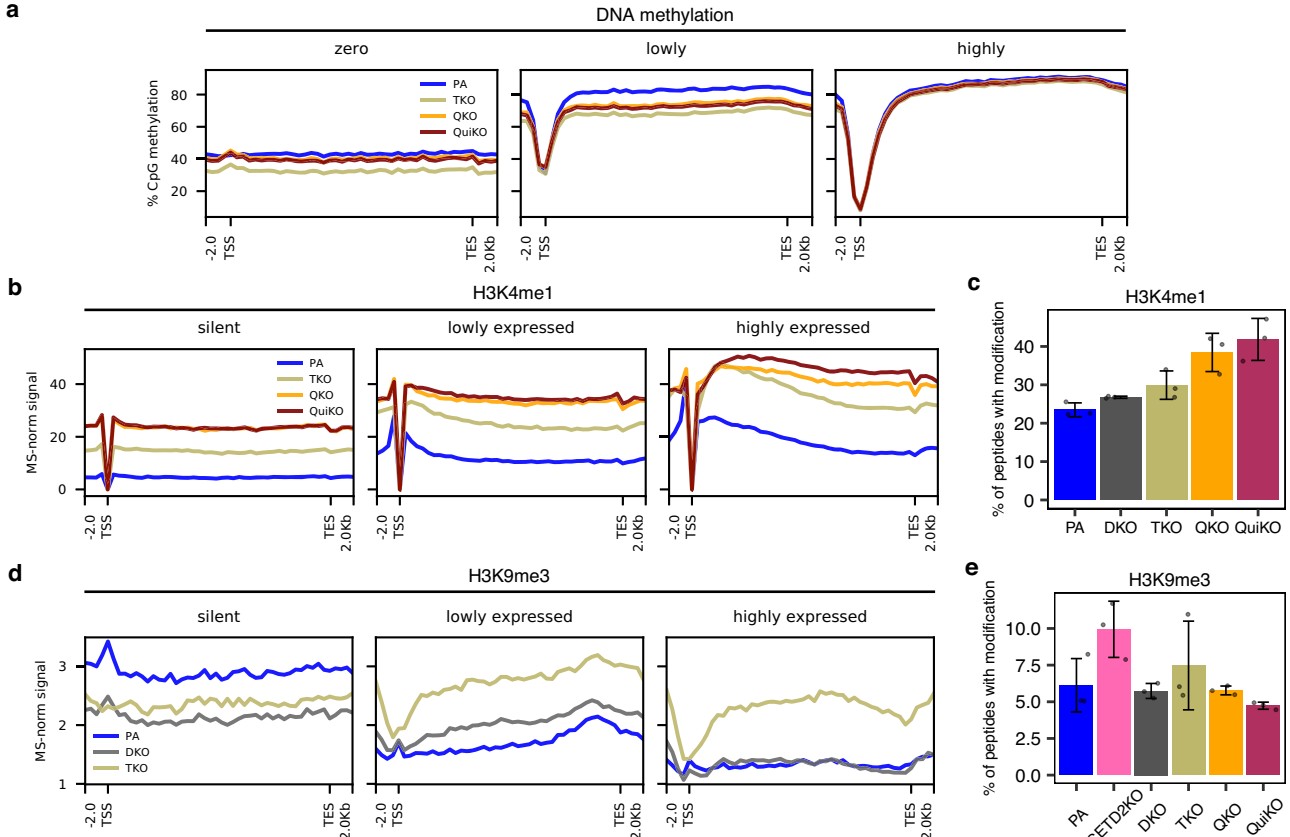

**Fig. 4 | Depletion of H3K36me induces changes in DNA methylation, H3K4me1 and H3K9me3 within gene bodies. a** Metagene plots of DNA methylation within gene bodies (same groups of genes used in Fig. 3), illustrating a decrease in DNA methylation in unexpressed and lowly expressed genes. **b** Metagene plots indicating increasing H3K4me1 within gene bodies following depletion of H3K36me. **c** Barplot illustrating the increasing global abundance of H3K4me1, as determined by mass spectrometry (MS), following successive deletions of K36MTs. **d** Metagene

plots indicating invasion of H3K9me3 within gene bodies following depletion of H3K36me2 in DKO and TKO cell lines. **e** Barplot derived from MS, showing that genome-wide H3K9me3 abundance does not significantly change across KO cell lines. For **b** and **d**, CUT&RUN or ChIP-seq signals, respectively, were MS-normalized as previously described. For **c** and **e**, error bars display the standard deviation around the mean. (*n* = 3 per condition). Source data are provided as a Source Data file.

what is observed with the near total depletion of H3K36me3 (Fig. 3a; middle and bottom rows). Although in vitro studies indicate a stronger antagonistic effect from H3K36me3[33], the greater observable effect of H3K36me2 depletion may reflect the generally higher absolute levels of H3K36me2 than H3K36me3 in gene bodies.

Finally, we used NSD1/2-SETD2-TKO cells to study the consequences of depleting both H3K36me2/3. Expectedly, in these cells we observe the strongest invasion of H3K27me2/3 into gene bodies (Fig. 3a; middle and bottom rows). Both H3K27me2/3 now appear free to populate genes, with little dependence on expression levels (Fig. 3a). Further depletion of H3K36me2 in our QKO and QuiKO cell lines leads to even greater invasion of H3K27me3 into gene bodies (Supplementary Fig. 3b). Interestingly, the cells overexpressing the H3K36M oncohistone (H3K36M-OE), which reduces both H3K36me2/3, show invasion of H3K27me that is intermediate between the DKO and TKO conditions (Fig. 3a, b).

To better illustrate the global changes to the distribution of H3K27me across the various conditions, we calculated H3K27me3 'peakiness scores', which represents the average ChIP-seq/CUT&RUN signal in the top 1% of 1-kb bins over the total signal, confirming the broadening (lower peakiness score) of H3K27me3 following sequential removal of H3K36me (Supplementary Fig. 3c). The lower peakiness scores in the multiple-KO conditions demonstrates that as H3K36me is sequentially removed, the distribution of H3K27me3 shifts from a focused distribution in parental cells, to a broader distribution in cells lacking H3K36me. Lastly, more than 1.5-fold of genes with significant

H3K27me3 invasion in their gene body decrease in expression (609 down-regulated compared to 391 upregulated), further supporting the hypothesis that the invasion of H3K27me3 into gene bodies contributes to suppressing gene expression (Fig. 3c).

Overall, while our results demonstrate that H3K36me plays a role in excluding H3K27me2/3 from gene bodies, additional factors are clearly responsible for preventing the full invasion of H3K27me. The two likely culprits are 1) nucleosome turnover, which results in replacement of modified with naïve nucleosomes during transcription[42], and 2) DNAme, which may hinder the recognition of CpG-rich sequences, such as those found in exons, by PRC2[43,44].

## The effect of H3K36 methylation on other epigenetic modifications within gene bodies

As previously described, H3K36me2/3 have been shown to serve as substrates for the localization of DNMT3A/B to deposit de novo DNAme[38]. Therefore, we expected that in the absence of H3K36me2/3, gene bodies may experience substantial reductions in CpG methylation. However, we find that even in the QuiKO cells, which lack nearly all H3K36me, the most highly expressed genes retain DNAme levels comparable to parental cells (Fig. 4a; third box). Methylation is visibly reduced in the lowest expression quantile, as well as in unexpressed genes – where the latter behave similarly to inactive intergenic regions where DNA hypomethylation has previously been shown to be associated with the reduction of intergenic H3K36me2 (Fig. 4a; first and second box)[3,18,38]. It appears, however, that in highly expressed genes,

H3K36me-mediated targeting of de novo DNMTs is unnecessary for the maintenance of high DNAme levels.

Within the gene bodies of the K36MT mutants, we detected no discernible changes in H3K27ac or H3K4me3, regardless of transcription activity (Supplementary Fig. 4a, b). In contrast, H3K4me1 levels increase substantially within gene bodies, with the degree of H3K36me2 depletion correlating with greater invasion of H3K4me1− even in previously silent genes and with minimal dependence on transcription (Fig. 4b). Consistent with these findings, we observe by MS that the global abundance of H3K4me1 increases in the multiple-KO conditions (Fig. 4c). Additionally, the distribution of H3K4me1 becomes increasingly diffuse across the multiple-KO conditions, as reflected by a progressive decrease in 'peakiness' scores (Supplementary Fig. 4c). Interestingly, this global increase in H3K4me1 is accompanied by alterations in the expression of the primary H3K4 methyltransferases: KMT2C and KMT2D. Specifically, KMT2C expression is reduced, while KMT2D expression is moderately upregulated in the multiple-KO mMSCs, raising the possibility that KMT2D may be responsible for the ectopic distribution of H3K4me1 (Supplementary Fig. 4d).

Surprisingly, we also found significant changes to the genic levels of the silencing-associated modification H3K9me3, whose relationship with H3K36me is currently unclear. In parental cells, H3K9me3 is generally excluded from transcribed genes, however, gradual removal of H3K36me results in an increase of H3K9me3 that appears to be correlated with both gene expression and the amount of residual H3K36me2/3 (Fig. 4d, Supplementary Fig. 3a). In DKO cells, where the loss of NSD1 and NSD2 substantially reduces H3K36me2, genic H3K9me3 increases, especially in lowly expressed genes (Fig. 4d; second box). In TKO cells, which also lack SETD2 and thereby most H3K36me3, the gene body levels of H3K9me3 increase further and appear to lose their dependence on transcription (Fig. 4d). These results intriguingly suggest that the presence of H3K36me2/3 within genes may antagonize the deposition of H3K9me3 in these regions. Given the considerable increase in genic H3K9me3 in the TKO cells, we used MS to quantify the global abundance of H3K9me3 across conditions. Interestingly, we find that the bulk levels of H3K9me3 increase following the depletion of H3K36me3 in both SETD2-KO and TKO cells (Fig. 4e), suggesting that H3K36me3, in particular, may antagonize the deposition of H3K9me3 within genes. However, the genome-wide abundance of H3K9me3 in the QKO and QuiKO conditions returns to levels comparable to parental cells, and do not necessarily reflect the observed increases in active genes (Fig.4e, Supplementary Fig. 4e). Overall, despite the increase of H3K9me3 in actively transcribed genes, the genome-wide abundance in the multiple-KO conditions does not significantly change. Thus, we hypothesized that there must be a corresponding decrease in other regions of the genome.

## Loss of H3K36me leads to SUV39H1-mediated redistribution of H3K9me3 from heterochromatin to euchromatin

In parental mMSCs, the distribution of H3K9me3 follows two characteristic patterns: at a broad scale, it is deposited in large domains spanning several megabases (Fig. 5a), corresponding to highly heterochromatic, transcriptionally silent, lamina-associated domains[6,45,46]. At a finer scale, peaks of H3K9me3 are present at the promoters of certain silent genes and transposable elements (TEs) (Fig. 4d; 'silent genes'). Remarkably, under H3K36me deficient conditions, these broad H3K9me3 domains entirely disappear (Fig. 5a). The depletion of heterochromatic domains is accompanied by increased H3K9me3 deposition in regions that are predominantly active, which, upon closer examination, are primarily gene bodies and TEs (Fig. 4d; Supplementary Fig. 5a). To identify and further characterize the genomic compartments with the greatest loss of H3K9me3, we subdivided the genome into 100-kb bins and compared profiles between parental and TKO cells, which are largely devoid of H3K36me. The H3K9me3

profiles of the genomic bins subdivide into two distinct clusters (Fig. 5b): cluster A (orange), comprising predominantly genic regions that gain H3K9me3 in TKO cells, and cluster B (blue) consisting mainly of intergenic regions that lose H3K9me3 in TKO cells.

To determine whether this redistribution of H3K9me3 in response to H3K36me loss is conserved in other cell types, we examined two human head and neck squamous cell carcinoma (HNSCC) cell lines: Cal27 and Detroit562. These HPV(-) HNSCC cell lines lack endogenous mutations affecting H3K36me[18]. In these cells, we overexpressed (OE) the H3K36M oncohistone−a mutation known to globally reduce all H3K36 methylation states and found in HPV(-) HNSCC tumors as well as other cancers (Supplementary Fig. 5b)[14,24,47]. As in mMSCs, parental Cal27 and Detroit562 cells display well-defined H3K9me3-marked heterochromatic domains (Fig. 5c, Supplementary Fig. 5c). Strikingly, overexpression of H3K36M in both cell lines resulted in marked depletion of these large H3K9me3 domains, corresponding to cluster B regions identified by our genome-wide density-based clustering analysis (Fig. 5b–d, Supplementary Fig. 5c–e). Furthermore, unlike in mMSCs, the loss of H3K9me3 domains in these cells was accompanied by a global reduction in H3K9me3 levels of approximately 35% (Supplementary Fig. 5b). These findings suggest that H3K36 methylation plays a conserved role in maintaining higher-order H3K9me3 organization across cell types.

Using the cluster A and B annotations, we next investigated the molecular mechanisms underlying H3K9me3 redistribution following H3K36me depletion. Specifically, we focused on SUV39H1, one of the primary H3K9 methyltransferases (K9MTs) which catalyzes H3K9me3 and plays a critical role in the propagation of H3K9me3 domains through a read-write mechanism, wherein its chromodomain recognizes existing H3K9me3 to enhance further methyltransferase activity[48,49]. Unlike SETDB1, which deposits focal peaks at repetitive elements, and SUV39H2, which is testis-enriched and minimally expressed in somatic cells, SUV39H1 has both the expression profile and enzymatic behavior consistent with widespread ectopic H3K9me3 deposition (Supplementary Fig. 5f)[50,51]. To profile the genomic distribution of SUV39H1, we overexpressed FLAG-tagged SUV39H1 in parental and TKO mMSCs and performed ChIP-Seq for FLAG-SUV39H1 (Supplementary Fig. 5g)[52]. In parental mMSCs, SUV39H1 exhibited enrichment patterns similar to H3K9me3 and was primarily localized to inactive cluster B regions (Fig. 5e, f; Supplementary Fig. 5h). In the TKO cells, however, SUV39H1 relocates to cluster A regions, with reduced occupancy in cluster B regions (Fig. 5e, f; Supplementary Fig. 5h). Additionally, SUV39H1 occupancy increased at genes that gained H3K9me3 in TKO cells (Fig. 5g). These results demonstrate that SUV39H1 relocalizes to the same genomic loci where ectopic H3K9me3 accumulates in the absence of H3K36me.

To test whether SUV39H1 contributes to both normal and ectopic H3K9me3 deposition, we knocked down (KD) SUV39H1 using shRNAs (Supplementary Fig. 5i). In parental cells, SUV39H1-KD resulted in a global H3K9me3 reduction of approximately 40%, primarily within cluster B regions (Fig. 5h; Supplementary Fig. 5j). Notably, SUV39H1-KD in parental cells preferentially depleted the broad heterochromatic domains, as evidenced by a reduction in the proportion of H3K9me3 signal within cluster B and a concomitant increase in cluster A (Fig. 5h, i). In contrast, in TKO cells, SUV39H1-KD reduced global H3K9me3 levels to a similar extent, but the depletion was comparable across both cluster A and B regions (Fig. 5h, i; Supplementary Fig. 5j). These findings support that SUV39H1 is primarily responsible for maintaining broad heterochromatic domains in parental cells, whereas in the absence of H3K36me, it redistributes genome-wide and mediates ectopic H3K9me3 deposition.

To further demonstrate that SUV39H1 becomes enriched in euchromatic cluster A regions following H3K36me loss and contributes to ectopic H3K9me3 deposition, we examined H3K9me3 distribution following SUV39H1 overexpression (OE) in mMSCs. In

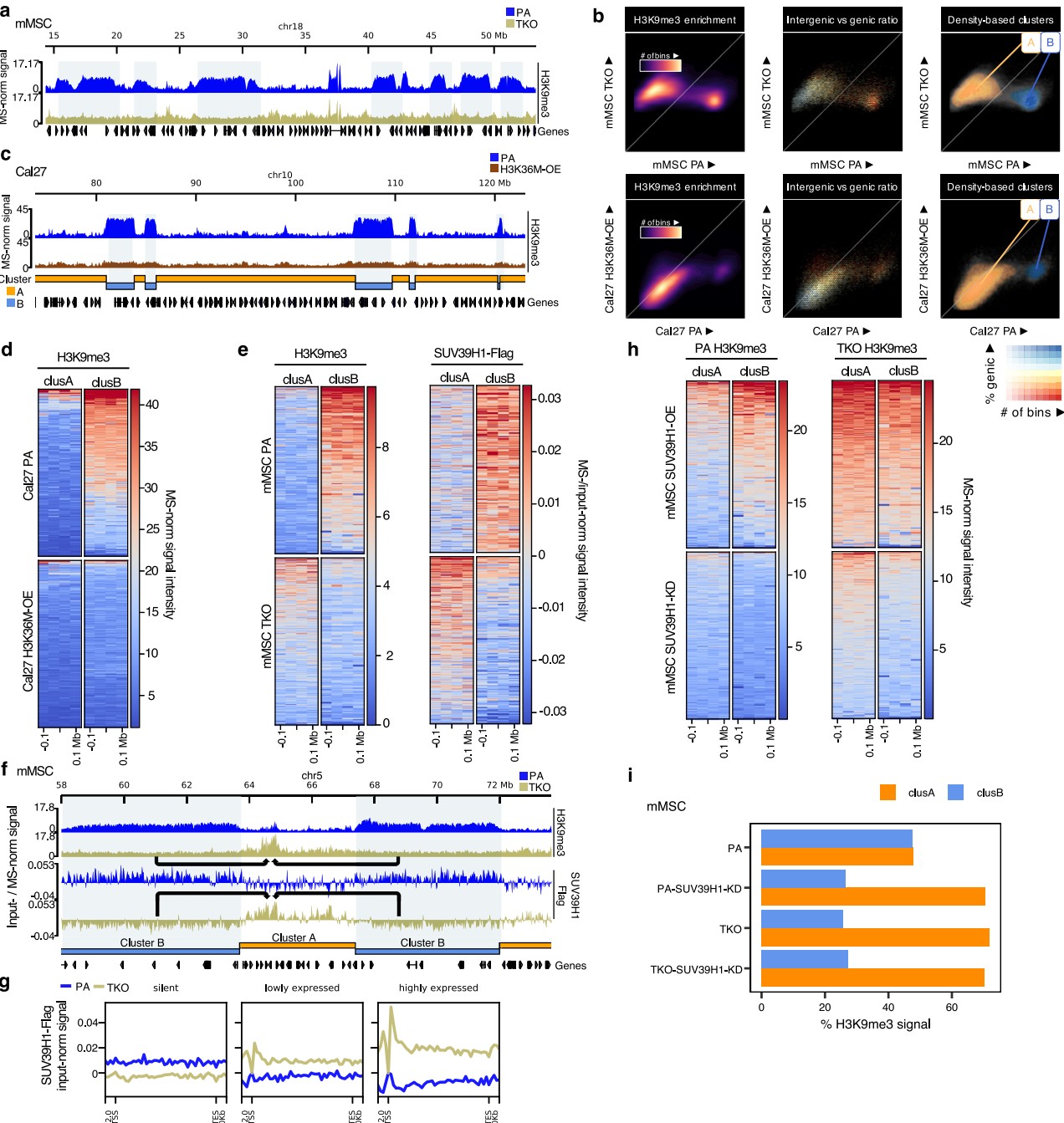

**Fig. 5 | Loss of H3K36me leads to a redistribution of SUV39H1-mediated H3K9me3 from heterochromatic regions to euchromatin. a** Genome browser tracks illustrating regions of H3K9me3 loss in TKO mMSCs. **b** Scatterplots of 100kb-binned H3K9me3 signal, comparing parental against TKO mMSCs, and parental against H3K36M-OE Cal27−indicating the formation of two clusters: cluster A (regions that gain H3K9me3) and cluster B (regions that lose H3K9me3). **c** Genome browser tracks illustrating loss of large H3K9me3 domains in H3K36M-OE Cal27 cells. Cluster B regions (blue) and Cluster A regions (orange) are indicated. **d** Heatmaps centered on cluster A and B regions, illustrating loss of H3K9me3 in cluster B regions in H3K36M-OE Cal27 cells. **e** Heatmaps centered on cluster A and B regions illustrating a reduction in H3K9me3 (left heatmap) and SUV39H1 (right heatmap) signal within cluster B, whereas cluster A regions exhibit increased H3K9me3 and SUV39H1 enrichment in TKO mMSCs. **f** Genome browser tracks showing a redistribution of H3K9me3 from cluster B to cluster A regions in TKO mMSCs. Similarly, SUV39H1 is redistributed to cluster A regions, with reduced

occupancy in cluster B regions in TKO mMSCs. **g** Aggregate plots centered on gene bodies stratified by gene expression levels, demonstrating enrichment of SUV39H1 in expressed genes in TKO mMSCs. **h** Heatmaps showing that in parental mMSCs, SUV39H1 overexpression (OE) modestly increased H3K9me3 in cluster A but remained primarily enriched in inactive cluster B regions. In TKO cells, following SUV39H1-OE, higher levels of H3K9me3 are found in both clusters, especially in cluster A. SUV39H1 knockdown (KD) primarily reduced H3K9me3 in cluster B of parental cells, while in TKO cells, depletion was similar in both clusters. **i** Barplots showing the percentage of H3K9me3 signal in cluster A and B regions, indicating that in mMSC parental cells, SUV39H1-KD decreases the signal in cluster B regions with a concomitant increase in cluster A. In TKO mMSCs, SUV39H1-KD reduces H3K9me3 levels evenly across both clusters, with no shift in signal proportion. Normalized signals were either MS-normalized (H3K9me3) or input-normalized (SUV39H1-Flag), as previously described. Source data are provided as a Source Data file.

parental cells, SUV39H1-OE led to a modest increase in H3K9me3 within cluster A, but enrichment remained primarily restricted to inactive cluster B regions and silent genes (Fig. 5h; Supplementary Fig. 5k). By contrast, in TKO SUV39H1-OE cells, H3K9me3 levels increased substantially across both clusters A and B, with cluster A regions—including genes that gained H3K9me3—becoming especially enriched (Fig. 5h; Supplementary Fig. 5k). Collectively, these findings reveal a previously underappreciated mechanistic dependency between the H3K36 and H3K9 methylation pathways. Loss of H3K36me redirects SUV39H1 from transcriptionally silent cluster B regions into euchromatic cluster A regions, where it contributes to the aberrant deposition of H3K9me3.

## Downstream epigenetic cascade following the depletion of H3K9me3 decorated heterochromatin

The loss of broad H3K9me3 domains in H3K36me-deficient cells suggests a fundamental disruption of constitutive heterochromatin. Therefore, we next asked how the chromatin landscape is remodeled in their absence. In normal cells, following the deposition of H3K9me3, heterochromatin protein 1 (HP1) binds to the methylated histone tail and promotes chromatin compaction and gene silencing by recruiting additional repressive factors and facilitating the spread of heterochromatin[48,53]. To assess changes in HP1 binding, we performed ChIP-seq. As expected, in parental mMSCs, HP1 was predominantly enriched in inactive cluster B regions (Fig. 6a, b). However, in TKO mMSCs, following SUV39H1-mediated H3K9me3 deposition in cluster A regions (Fig. 5e, f), we observed corresponding recruitment of HP1 to these euchromatic domains (Fig. 6a, b; Supplementary Fig. 6a), consistent with established H3K9me3-HP1 binding dynamics. Notably, HP1 accumulation in these regions was particularly enriched within gene bodies and TEs (Fig. 6c; Supplementary Fig. 6b).

Conversely, in cluster B regions where H3K9me3 is depleted and HP1 binding is lost, we observed the invasive broadening of H3K27me3 (Fig. 6a, b). In parental mMSCs, large heterochromatic, gene-poor regions are characterized by "convex" H3K9me3 domains flanked by reciprocal "concave" domains of H3K27me3 (Supplementary Fig. 6c). These regions are transcriptionally inactive and lack open chromatin sites or other active epigenetic marks (Supplementary Fig. 6c). In the absence of H3K36 methylation in TKO cells, H3K9me3 is lost, and many of these regions become occupied by H3K27me3, possibly as a compensatory silencing mechanism (Fig. 6a, b; Supplementary Fig. 6c). Thus, in addition to invading gene-rich, active regions (cluster A), H3K27me3 also invades gene-poor, inactive regions that lose H3K9me3 (cluster B), suggesting a shift from constitutive to facultative heterochromatin (Fig. 6b; Supplementary Fig. 6c, d). Similarly, in Cal27 and Detroit562 H3K36M-OE cells, H3K27me3 spreads into both euchromatic cluster A regions and heterochromatic cluster B regions depleted of H3K9me3 (Supplementary Fig. 6e, f). In both cell lines, these chromatin changes are accompanied by a global increase in H3K27me3 (Supplementary Fig. 6g) and a corresponding decrease in H3K36me and H3K9me3 (Supplementary Fig. 5b).

Building on these observations, we next examined whether the loss of H3K9me3 and the invasion of H3K27me3 are also associated with changes in chromatin accessibility and regulatory potential. We observed the emergence of newly accessible regions, as measured by ATAC-Seq, within cluster B regions under H3K36me-deficient conditions. In mMSCs, we identified 1914 new open chromatin regions in TKO compared to parental cells (Fig. 6d, e). Interestingly, most of these newly accessible regions are substantially smaller than typical open chromatin peaks found at active promoters and enhancers (Fig. 6d), suggesting they may represent focal disruptions of chromatin compaction rather than canonical regulatory elements. Similarly, in Cal27 and Detroit562 H3K36M-OE cells, we observed new open chromatin peaks in regions that lose H3K9me3, with 1008 new peaks identified in Cal27 (Fig. 6f, g) and 481 new peaks in Detroit562 (Supplementary

Fig. 6h, i). In TKO mMSCs, the gain in chromatin accessibility is accompanied by a significant increase in H3K4me1, and in some regions, H3K27ac (Fig. 6h). This accumulation of histone marks associated with enhancer-like chromatin further supports the global increase of H3K4me1 observed by mass spectrometry (Fig. 4c). Notably, in addition to its previously observed accumulation within gene bodies under H3K36me-deficient conditions (Fig. 4b), H3K4me1 also becomes enriched in regions that lose H3K9me3 (Fig. 6h), suggesting a broader redistribution of enhancer-associated marks into formerly heterochromatic compartments. Furthermore, these newly accessible regions exhibit reduced DNAme, consistent with a release from constitutive repression (Supplementary Fig. 6j). RNA-Seq analysis further reveals increased transcriptional activity within these regions (Fig. 6h). Similarly, in Cal27 and Detroit562 H3K36M-OE cells, the newly accessible chromatin regions exhibit elevated H3K27ac levels and increased transcriptional output (Fig. 6i; Supplementary Fig. 6k), reinforcing that the loss of H3K36me triggers a conserved epigenomic response across cell types—one that includes the collapse of many constitutive heterochromatin domains and the acquisition of regulatory features characteristic of active chromatin.

Together, these findings reveal that loss of H3K36me leads to widespread epigenomic restructuring, characterized by the erosion of constitutive heterochromatin and the emergence of enhancer-like features in normally silenced regions. This includes a coordinated gain of chromatin accessibility, activation-associated histone marks, and transcriptional de-repression. These results underscore a critical role for H3K36me, and/or the respective K36MTs, in preserving heterochromatin integrity and highlight the remarkable plasticity of silent chromatin in response to disruptions to the balance of histone methylation marks.

## H3K9me3 loss drives transcriptional activation despite compensatory H3K27me3 accumulation

Despite the compensatory accumulation of H3K27me3 in cluster B regions following H3K9me3 loss, these repressive changes appear insufficient to fully maintain transcriptional silencing. Partial correlation analysis reveals that although H3K27me3 levels increased within cluster B regions, this gain did not offset transcriptional activation and was not significantly associated with gene expression changes (Supplementary Fig. 7a). By contrast, H3K9me3 loss exhibited a strong correlation with gene upregulation in cluster B regions (Supplementary Fig. 7a), implicating this mark as the primary regulator of transcriptional changes at these loci. We next sought to characterize the genes undergoing H3K9me3 depletion. In TKO mMSCs, approximately 90% (2119/2354) of genes in these regions lose H3K9me3 (Fig. 7a). Among these, genes with the most pronounced loss of H3K9me3 become upregulated (157/179 genes) (Fig. 7b). A similar pattern is observed in Cal27 and Detroit562 H3K36M-OE cells, where genes within cluster B domains lose H3K9me3 and are significantly upregulated relative to an expression-matched random control gene set (Fig. 7a; Supplementary Fig. 7b).

To better understand the biological consequences of H3K9me3 depletion from cluster B regions, we next examined the subset of genes that lose H3K9me3 in both mMSC and HNSCC models to determine whether common regulatory targets or pathways are affected across species. Among the genes that lost H3K9me3 in cluster B regions in both mMSC TKO and Cal27 H3K36M-OE cells (Fig. 7a), 132 genes were shared across the two models (Fig. 7c). Remarkably, approximately 35% (46/132) of these genes encode olfactory receptors (ORs) (Fig. 7c), consistent with prior findings that OR gene clusters are densely marked by H3K9me3 and subject to tight epigenetic repression in the mouse genome[54]. Indeed, KEGG[55] pathway analysis of the 132 genes indicates olfactory transduction to be the most significantly overrepresented pathway (Fig. 7d). Complementary Gene Ontology (GO)[56] analysis confirmed this pattern, with smell-related terms

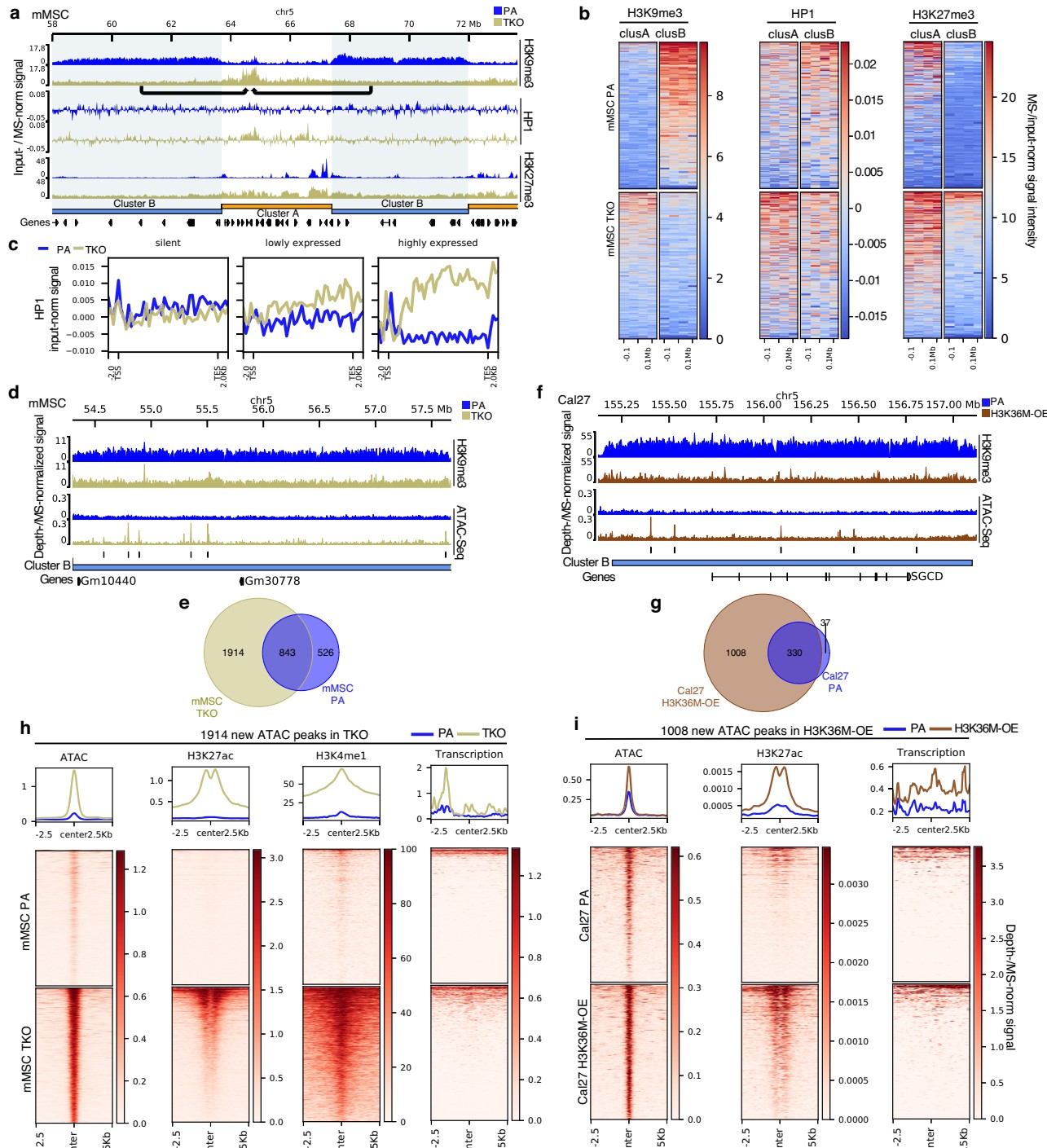

**Fig. 6 | Loss of H3K9me3 in previously inactive heterochromatin leads to epigenome restructuring. a** Genome browser tracks of a representative genomic region, demonstrating that in TKO mMSCs, HP1 redistributes to cluster A regions whereas H3K27me3 broadens into cluster B regions. **b** Heatmaps centered on clusters A and B regions demonstrating a redistribution of HP1 from cluster B to A regions, whereas H3K27me3 broadens from cluster A into B regions following TKO in mMSCs. **c** Aggregate plots showing increased HP1 signal in expressed genes following TKO in mMSCs. **d** Genome browser tracks depicting newly accessible regions within cluster B regions in mMSC TKO cells. **e** Venn diagram comparing the number of ATAC-Seq peaks found in cluster B regions, where 1914 de novo accessible regions appear following TKO in mMSCs. **f** Genome browser tracks

depicting newly accessible regions within cluster B regions following H3K36M-OE in Cal27 cells. **g** Venn diagram comparing the number of ATAC-Seq peaks found in cluster B regions, where 1008 de novo accessible regions appear following H3K36M-OE in Cal27 cells. **h** Heatmaps centered on the 1914 new ATAC-Seq peaks opening in mMSC TKO cells, indicating an upregulation of active marks (H3K27ac and H3K4me1) and transcriptional activity. **i** Heatmaps centered on the 1008 new ATAC-Seq peaks opening in Cal27 H3K36M-OE cells, indicating an upregulation of H3K27ac and transcriptional activity. Normalized signals were either input-normalized (HP1), MS-normalized (for H3K27me3, H3K9me3, H3K27ac and H3K4me1), depth-normalized (ATAC-Seq) or RPKM (reads per kilobase per million mapped) transformed.

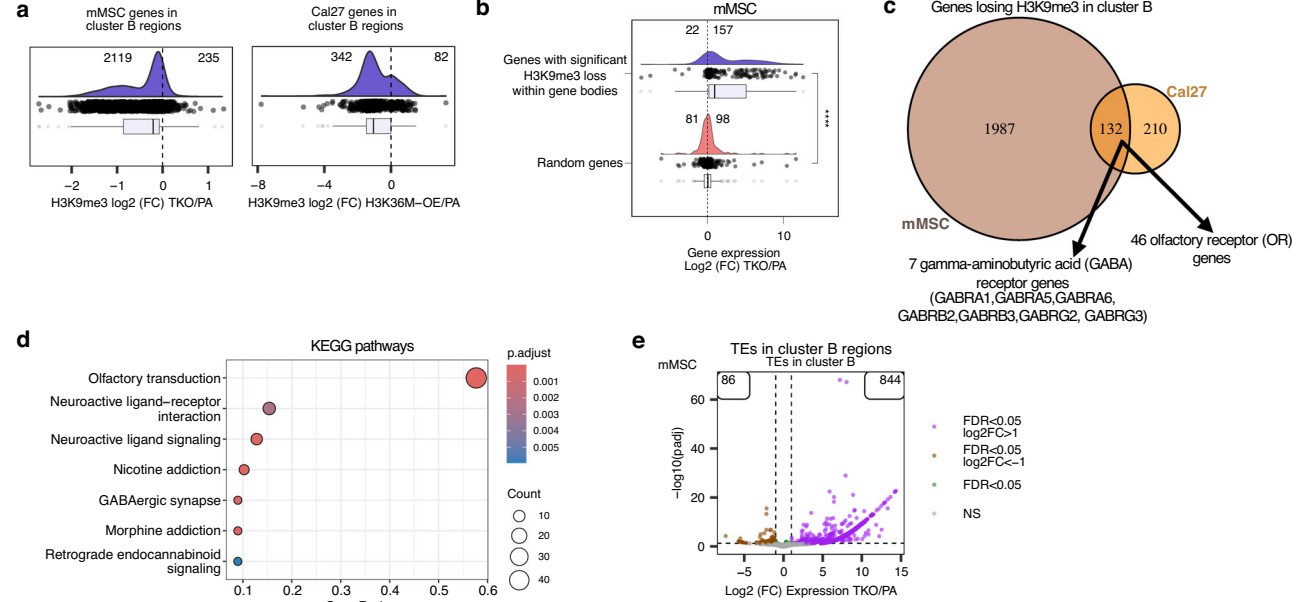

**Fig. 7 | Upregulation of genes and transposable elements in regions of H3K9me3 loss. a.** Log2-fold changes of H3K9me3 signal comparing TKO mMSCs and Cal27 H3K36M-OE cells to their respective parental controls. Following TKO in mMSC, over 90% of genes in cluster B regions (2119/2354) lose H3K9me3 in their gene bodies whereas following H3K36M-OE in Cal27, 81% of genes (342/424) lose H3K9me3 in cluster B regions. **b** Log2-fold changes of gene expression comparing TKO to parental mMSCs, showing that most of the genes in cluster B regions with significant H3K9me3 loss become upregulated (157 upregulated versus 22 down-regulated), whereas a randomized control set of genes have similar numbers of genes up- and downregulated (98 and 81, respectively). A two-sided Wilcoxon rank-sum test was used. **c** Venn diagram comparing the genes losing H3K9me3 in TKO mMSCs and Cal27 H3K36M-OE cells, respectively, indicates 132 genes common to both conditions; 46 of which are olfactory receptor genes and 7 are gamma-aminobutyric acid (GABA) receptor genes. **d** Over-representation analysis of KEGG pathways, confirming olfactory transduction as the top enriched pathway for the genes commonly found between TKO mMSCs and Cal27 H3K36M-OE cells that are losing H3K9me3. **e** Volcano plots showing a large number of transposable elements become upregulated in regions of H3K9me3 loss (cluster B domains) in the TKO mMSCs. **** refers to *p*-value = 7.7e-22 from a Wilcoxon rank-sum test. In the box plots for **a** and **b**, boxes span the lower (first quartile) and upper quartiles (third quartile), median is indicated with a center line and whiskers extend to a maximum of 1.5 times the interquartile range. Source data are provided as a Source Data file.

ranking in the top enriched categories (Supplementary Fig. 7c). In addition to ORs, another gene family significantly enriched among the shared H3K9me3-depleted loci are gamma-aminobutyric acid (GABA) receptor genes (Fig. 7c, d; Supplementary Fig. 7c). Although traditionally associated with neuronal function, GABA receptor genes are increasingly recognized for their broader roles in cellular signaling, and there is emerging evidence that H3K9me3 contributes to their transcriptional silencing[57]. Notably, GABA receptor activity influences the expression of brain-derived neurotrophic factor (BDNF), which itself is negatively regulated by H3K9me3 in mouse models[58].

In mMSCs, numerous TEs located in cluster B regions also become differentially expressed, with a strong bias toward upregulation (844 upregulated versus 86 downregulated) (Fig. 7e). Further analysis reveals that 44 TE subfamilies are upregulated, compared to only 9 that are downregulated, with the majority of upregulated TEs belonging to the endogenous retrovirus (ERV) family (Supplementary Fig. 7d). Interestingly, this widespread dysregulation of TEs is not observed in Cal27 or Detroit562 cells (data not shown). This discrepancy may reflect fundamental species-specific differences in TE content and regulation. The mouse genome harbors a higher proportion of young and transcriptionally competent ERVs, which rely heavily on H3K9me3 and SETDB1 for silencing, particularly in stem and progenitor cellular contexts[59,60]. In contrast, the majority of ERVs in the human genome are older, more degenerate, and often rendered transcriptionally inactive by fixed mutations, DNA methylation, or Polycomb-mediated H3K27me3, thereby reducing their reliance on H3K9me3 for repression[61–63]. These findings suggest that while gene de-repression following the loss of H3K9me3 is conserved across species, TE activation is a more prominent feature in murine cells.

Taken together, these results demonstrate that the loss of H3K9me3 contributes to focal derepression of select gene families and transposable elements in a species- and context-specific manner, and that the compensatory accumulation of H3K27me3 in cluster B regions is insufficient to restore transcriptional silencing.

## H3K36me is essential for maintenance of nuclear architecture and long-range chromatin interactions

At a fundamental level, chromatin is spatially organized into active and inactive compartments, typically referred to as A and B compartments, respectively[64]. Given the widespread alterations in both euchromatic and heterochromatic histone modifications following H3K36me depletion, we examined whether chromatin conformation and nuclear compartmentalization were similarly affected. To test this, we performed Hi-C on parental and QuiKO mMSCs, which lack all H3K36me, as well as on parental and H3K36M-OE Cal27 and Detroit562 cells.

Parental mMSCs have distinct A/B compartments and euchromatin/heterochromatin structure, as illustrated by the strong checkerboard pattern at the chromosomal scale (Fig. 8a; upper diagonal). The inactive B compartments are highly correlated with H3K9me3 domains (Supplementary Fig. 8a) and display significant B-B compartment interactions (Fig. 8b), indicative of strong heterochromatin compaction. Notably, the frequency of B-B interactions is nearly double that of A-A interactions in parental cells (Fig. 8b). In contrast, QuiKO cells exhibit a drastic disruption in chromatin compartmentalization, coinciding with the loss of H3K9me3 domains (Fig. 8a; lower diagonal). This is reflected by a twofold reduction in both B-B and A-A interactions (Fig. 8b) and a homogenization of compartment scores resulting in the collapse of the bimodal nature of compartment

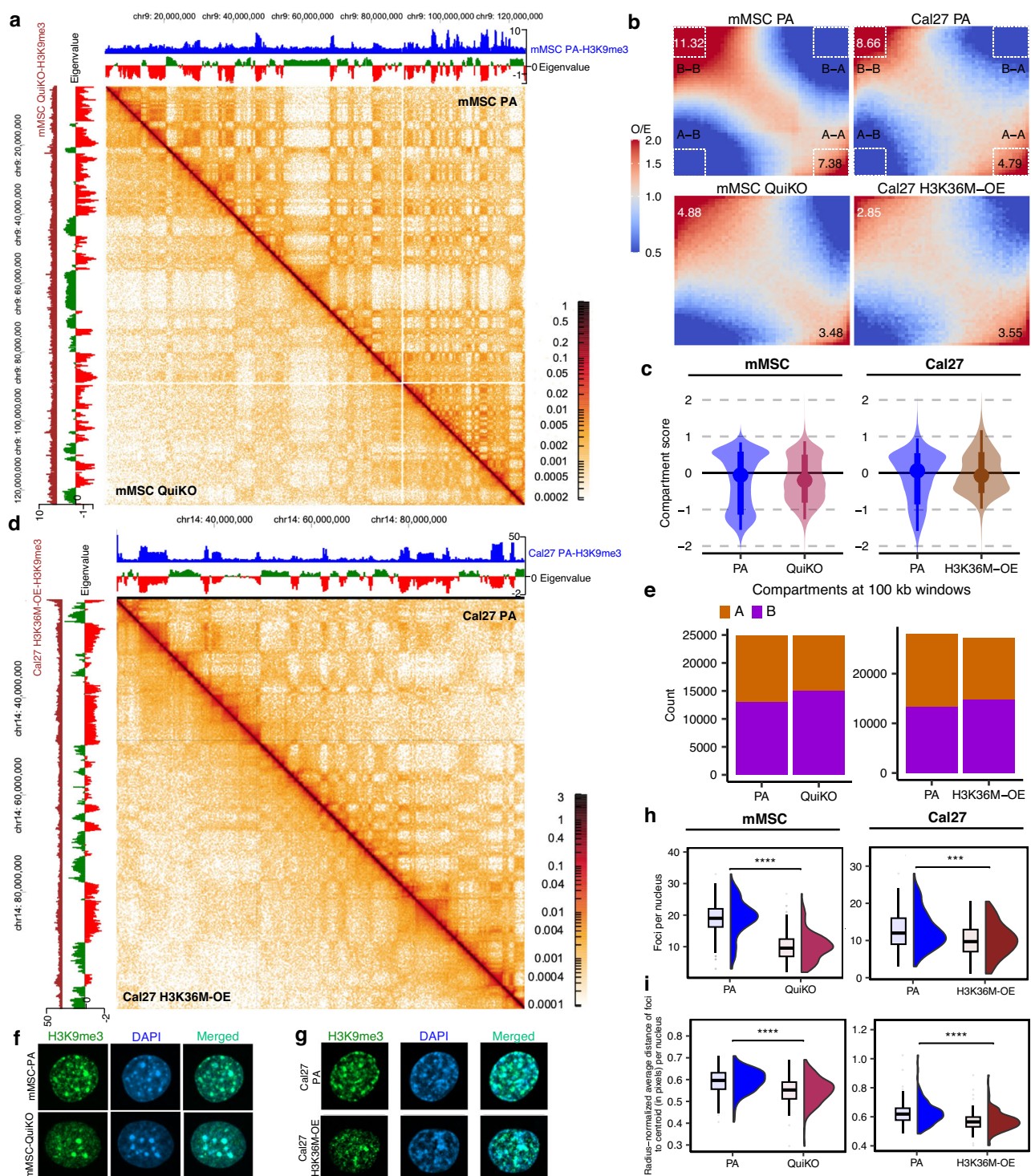

identity, particularly affecting B-compartments (Fig. 8c). A similar loss of compartment integrity is observed in human HNSCC cell lines. In Cal27 and Detroit562, parental cells exhibit distinct A/B compartmentalization (Fig. 8d; upper diagonal and Supplementary Fig. 8b; upper diagonal), which becomes attenuated upon H3K36M-OE (Fig. 8d; lower diagonal and Supplementary Fig. 8b; lower diagonal). In Cal27 H3K36M-OE cells, B-B interactions decrease threefold, while A-A interactions are reduced by ~30% (Fig. 8b; second column). Similarly, Detroit562 H3K36M-OE cells show an ~40% reduction in B-B interactions and ~25% reduction in A-A interactions (Supplementary Fig. 8c). In both HNSCC cell lines, compartment scores shift toward

zero, indicating a global weakening of compartmental identity (Fig. 8c; Supplementary Fig. 8d).

Across all three cell types, the total number of A and B compartments remains largely unchanged between parental and H3K36me-depleted conditions, although a modest shift from A to B compartments is observed (Fig. 8e; Supplementary Fig. 8e). Nonetheless, a more detailed analysis reveals that in mMSCs, long-range *cis*-interactions (>10 Mb) are particularly diminished (Supplementary Fig. 8f), suggesting weakened chromosomal compartmentalization at extended genomic distances. To evaluate additional aspects of genome architecture, we assessed topologically associating domains (TADs)

**Fig. 8 | Depletion of H3K36me results in decreased nuclear compartmentalization. a** Hi-C heatmaps illustrating reduced nuclear compartmentalization in QuiKO mMSCs. **b** Saddleplots showing decreased A-A and B-B *cis* compartment interactions in QuiKO mMSCs and Cal27 H3K36M-OE. **c** Violin plots showing a shift of compartment scores towards 0, indicating a decrease in compartmentalization in QuiKO mMSCs (*n* = 24832) and Cal27 H3K36M-OE (*n* = 27148) cells. Boxes represent the 66% quantile interval (from the 17th to 83rd percentiles), with the central point indicating the median. Whiskers extend to the 95% quantile interval (from the 2.5th to 97.5th percentiles). **d** Hi-C heatmaps illustrating a reduction of nuclear compartmentalization in H3K36M-OE Cal27 cells. **e** Barplots illustrating a shift of A to B compartments in QuiKO mMSCs and Cal27 H3K36M-OE cells. **f-g** Representative immunofluorescence images, depicting a reduction of H3K9me3 foci, especially those at the periphery, in mMSC QuiKO and Cal27 H3K36M-OE cells. Nuclei were stained with anti-H3K9me3 antibody and DAPI respectively. **h** Boxplots

showing reduced foci per nuclei in QuiKO mMSCs and Cal27 H3K36M-OE cells. **i** Boxplots of the average distance of foci to the center of each nucleus, showing that the foci in respective parental cells are further away from the center than in the QuiKO mMSCs or Cal27 H3K36M-OE cells. For **a** and **d**, the blue (x-axis) and maroon (y-axis) tracks correspond to MS-normalized H3K9me3 for PA and QuiKO/H3K36M-OE cells, respectively. The 1st eigenvalue is shown, with positive and negative eigenvalues values corresponding to active A (green) and inactive B compartments (red), respectively. In **h** and **i**, *n* = 130 nuclei per condition were sampled for mMSC whereas *n* = 103 nuclei for Cal27. *** represents *p*-value < 0.001 and **** represents *p*-value < 0.0001 from a two-sided Wilcoxon rank-sum test. In **h**, the *p*-value for mMSC is 1.7e-24; in Cal27 *p* = 1.9e-04. In **i**, the *p*-value for mMSC is 3.9e-09; in Cal27 *p* = 5.3e-08. Boxes span the first quartile and third quartile, median is indicated with a center line, and whiskers extend to 1.5 times the interquartile range. Source data are provided as a Source Data file.

and chromatin loops. Interestingly, the number and strength of TAD boundaries remain largely unaffected across all conditions and cell types (Supplementary Fig. 8g, h). However, both QuiKO and H3K36M-OE cells exhibit a greater than twofold reduction in the number of chromatin loops compared to their respective parental controls (1448 versus 701 in mMSC; 2379 versus 1037 in Cal27; 497 versus 367 in Detroit562) (Supplementary Fig. 8i). Notably, this reduction is more pronounced for long-range loops (>500 kb) than for short-range loops across all cell lines (Supplementary Fig. 8j). To assess whether disruptions to chromatin loops impact enhancer activity and target gene expression, we identified annotated enhancers located at the anchors of the affected loops, determined their corresponding target genes, and quantified changes in gene expression. Notably, we found that genes associated with these disrupted loops were preferentially downregulated (Supplementary Fig. 8k). Overall, these data demonstrate that H3K36me depletion leads to profound alterations in 3D genome architecture, characterized by a loss of long-range compartmental interactions, attenuation of heterochromatin compaction, and a marked reduction in chromatin loops—particularly those spanning large genomic distances. The preferential downregulation of genes linked to disrupted loops suggests that impaired chromatin topology may be associated with transcriptional dysregulation.

To visualize these structural alterations at the microscopic level, we performed immunofluorescence (IF) to examine the subnuclear distribution of H3K9me3 (Supplementary Fig. 8l). In parental mMSCs, H3K9me3 localizes to numerous discrete nuclear foci, likely corresponding to large heterochromatic domains (Fig. 8f; Supplementary Fig. 8l). Many of these foci are enriched at the nuclear periphery (Fig. 8f), consistent with interactions between heterochromatin and the nuclear lamina, a hallmark of spatial genome organization[65,66]. Similarly, Cal27 parental cells predominantly exhibit peripherally localized H3K9me3 foci (Fig. 8g). In contrast, QuiKO mMSCs show a markedly diffuse distribution of H3K9me3 with fewer defined foci and significantly reduced peripheral localization (Fig. 8f, h, i; Supplementary Fig. 8l). Instead, residual H3K9me3 signal is redistributed toward the nuclear interior. Notably, while H3K9me3 continues to co-localize with DAPI, which preferentially stains AT-rich, heterochromatic DNA, the DAPI signal itself also shifts towards the center of the nucleus in QuiKO cells (Fig. 8f), further supporting a large-scale reorganization of nuclear architecture. Similar alterations are observed in Cal27 H3K36M-OE cells (Fig. 8g–i; Supplementary Fig. 8m), indicating that this effect is conserved across species and cellular contexts.

Collectively, these results demonstrate that the loss of H3K36 methylation not only perturbs euchromatin but also fundamentally disrupts the spatial organization and compartmentalization of heterochromatin. The displacement of H3K9me3 from the nuclear periphery toward the interior may reflect a collapse in nuclear lamina-heterochromatin interactions, possibly contributing to the loss of long-range chromatin contacts observed by Hi-C. Given the central role of 3D genome organization in maintaining gene silencing and

genome stability, our results suggest that H3K36 methylation is an upstream regulator of both chromatin topology and nuclear compartmentalization.

## Discussion

H3K36 methylation is emerging as a key factor shaping the epigenome, yet the full extent of its mechanistic roles in chromatin regulation remain unclear[3,14,38,67–70]. In this study we leveraged a comprehensive panel of mMSC lines harboring combinatorial deletions of all five K36MTs—NSD1, NSD2, NSD3, ASH1L, and SETD2—alongside complementary analyses in human HNSCC lines expressing the H3K36M oncohistone, to dissect the downstream consequences of H3K36me loss at multiple levels of chromatin function. Our findings reveal that H3K36me acts as a critical barrier to the encroachment of repressive chromatin states and is essential for preserving both local enhancer activity and higher-order chromatin architecture.

One of the key findings of this study is the enhancer-specific function of H3K36me2. Notably, depletion of H3K36me2 has a more pronounced effect on enhancer activity than on promoters, as evidenced by a greater loss of enhancer-associated histone marks and chromatin accessibility. This is accompanied by increased deposition of the repressive H3K27me3 mark at enhancers, supporting the role of H3K36me2 as a protective barrier against Polycomb-mediated silencing[33,34]. Consistent with this, we find that most genes dependent on H3K36me2-marked enhancers are downregulated upon loss of H3K36me2, suggesting a selective dependency of distal regulatory elements on this modification. These transcriptional changes likely reflect both direct consequences of enhancer inactivation and broader downstream effects from global loss of H3K36me2 and/or ablation of the respective K36MTs. The latter has been partially described in prior studies of the non-catalytic functions of K36MTs and is further supported by the existence of catalytically inactive isoforms, such as NSD2-short and NSD3-short[25,68,71]. Moreover, in NSD3- and ASH1L-KO cells, H3K36me2 levels remain largely unchanged at the H3K36me2-dependent target genes identified in TKO and QKO cells, suggesting functional compensation by NSD1 and/or NSD2, which may exhibit less stringent requirements for the localization of their catalytic activity. Nevertheless, the genes regulated by H3K36me2-marked enhancers show clear transcriptional downregulation that corresponds to diminished enhancer activity, demonstrated by the loss of activating histone marks and increased H3K27me3 deposition. These findings are consistent with previous studies in mouse embryonic stem cells and HNSCC cells, where the loss of NSD1 and H3K36me2 have been associated with impaired enhancer function[3,18,25]. Altogether, these results demonstrate that H3K36me2 at CREs primarily influences transcription through its deposition at enhancers.

In line with prior biochemical evidence that H3K36me2 and H3K36me3 inhibit PRC2 activity, we observe that their progressive depletion in mMSCs permits broadening of H3K27me2 and H3K27me3 across gene bodies—regardless of transcriptional status[33,34]. While

H3K36me3 loss alone (SETD2-KO) has limited effect on H3K27me3 invasion, loss of H3K36me2 (NSD1/2-DKO) or both marks (TKO) permits widespread genic occupancy by PRC2 products. However, even in the complete absence of H3K36me in QuiKO cells, H3K27me2/3 does not fully invade active gene bodies. We propose that the remaining barriers are the result of histone turnover during transcription—which dilutes H3K27me-marked nucleosomes—and potentially DNAme, which is known to inhibit PRC2 binding[44]. Supporting this, the complete loss of H3K36me in mMSCs does not lead to a significant reduction in gene body DNAme, suggesting that DNMT1-mediated maintenance methylation is sufficient to preserve DNAme—even in the absence of recruitment of de novo DNMTs (Fig. 4a).

Perhaps most strikingly, we uncover a novel and unanticipated relationship between H3K36me and H3K9me3. In cells lacking H3K36me2/3, large constitutive heterochromatin domains marked by H3K9me3 (cluster B) lose their defining features, including HP1 occupancy and compartmental insulation. While some of these regions undergo chromatin opening, gain H3K27ac and H3K4me1, and become transcriptionally active, other cluster B regions gain H3K27me3, suggesting that loss of H3K9me3 permits PRC2-mediated reprogramming at previously silenced domains. Concurrently, SUV39H1 redistributes to previously euchromatic (cluster A) domains and deposits ectopic H3K9me3, accompanied by redistribution of HP1 and local gene silencing. These findings, visualized in our working model (Fig. 9), suggest that H3K36me not only prevents PRC2-mediated silencing, but also blocks inappropriate activity of constitutive heterochromatin machinery in euchromatin. This redistribution of H3K9me3 and SUV39H1 coincides with dramatic changes in nuclear architecture—including compartment decompaction, weakened long-range chromatin interactions, and altered nuclear H3K9me3 foci—highlighting the pivotal role of H3K36me in genome organization.

Our work also provides insight into the epigenomic consequences of the cancer-associated H3K36M mutation, which reduces both H3K36me2 and H3K36me3. In Cal27 and Detroit562 HNSCC cells, H3K36M overexpression phenocopied the major findings in mMSCs, underscoring the generalizability and potential clinical relevance of these chromatin changes. These cross-species validations strengthen the broader implication that H3K36me loss is not only epigenetically disruptive, but may also underlie oncogenic chromatin remodeling.

While our study reveals extensive consequences of H3K36me loss, several questions remain. First, although we demonstrate SUV39H1 redistribution in H3K36me-deficient cells, the molecular mechanism guiding its ectopic recruitment to euchromatin remains undefined. Whether SUV39H1 responds to a loss of antagonistic chromatin modifications, changes in chromatin accessibility, or altered binding partners will require future investigation. Second, while we attribute enhancer deactivation and gene silencing to the loss of H3K36me2, it remains possible that non-catalytic roles of individual K36MTs—such as NSD3 or ASH1L—also contribute to the observed effects. Deconvoluting catalytic from scaffolding functions will require domain-specific rescue approaches. Third, it will be particularly important to systematically deconvolve the context-specific roles of each H3K36 methyltransferase in shaping distinct layers of chromatin regulation. Although our multiple-KO approach allowed us to progressively strip away H3K36me marks and observe additive epigenomic effects, this design does not fully capture the specialized genomic targeting or context-specific activity of individual enzymes. For example, NSD1 and NSD2 deposit broad intergenic H3K36me2 domains, whereas NSD3 targets focal CRE regions, and ASH1L appears to mark select developmentally relevant loci[26]. Future work employing selective domain and/or catalytically dead mutants, targeted degradation systems, or locus-specific tethering approaches will be instrumental in delineating which enzymes mediate which aspects of enhancer stability, repressive mark exclusion, and 3D genome maintenance. Moreover, exploring how these enzymes function in distinct developmental lineages, stress responses, or disease states will be essential to understanding their broader physiological roles and therapeutic potential.

In summary, our study establishes H3K36 methylation as a central regulator of enhancer activity, transcriptional insulation, and 3D genome integrity. By revealing its broad and multi-layered protective roles against epigenetic reprogramming, our findings provide a conceptual framework for understanding how the loss of a single chromatin mark can reverberate across the genome to destabilize the entire epigenetic landscape.

# Methods

## Cell culture

C3H10T1/2 mouse mesenchymal stem cells (ATCC) were cultured in DMEM (1X) (Invitrogen) with 10% fetal bovine serum (FBS) (Wisent) and supplemented with 1% GlutaMax and penicillin-streptomycin. C3H10T1/2 cell lines harboring individual knockouts of NSD1, NSD2, NSD3, SETD2, and ASH1L, as well as the NSD1/2/3-SETD2-QKO and NSD1/2/3-SETD2-ASH1L-QuiKO cell lines were previously established by our group using CRISPR-Cas9 RNP-mediated gene editing in Shipman and Padilla et al.[26]. C3H10T1/2 cells overexpressing the H3K36M mutation were provided by the lab of Dr. Chao Lu (Stanford University), and C3H10T1/2 NSD1/2-DKO and C3H10T1/2 NSD1/2-SETD2-TKO cells were provided by the lab of Dr. C. David Allis (Rockefeller University). Cal27 (ATCC, CRL-2095) and Detroit562 (ATCC, CCL-138) HNSCC cells were cultured in DMEM (F12) (Invitrogen) with 10% FBS and penicillin-streptomycin. HEK293T cells were cultured in DMEM (Invitrogen) supplemented with 10% FBS, 1% GlutaMax and penicillin-streptomycin. *Drosophila* S2 cells were cultured in Schneider's *Drosophila* medium (ThermoFisher) containing 10% FBS. All cell lines tested negative for mycoplasma contamination.

## Generation of Cal27 and Detroit562 HNSCC cells overexpressing the H3K36M mutation

Epitope-tagged H3.3 was previously cloned into pCDH-EF1-MCS-Puro lentiviral vector and site-directed mutagenesis techniques were used to generate the H3K36M mutation[14]. The H3.3K36M pCDH-EF1-MCS-Puro construct was provided by Dr. Chao Lu (Stanford University). To produce lentivirus, 293T cells were transfected with the lentiviral vector containing H3.3K36M and helper plasmids (psPAX2 and pMD2.G). Supernatant containing lentivirus was collected after 48 h, filtered and used to infect Cal27 and Detroit562 cells. Transduced cells were grown under puromycin selection (1 μg/ml) for 48 h after transduction.

## Generation of stable C3H10T1/2 SUV39H1 knockdown cell lines

To establish stable knockdown of SUV39H1, a collection of five different mouse SUV39H1 shRNA plasmids were obtained from the McGill University Life Science Complex High-Throughput Screening Facility (Sigma-Aldrich Mission shRNAs: TRCN0000097439, TRCN0000097 440, TRCN0000097441, TRCN0000097442 and TRCN0000097443) (Supplementary Table 1). Lentiviral particles were produced by transiently transfecting individual lentiviral SUV39H1 or control shRNA vectors together with the helper plasmids psPAX2 and pMD2.G at a ratio of 5:4:1 respectively in HEK293T cells. Briefly, HEK293T cells were grown to 70% confluence in 60 mm cell culture dishes prior to transfection. Cells were transfected with the TransIT-LT1 transfection reagent following the manufacturer's protocol. The lentivirus-containing supernatants were collected and filtered 72 h later using a 0.45 μm syringe filter and stored at −80 °C. To maximize the knockdown of SUV39H1 expression, C3H10T1/2 parental and NSD1/2-SETD2 TKO cells were transduced simultaneously with all five SUV39H1 shRNA lentiviruses. Cells were grown to 70% confluence in a well of a 6 well plate prior to infection. Cells were infected in DMEM complete media supplemented with 5 mg/ml of polybrene and infected using 125 μl of pooled lentiviral particles. Media was refreshed

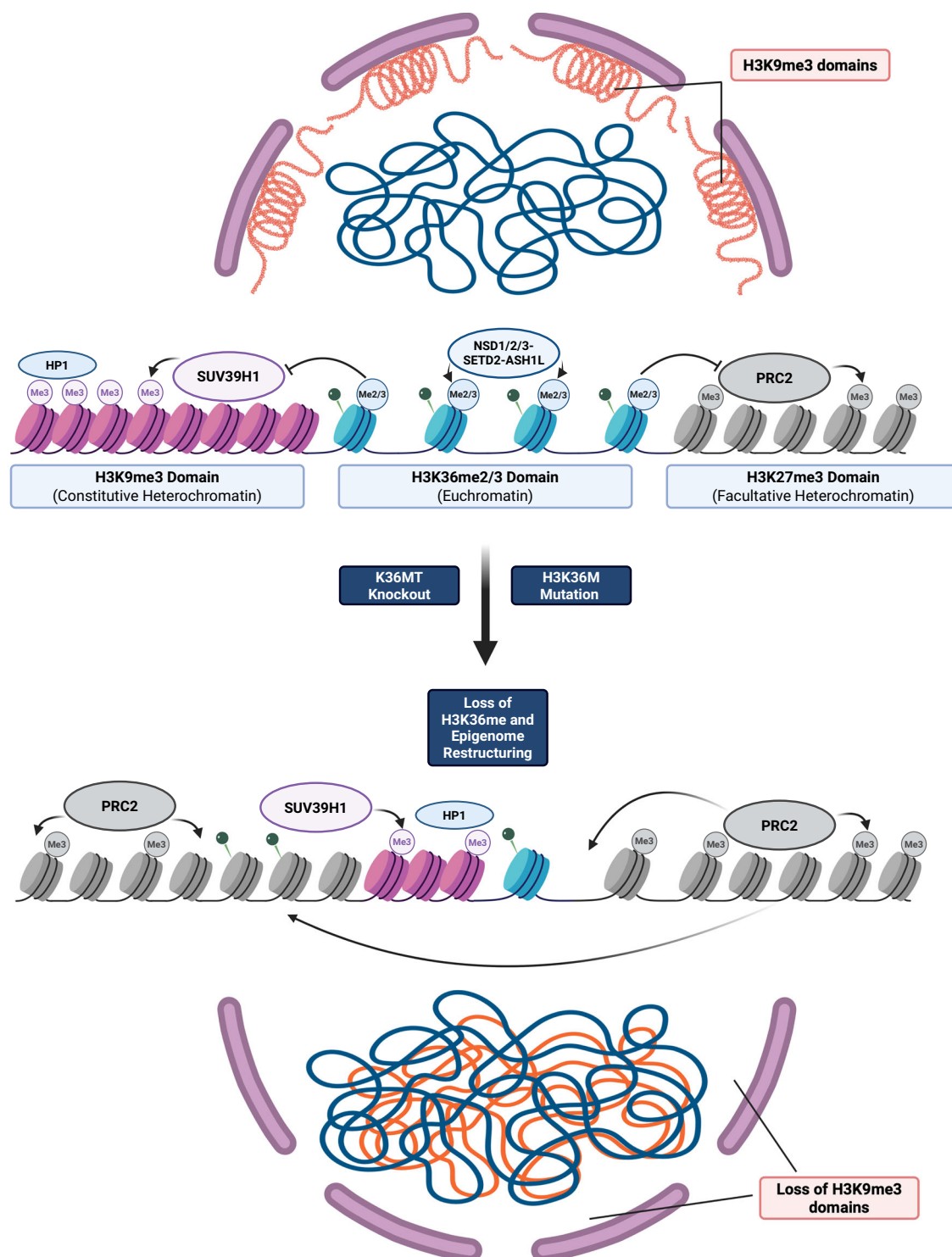

**Fig. 9 | Schematic model for H3K36me-mediated epigenome integrity.** H3K36 methylation prevents the invasion of PRC2-mediated H3K27me3 and SUV39H1-mediated H3K9me3 into euchromatin and is essential for maintaining the segregation of euchromatin and heterochromatin as well as overall genome compartmentalization. Created in BioRender. Shipman, G. (2025) https://BioRender.com/0iz7mrr.

24 hours post-infection, and the cells were allowed to grow for an additional 48 hours prior to selection. The infected cells were selected with puromycin (1 μg/ml) for 7 days, and all surviving cells were pooled to generate stable cell lines with SUV39H1 knockdown. SUV39H1 expression was evaluated by Western blotting using an anti-SUV39H1 antibody (Supplementary Table 2).

## Generation of stable C3H10T1/2 cell lines overexpressing SUV39H1

To establish stable overexpression of human SUV39H1, the pCDH-blast-HA-3XFlag-SUV39H1-WT plasmid was provided by the laboratory of Dr. Scott Rothbart[52]. Lentivirus production, infection and establishment of C3H10T1/2 parental and NSD1/2-SETD2-TKO cells

overexpressing SUV39H1 was performed as described for the SUV39H1 knockdown. Infected cells were selected using blasticidin (4 μg/ml) for 10 days, and all surviving cells were pooled to generate stable cell lines overexpressing FLAG-tagged SUV39H1. Overexpression of FLAG-tagged SUV39H1 was confirmed by Western blotting using an anti-FLAG antibody (Supplementary Table 2).

## Western blots

For each sample, 1 million cells were collected and counted using an automated Countess counter. Cell pellets were washed using PBS and resuspended in 100 ul of 1X RIPA buffer, 0.1 mM proteinase inhibitors cocktail, 0.1 mM PMSF, and incubated on ice for 1 hour. Samples were centrifuged for 10 minutes at 21,000 x g. Supernatants were transferred into new tubes. Protein quantification was performed using the BCA-Pierce Protein assay ThermoScientific/Pierce (Cat # 23227) and the Infinite 200 Pro Tecan-iControl. For each sample, 40 μg of protein was mixed with 1X Laemmli buffer (0.35 M Tris HCl pH 6.8, 30% Glycerol, 10% SDS, 20% Beta-mercaptoethanol, 0.04% Bromophenol blue, $H_2O$) and loaded into stain-free TGX 4%–15% gradient pre-cast gels (Bio-Rad, Cat # 4568084), and All-Blue (Bio-Rad, Cat # 1610373) and Unstained (Bio-Rad, Cat # 1610363) protein standards were mixed in equal amounts and loaded. Semi-dry electrotransfer onto a PVDF membrane was completed using the Trans-blot Turbo Transfer system (Bio-Rad, Trans-blot® Turbo RTA Mini LF PVDF Transfer Kit, 1704274). PVDF membranes were blocked with 5% BSA in TBSt, and probed overnight with the respective primary antibody (Supplementary Table 2). Following washes with TBSt, membranes were incubated for 1 hour with horseradish peroxidase-conjugated secondary antibody (anti-rabbit or goat anti-mouse) in 2% BSA-TBSt. Imaging was completed using the ECL Clarity or Clarity Max solutions (Bio-Rad, Cat # 1705060 and 1705062).

## Crosslinking and Histone PTM ChIP-seq

Approximately 20 million cells per cell line were used. Crosslinking was performed in 150 mm cell culture plates using 1% formaldehyde (Sigma) at room temperature with gentle rocking for 10 minutes. The crosslinking reaction was quenched using 1.25 M Glycine with gentle rocking at room temperature for 5 minutes. Fixed cell preparations were washed with ice-cold PBS, scraped, and then washed twice more with ice-cold PBS. Crosslinked pellets were resuspended in 500 μl cell lysis buffer (5 mM pH 8.5 PIPES, 85 mM KCl, 1% (v/v) IGEPAL CA-630, 50 mM NaF, 1 mM PMSF, 1 mM phenylarsine oxide, 5 mM sodium orthovanadate, EDTA-free protease inhibitor tablet) and incubated for 30 minutes on ice. Samples were centrifuged at 12,000 x g for 10 minutes at 4 °C, and pellets were resuspended in 500 μl of nuclei lysis buffer (50 mM pH 8.0 Tris-HCl, 10 mM EDTA, 1% (w/v) SDS, 50 mM NaF, 1 mM PMSF, 1 mM phenylarsine oxide, 5 mM sodium orthovanadate, EDTA-free protease inhibitor tablet) and then incubated for 30 minutes on ice. Sonication of lysed nuclei was performed using the BioRuptor UCS-300 at maximum intensity for 75-90 cycles (10 s on, 20 s breaks). Sonication efficiency to achieve fragments between 150 - 500 bp was evaluated using gel electrophoresis on a reversed-crosslinked and purified aliquot from each sample. After sonication, chromatin was diluted to reduce SDS levels to 0.1% and concentrated using Nanosep 10 K OMEGA (Pall) columns. Prior to the chromatin immunoprecipitation (ChIP) reaction, 2% sonicated *Drosophila* S2 cell chromatin was spiked into each sample for quantification of total levels of histone marks. The ChIP reactions were performed using the Diagenode SX-8G IP-Star Compact and Diagenode automated iDeal ChIP-seq Kit for Histones. Dynabeads Protein A or G (Invitrogen) were washed, then incubated with specific antibodies (Supplementary Table 2), 1.5 million cells of sonicated cell lysate, and protease inhibitors for 10 hrs, followed by a 20 min wash cycle using the provided wash buffers (Diagenode Immunoprecipitation Buffers, iDeal ChIP-seq

Kit for Histones). Reverse cross-linking was then performed using 5 M NaCl at 65 °C for 4 h. ChIP samples were then treated with 2 μl RNase Cocktail at 65 °C for 30 min followed by 2 μl Proteinase K at 65 °C for 30 min. Samples were then purified with QIAGEN MinElute PCR purification kit (QIAGEN) as per the manufacturer's protocol. In parallel, input samples (chromatin from about 50,000 cells) were reverse crosslinked and DNA was isolated following the same protocol. Library preparation was performed using the Kapa Hyper Prep library preparation reagents following the manufacturer's protocol (Kapa Hyper Prep Kit, Roche 07962363001). ChIP libraries were sequenced by the McGill University Genome Centre using the Illumina HiSeq 4000 at 50 bp single reads or NovaSeq 6000 at 100 bp single reads.

## Crosslinking and protein ChIP-seq

Approximately 40 million cells per cell line were used. Crosslinking was performed in 150 mm cell culture plates using 1% formaldehyde (Sigma) at room temperature with gentle rocking for 10 minutes. The crosslinking reaction was quenched using 1.25 M Glycine with gentle rocking at room temperature for 5 minutes. Fixed cell preparations were washed with ice-cold PBS, scraped, and then washed twice more with ice-cold PBS. Crosslinked pellets were resuspended in 500 μl cell lysis buffer (5 mM pH 8.5 PIPES, 85 mM KCl, 1% (v/v) IGEPAL CA-630, 50 mM NaF, 1 mM PMSF, 1 mM phenylarsine oxide, 5 mM sodium orthovanadate, EDTA-free protease inhibitor tablet) and incubated for 30 minutes on ice. Samples were centrifuged at 12,000 x *g* for 10 min at 4 °C, and pellets were resuspended in 500 μl of nuclei lysis buffer (50 mM pH 8.0 Tris-HCl, 10 mM EDTA, 1% (w/v) SDS, 50 mM NaF, 1 mM PMSF, 1 mM phenylarsine oxide, 5 mM sodium orthovanadate, EDTA-free protease inhibitor tablet) and then incubated for 30 minutes on ice. Sonication of lysed nuclei was performed using the BioRuptor UCS-300 at maximum intensity for 75–90 cycles (10 s on, 20 s breaks). Sonication efficiency to achieve fragments between 150–500 bp was evaluated using gel electrophoresis on a reversed-crosslinked and purified aliquot from each sample. After sonication, chromatin was diluted to reduce SDS levels to 0.1% and concentrated using Nanosep 10 K OMEGA (Pall) columns. Prior to the chromatin immunoprecipitation (ChIP) reaction, 2% sonicated *Drosophila* S2 cell chromatin was spiked into each sample for quantification. For each IP, Dynabeads Protein A or G (Invitrogen) were washed with RIPA buffer and then resuspended in RIPA buffer. Specific antibodies were added (Supplementary Table 2) and then incubated at 4 °C for at least 3 hours on a rotator. Following antibody conjugation, Dynabeads were washed with RIPA buffer, and then resuspended in RIPA buffer, ~5 million cells of sonicated cell lysate, and protease inhibitors, followed by overnight incubation on a rotator at 4 °C. Four successive washes were then performed using **1.** RIPA buffer; **2.** RIPA buffer with 500 mM NaCl (1% (v/v) Triton X-100, 0.10% SDS, 500 mM NaCl, 20 mM Tris-HCl pH 8.0, 2 mM EDTA, $H_2O$); **3.** LiCl buffer (1% NP-40, 250 mM LiCl, 10 mM Tris-HCl pH 8.0, 1 mM EDTA, $H_2O$); **4.** TE buffer (50 mM NaCl, 10 mM Tris-HCl pH 8.0, 1 mM EDTA, $H_2O$). Samples were then resuspended in Elution buffer (50 mM Tris-HCl pH 8.0, 10 mM EDTA, 1% (v/v) SDS, $H_2O$) and incubated at room temperature for 30 minutes. Reverse cross-linking was then performed using 5 M NaCl at 65 °C for 4 h. ChIP samples were then treated with 2 μl RNase Cocktail at 65 °C for 30 min followed by 2 μl Proteinase K at 65 °C for 30 min. Samples were then purified with QIAGEN MinElute PCR purification kit (QIAGEN) as per the manufacturer's protocol. In parallel, input samples (chromatin from about 50,000 cells) were reverse crosslinked and DNA was isolated following the same protocol. Library preparation was performed using the Kapa Hyper Prep library preparation reagents following the manufacturer's protocol (Kapa Hyper Prep Kit, Roche 07962363001). ChIP libraries were sequenced by the McGill University Genome Centre using the Illumina HiSeq 4000 at 50 bp single reads or NovaSeq 6000 at 100 bp single reads.

## CUT&RUN

CUT&RUN reactions were performed with the Epicypher CUTANA ChIC/CUT&RUN Kit Version 3 (cat# 14-1048) using 550,000 fresh C3H10T1/2 cultured cells and following the manufacturer's protocol (Epicypher User Manual Version 3.5) for adherent cells. Prior to performing CUT&RUN, optimization of C3H10T1/2 cell permeabilization was performed following Appendix 1.1 (Epicypher User Manual Version 3.5). Cell permeabilization was found to be most optimal using a final working concentration of 0.01% Digitonin (Promega). Successful binding of cells to Concanavalin A beads was confirmed as recommended and outlined by the manufacturer. Manufacturer supplied H3K4me3 positive control and IgG negative control antibodies were used, and the reactions were performed in parallel with processed samples. Processed samples were incubated with specific antibodies (Supplementary Table 2). Manufacturer also supplied *E.coli* spike-in DNA, which was used in each reaction for sequencing normalization. Quantification of purified CUT&RUN DNA was performed using the Qubit fluorometer with the 1X dsDNA HS Assay Kit (Invitrogen). Library preparations were performed using the Epicypher CUTANA CUT&RUN Library Prep Kit Version 1 (cat# 14-1001) following the manufacturer's protocol. Final CUT&RUN library concentrations were quantified as previously described, and enrichment of mononucleosome-sized fragments was assessed by Bioanalyzer (Agilent) prior to sequencing. CUT&RUN libraries were sequenced at a depth of 15 M reads per sample by the McGill University Genome Centre using the Illumina NovaSeq 6000 at 100 bp paired-end reads.

## ATAC-Seq

ATAC-Seq library preparation was performed according to the Omni-ATAC protocol[72]. 50,000 C3H10T1/2 cultured cells were resuspended in 1 ml of cold ATAC-seq resuspension buffer (RSB; 10 mM Tris-HCl pH 7.4, 10 mM NaCl, and 3 mM MgCl2 in water). Cells were centrifuged at 500 x *g* for 5 min in a pre-chilled (4 °C) fixed-angle centrifuge. After centrifugation, supernatant was aspirated and cell pellets were then resuspended in 50 µl of ATAC-seq RSB containing 0.1% IGEPAL, 0.1% Tween-20, and 0.01% digitonin by pipetting up and down three times. This cell lysis reaction was incubated on ice for 3 min. After lysis, 1 ml of ATAC-seq RSB containing 0.1% Tween-20 (without IGEPAL and digitonin) was added, and the tubes were inverted to mix. Nuclei were then centrifuged for 10 min at 500 x *g* in a pre-chilled (4 °C) fixed-angle centrifuge. Supernatant was removed and nuclei were resuspended in 50 uL transposition mix (2x TD Buffer, 100 nM final transposase, 16.5 uL PBS, 0.5 uL 1% digitonin, 0.5 uL 10% Tween-20, 5 uL H2O) (Illumina Tagment DNA Enzyme and Buffer Small Kit, 20034197). Transposition reactions were incubated at 37 °C for 30 min in a thermomixer with shaking at 1000 rpm. Reactions were cleaned up with DNA Clean and Concentrator 5 columns (Zymo Research). Illumina Nextera DNA Unique Dual Indexes Set C (Illumina, 20027215) were added and amplified (12 cycles) using NEBNext 2x MasterMix. Sequencing of the ATAC-Seq libraries was performed on the Illumina NovaSeq 6000 system using 100-bp paired-end sequencing.

## RNA-Seq

Total RNA was extracted from approximately 1 million cells using the AllPrep DNA/RNA/miRNA Universal Kit (QIAGEN) and following the manufacturer's protocol, including DNase treatment option. Library preparation was performed using the NEBNext® rRNA Depletion Kit v2 (Human/Mouse/Rat) (New England Biolabs) following the manufacturer's protocol (including RNase H and DNase I digestions). 100-bp paired-end sequencing was performed on the Illumina HiSeq 4000 or NovaSeq 6000 platform.

## Histone acid extraction, histone derivatization, and analysis of post-translational modifications by nano-LC-MS

4 million cells from each cell line were collected and frozen at -80°C. Thawed pellets were lysed in nuclear isolation buffer (15 mM Tris pH 7.5, 60 mM KCl, 15 mM NaCl, 5 mM MgCl2, 1 mM CaCl2, 250 mM sucrose, 10 mM sodium butyrate, 0.1% (v/v) beta-mercaptoethanol, commercial phosphatase and protease inhibitor cocktail tablets) containing 0.3% NP-40 alternative on ice for 5 min. Nuclei were subsequently washed twice in the same buffer without NP-40, and pellets were resuspended using gentle vortexing in chilled 0.4 N H2SO4, followed by a 3 h incubation while rotating at 4°C. After centrifugation, supernatants were collected and proteins were precipitated in 20% TCA overnight at 4°C, washed with 0.1% HCl (v/v) acetone once, followed by two washes with acetone alone. Histones were resuspended in deionized water. Acid-extracted histones (20 µg) were resuspended in 50 mM ammonium bicarbonate (pH 8.0), derivatized using propionic anhydride and digested with trypsin as previously described (15). After the second round of propionylation, the resulting histone peptides were desalted using C18 Stage Tips, dried using a centrifugal evaporator and reconstituted using 0.1% formic acid in preparation for LC-MS analysis. Nanoflow liquid chromatography was performed using a Thermo Fisher Scientific Vanquish Neo UHPLC equipped with an Easy-Spray™ PepMap™ Neo nano-column (2 µm, C18, 75 µm X 150 mm). Buffer A was 0.1% formic acid and Buffer B was 0.1% formic acid in 80% acetonitrile. Peptides were resolved using at room temperature with a mobile phase consisting of a linear gradient from 1 to 45% solvent B (0.1% formic acid in 100% acetonitrile) in solvent A (0.1% formic acid in water) over 85 mins and then 45 to 98% solvent B over 5 min at a flow rate of 300 nL/min. The HPLC was coupled online to an Orbitrap Exploris 240 (Thermo Scientific) mass spectrometer operating in the positive mode using a Nanospray Flex Ion Source (Thermo Fisher Scientific) at 1.9 kV. A full MS scan was acquired in the Orbitrap mass analyzer across 350–1050 m/z at a resolution of 120,000 in positive profile mode with an auto maximum injection time and an AGC target of 300%. Parallel reaction monitoring experiments were followed for monitoring the targeted peptides based on the inclusion list. Targeted ions were fragmented using HCD fragmentation. These scans typically used an NCE of 30, an AGC target standard, and an auto maximum injection time. Raw files were analyzed using EpiProfile 2.0[73] and Skyline[74]. The area for each modification state of a peptide was normalized against the total signal for that peptide to give the relative abundance of the histone modification. Calculated MS ratios (Supplementary Table 3).

## Whole genome bisulfite sequencing

Whole genome sequencing libraries were generated from 1000 ng of genomic DNA extracted from approximately 4 million cells and fragmented to 300–400 bp using the Covaris focused-ultrasonicator E210. The NxSeq AmpFREE Low DNA Library Kit (Lucigen - CA10837-144) was then used. End repair, A-tailing, adapter ligation, and clean-up steps were performed as per the manufacturer's recommendations. Bisulfite conversion was performed using the EZ-96 DNA Methylation-Gold™ MagPrep Kit (Zymo Research) according to the manufacturer's protocol. Bisulfite-converted DNA was amplified by PCR using the KAPA Hifi HotStart Uracil+ Kit (Roche) according to the manufacturer's protocol. The amplified libraries were then purified using AMPure XP Beads (Beckman Coulter). 150-bp paired-end sequencing of the WGBS libraries was performed using the Illumina HiSeqX system.

## Hi-C

For each sample, two million cells were crosslinked and processed using the Arima-HiC+ kit and following the Arima Genomics HiC protocol for Mammalian Cell Lines (A510008). Proximally-ligated DNA was fragmented to an average size of 400 bp using the BioRuptor UCS-300 at low intensity for 10 cycles (30 s ON, 90 s OFF). Library preparation of fragmented proximally-ligated DNA was performed using the NEBNext® Ultra™ II DNA Library Prep Kit (E7645S), NEBNext Multiplex Oligos for Illumina (E6440S), and the Arima HiC+ Kit according to manufacturer protocols. Hi-C libraries were sequenced by the McGill

University Genome Centre at 150 bp paired-end reads using the Illumina NovaSeq 6000.

## Immunofluorescent staining and analysis

Cells were seeded on glass coverslips until between 60-80% confluent and fixed with a cold solution of methanol:acetone at 80:20 ratio for 15 minutes. The samples were blocked 1 h with 5% skim milk followed by 1 h with rabbit anti-H3K9me3 antibody (Supplementary Table 2) at 1:400 dilution and 1 h with Alexa Fluor-488 goat anti-rabbit secondary antibody, all in blocking solution. Samples were then counterstained with 4′,6-diamidino-2-phenylindole (DAPI) solution in PBS for 5 min. All the different treatments were separated by 3 washes with PBS. The coverslips were inverted onto permafluor mountant (epredia) on microscope slides, viewed and imaged on confocal LSM800 at the McGill University Advanced BioImaging Facility (ABIF). Z-stacks of 10 slices were taken, and an image projection (average setting) was built using Fiji[75] v.2.15.1. All images were taken under the same conditions, but laser strength and exposure were slightly adjusted for each image so that there is no saturation and to ensure similar fluorescence intensity in each image. This allowed a proper comparison of the spatial distribution of the foci. Images were filtered using ImageJ[76] v.1.54 for any nuclei that were dividing and any nuclei that were cutoff at the border of the images. Analysis was performed using the open-source software CellProfiler[77] v.4.2.6. A CellProfiler pipeline was designed to detect and count nuclei as well as the number of H3K9me3 foci within each nucleus using the same parameters for all images. For detecting nuclei, a manual thresholding method was applied with a threshold of 0.015 and only nuclei with a diameter from 50 to 350 were included. 130 nuclei per condition was included for analysis in mMSCs whereas 103 nuclei per condition was included in Cal27. For detecting the number of H3K9me3 foci within each nucleus, a manual thresholding method was applied with a threshold of 0.1, a smoothing filter of 11 and a minimum allowed distance of 6 (which distinguishes between clumped foci). The number of foci per nucleus in the H3K36me-deficient cells was normalized by multiplying the foci count per nucleus by the ratio of the mean surface area of the parental nuclei to the mean surface area of the H3K36me-deficient nuclei. Specifically, the foci count per nucleus in the QuiKO cells were normalized by the ratio of the mean nuclei surface area of the parental cells (13747.99 pixels) to that of the QuiKO cells (14394.36 pixels). Similarly, the foci count per nucleus in the H3K36M-OE cells were normalized by the ratio of the mean nuclei surface of the parental cells (6588.77 pixels) to that of the H3K36M-OE cells (6109.94 pixels). Lastly, the average distance of foci to the center of each nucleus was normalized by dividing by the radius.

## Visualization

Unless otherwise stated, figures were created using ggplot2[78] v3.3.0. Coverage/alignment tracks were visualized using pyGenomeTracks[79] v3.2.1 or Trackplot[80] v.1.5.10. Heatmaps and aggregate (average) profiles of bigwig pileups at enhancers, strong enhancers, promoters, gene bodies, clusters A and B regions, and other ATAC-seq peaks sets were generated using the computeMatrix, plotProfile and plotHeatmap functions from deepTools[81] v3.3.1. For all these heatmaps and aggregate profiles, replicates ($n = 2$ or $n = 3$) were merged prior to plotting, when applicable.

## ChIP-seq, CUT&RUN, ATAC-seq and RNA-seq processing and analysis

ChIP-seq reads were aligned using BWA[82] v.0.7.17 to a combined reference of mm10 and dm6 and afterwards filtered using a cut-off of MAPQ < 3 using Samtools[83] v.1.22.0. For HNSCC samples, ChIP-seq reads were aligned using a combined reference of hg38 and dm6. Samclip (https://github.com/tseemann/samclip) v.0.2 (samtools view -h in.bam | samclip --ref ref.fa | samtools sort > out.bam) was used to filter bacterial sequences prior to alignment for all ChIP-seq reads.

CUT&RUN reads were trimmed of adapter sequences using Fastp[84] v.0.23.3. Bowtie[85] v.2.5.1 was used to align the reads to the mm10 assembly using the parameters "-I 10 -X 700 --end-to-end --local --very-sensitive-local --no-unal --no-overlap --no-dovetail --no-mixed --no-discordant -1 ${SAMPLE}.R1.fq.gz -2 ${SAMPLE}.R2.fq.gz". Samtools was used to remove duplicated reads and reads below a cutoff of MAPQ < 3. Samclip was used to filter bacterial sequences prior to alignment.

The bamCoverage function from deepTools was used to normalize ChIP and CUT&RUN signals by dividing by the total alignments (in millions) (bamCoverage -b $BAM -o $OUTPUT.bigWig --normalizeUsing CPM --centerReads -bs 10 -e 200 -bl $mm10.BLACKLIST.bed). To generate coverage tracks, following CPM (counts per million) normalization as indicated above, replicates ($n = 3$) were merged in a stepwise fashion using bigwigCompare from deepTools with parameters '-b1 rep1 -b2 rep2 $outdir --operation mean -bs 10 -o $merged.step1.cpm.bw' and '-b1 merged.step1.cpm.bw -b2 rep3 $outdir --operation mean -bs 10 -o $merged.final.cpm.bw'. A normalization factor was computed by multiplying the genome-wide modification percentage values obtained from mass spectrometry (these values were averaged per condition) by the total number of bins and dividing by the total signal (in CPM) for a given bigWig file, which consists of merged replicates. This normalization factor was then multiplied to the depth-normalized signal (in CPM) for each merged bigWig track to generate MS-normalized coverage tracks. Finally, ChIP-seq/CUT&RUN bins greater than 100 were set to 100. The ENCODE blacklist[86] was used for filtering.

For ATAC-seq, raw reads were trimmed and filtered for quality using Trimmomatic[87] v.0.39. Trimmed reads were mapped to the mm10 genome assembly using Bowtie2, and non-uniquely mapping reads were removed. Afterwards, the reads were adjusted by shifting all positive-strand reads 4 bp downstream and all negative-strand reads 5 bp upstream to the center of the reads on the transposase binding event. Peak calling was performed on each replicate using MACS2[88] v.2.2.6 with '-extsize 200 -shift -100 -nomodel' parameters. To find a set of reproducible peaks across replicates, we calculated the irreproducible discovery rate (IDR)[89] v.2.0.3 and excluded peaks with an IDR greater than 0.05 across every pair of replicates. Subsequently, the ENCODE blacklist was used to filter the peaks. Coverage tracks were generated using bamCoverage with parameters '-b $BAM -o $OUTPUT.bigWig --normalizeUsing CPM --centerReads -e 200 --minMappingQuality 5". Replicates ($n = 2$) were merged using bigwigCompare from deepTools for aggregate plots.

To generate accessible enhancers for Fig. 1, an intersection of ATAC-seq peaks between TKO and QKO were first generated. From this set of ATAC-seq peaks, peaks within +/- 3 kb of TSSs were excluded to account for promoter biases - these filtered ATAC-seq peaks were then designated as accessible enhancers. Accessible promoters were then defined as ATAC-seq peaks overlapping protein-coding promoters, which were defined as 2-kb regions centered at the TSS (1500 bp upstream and 500 bp downstream). Strong enhancers were called from TKO ChIP-seq H3K27ac using ROSE[40] v1.0.0 with "-s 12500 -t 2500", to ensure that H3K27ac peaks within 12.5 kb were stitched together and regions within 2.5 kb of TSSs were excluded. Nearest enhancers, as indicated in Fig. S1g, were derived using the closestBed function from BEDTools[90] v.2.31.1.

For RNA-seq, raw reads were aligned to mm10 genome build using STAR[91] v.2.5.3a. Afterwards, featureCounts[92] v.1.5.3 was used to count exonic reads from the GTF transcript annotation (GENCODE version from UCSC). DESeq2[93] v.1.26.0 was used to perform differential gene expression analyses. Adjusted log fold changes (LFC) were calculated using 'apeglm'[94] v.1.8.0. Significantly differentially expressed genes were selected based on absolute log2FC > 2 and adjusted $p$-value less

than 0.05. Coverage bigWig tracks for RNA-seq signal were computed using bamCoverage from deepTools with the parameters "bamCoverage -b $SAMPLE.bam -o $SAMPLE.RPKM.bw -bl $BLACKLIST.bed --normalizeUsing RPKM".

To find large H3K36me2 peaks marking clusters of enhancers in TKO, enhancers from Ensembl (www.ensembl.org) and FANTOM[95] were first pooled together for the mm10 assembly. Epic2[96] v.0.0.52 was used to call peaks on two replicates of ChIP-seq H3K36me2 for the TKO cell line and their intersection was taken using BEDTools - these merged peaks were subsequently used to generate a read count matrix of ChIP-seq TKO H3K36me2 triplicates ($n = 3$) using featureCounts. These peaks were then filtered for having greater than 100 average normalized read count across TKO replicates and overlapping at least one annotated enhancer. Afterwards, the peaks were extended by 50000 bp on each side and these regions were designated as large H3K36me2 enhancer peaks. Next, to generate two groups of genes of similar expression, but differ in whether or not they are located within these large H3K36me2 enhancer peaks, we filtered for protein-coding genes with at least 1 FPKM gene expression at baseline (in the parental sample). Afterwards, the genes were stratified into bins, with each bin grouping genes of similar expression (+/- 3 FPKM). We selected genes with 9-12 FPKM to generate the two groups of genes in Fig. 1f with equal group sizes ($n = 219$ per group): one group of genes within the large H3K36me2 peaks and a randomized, control set of genes outside of these large H3K36me2 peaks.

For assessing replicate concordance, genome-wide Spearman's correlation analysis was performed using the multiBigWigSummary function from DeepTools using 500 bp bins. The results were plotted using gplots' (https://github.com/cran/gplots) v.3.1.3 'heatmap.2' function as heatmaps.

For Fig. 2, genes marked with residual H3K36me2 were determined by taking genes overlapping QKO H3K36me2 peaks by at least 1 bp. These residual 119 QKO H3K36me2 peaks were generated as described in Shipman et al.[26]. To generate the correlation scatter plot between the size of H3K36me2 peak and gene expression fold changes, featureCounts was used to generate a count matrix from QKO ChIP-seq H3K36me2 replicates ($n = 3$), using the remaining 119 peaks in QKO as reference. Afterwards, the raw counts per replicate were normalized by dividing by their total alignments (in millions) to generate counts per million (CPM) and the average was taken amongst the replicates for each of the 119 peaks. Afterwards, BEDtools was used to find the overlapping gene for each of the 119 peaks, and subsequently the gene/peak pairs were plotted using R. To generate accessible enhancers for Fig. 2, an intersection of ATAC-seq peaks between QKO and QuiKO were first generated. From this set of ATAC-seq peaks, peaks within +/- 3 kb of TSSs were excluded to account for promoter biases - these filtered ATAC-seq peaks were then designated as accessible enhancers. Next, we filtered for accessible enhancers overlapping the 119 remaining H3K36me2 peaks found in QKO. Furthermore, ATAC-seq peaks that overlapped with promoters that were marked by residual QKO H3K36me2 peaks were designated as accessible promoters for Fig. 2. Accessible enhancers and promoters devoid of H3K36me2 were ATAC-seq peaks that did not overlap the 119 remaining H3K36me2 peaks in QKO.

To generate metagene plots, 21833 protein-coding genes (from Ensembl) were selected with gene length >= 25th percentile (6842 bp) and <= 75th percentile (48614 bp) to avoid over/under-scaling gene sizes when generating pileups for the aggregate plots. Furthermore, adjacent/neighboring genes within -2 kb TSS and +2 kb TES regions were removed to eliminate edge effects. From this set of filtered protein-coding genes, genes with zero normalized counts were designated as "silent genes" ($n = 2000$). Lastly, to generate the expressed gene groups, protein-coding genes below 0.1 FPKM in the parental cell line were removed and from the remaining genes, the bottom 2000 genes designated as lowly expressed genes whereas the

top 2000 genes were designated as highly expressed genes. These three gene groups ($n = 2000$ per group) were then used as a reference to generate metagene plots using computeMatrix from deepTools with the parameters "computeMatrix scale-regions -R $SILENT_GENES.bed $LOWLY_EXPRESSED.bed $HIGHLY_EXPRESSED.bed -S $SAMP.bw -bl $BLACKLIST.bed -b 2000 -a 2000 -m 20000 -bs 500 --missingDataAsZero --skipZeros" and plotted using plotProfile from deepTools. Genes with significant H3K27me3 or H3K9me3 in their gene bodies were selected (adjusted p-value < 0.05) from a pool of mm10 protein-coding genes, in which featureCounts was used on aligned ChIP-seq files with mm10 protein-coding genes as the annotation/reference, followed by DESeq2 to generate differentially-marked (either by H3K27me3 or H3K9me3, respectively) protein-coding genes.

"Peakiness" scores were computed as the average read-depth normalized coverage of the top 1% most covered 1-kb windows across the genome, excluding those overlapping blacklisted regions, as previously described[3].

To generate genome-wide binned analysis in Fig. 5, raw H3K9me3 ChIP-seq tag counts were firstly binned into 100 kb windows using BEDtools with intersectBed (-c) in combination with the makewindows command, as previously described[18]. Binned tag counts were library-sized normalized by dividing by the total number of mapped reads after filtering and subsequently input-normalized by taking the log2 ratio of ChIP signals by those of the input (i.e., without immunoprecipitation), with the addition of a pseudocount (1e-15) to avoid division by 0: log2(((H3K9me3 ChIP binned tag counts)/(H3K9me3 total ChIP mapped reads)+1e-15)/((input binned tag counts)/(total input mapped reads)+1e-15)). Additionally, quantitative normalization was performed by multiplying the signal (log2 ratio over input as described above) by the genome-wide modification percentage information obtained from mass spectrometry. Genic and intergenic regions were defined based on ENSEMBL annotations, with genic regions corresponding to the union of all intervals annotated as "gene" in ENSEMBL, and intergenic regions defined as the remaining complementary regions[18]. Similarly behaving bin clusters were identified using HDBSCAN[97] v.0.8.24 with the same parameters for all comparisons (minPts = 1000, eps = 1000). To better reflect the large megabase-sized domains of H3K9me3, 100-kb bins within 1-mb were merged for each bin cluster. Afterwards, regions in cluster B overlapping with regions in cluster A were excluded. ChIPbinner v.0.99.1[98] was used to generate scatterplots of genic/intergenic regions and density-based clusters.

plotEnrichment from DeepTools[81] was used to compute the percentage of read counts found in cluster A and B regions. Profileplyr[99] v.1.6.0 was used to extract and summarize signal values from DeepTools' computeMatrix output for visualization with ggplot2 and statistical comparison across cluster A and B regions.

Over-representation analysis (ORA) for KEGG and gene ontology (GO) pathways was performed using clusterProfiler v.4.17.0[100], with a p-value cutoff of 0.05 and adjusted using FDR. Partial correlation analysis was performed using the cor() function from R's stats package, with statistical significance set at α = 0.05 and p-values adjusted for false discovery rate (FDR).

Transposable elements (TEs) were obtained from RepeatMasker (www.repeatmasker.org). These TEs were used as the annotated reference file for featureCounts to count RNA-seq transcript expression and subsequently generate a count matrix for input into DESeq2 to compute differentially expressed TEs. Adjusted LFC were calculated using 'apeglm' and significantly differentially expressed TEs were selected based on absolute log2FC > 1 and adjusted p-value less than 0.05.

## WGBS processing and analysis

Raw WGBS reads were processed and analyzed as previously described[3], where they were mapped and methylation calling was

performed using Bismark[101] v.0.24.2 against the mouse assembly mm10, ENCODE blacklisted regions were removed and only CpGs covered by ≥5× in all samples were retained for the computation of DNA methylation levels.

## Hi-C processing and analysis

Raw Hi-C reads were trimmed of adapter sequences using Fastp and aligned to the mm10 assembly using BWA with default parameters. Afterwards, ligation junctions (Hi-C pairs) were identified in aligned paired-end sequences using Pairtools (https://pairtools.readthedocs.io/en/latest/) v.1.1.0. The restriction enzyme specified to Pairtools was the Arima kit, which uses the two restriction enzymes: DpnII and HinfI. Duplicated reads were marked using 'pairtools dedup' with the parameters: "--mark-dups --backend cython --output-stats $SAMPLE.stats -o $SAMPLE.dedup.pairs.gz". Then, duplicated reads were removed and only uniquely-mapped reads were selected using 'pairtools select' with the parameters: "--chrom-subset mm10.canon.chroms '(rfrag1!=rfrag2) and ((pair_type == "UU" or pair_type == "RU" or pair_type == "UR"))' -o $SAMPLE.pairs.gz ${samp}.dedup.pairs.gz'. Junctions were subsequently binned and counted using Cooler[102] v.0.10.0, in which pairs (.pairs.gz) files were converted to a cooler (.cool) file using "cooler cload pairix --assembly mm10 mm10.canon.sizes:1000 $SAMPLE.pairs.gz $SAMPLE.1000.cool". The cooler file underwent matrix balancing using "cooler balance $SAMPLE.1000.cool --blacklist $BLACK-LIST.bed". The matrix was then binned into different window sizes contained within a single multi-resolution cooler (.mcool) file using "cooler zoomify --out $SAMPLE.mcool --resolutions 1000,5000, 10000,25000,50000,100000,250000,500000,1000000,2500000 --balance --balance-args --max-iters 1000 $SAMPLE.1000.cool".

Cooltools[103] v.0.4.0 was used to compute compartment scores (eigenvalues): "cooltools eigs-cis --phasing-track mm10.100000.gc $SAMPLE.mcool::/resolutions/100000", with the binned profile of GC content (mm10.100000.gc) as the reference track for sign flipping. To assess interaction patterns between bins belonging to different quantiles of compartment scores, the compute-saddle function from cooltools with default parameters at 100-kb resolution was implemented, using eigenvectors and expected values from cooltools' eigs-cis (calling compartments) and compute-expected modules, respectively. To quantify the interaction strength between A-A and B-B compartments, we took the top 20% of B-B interactions and the top 20% of A-A interactions, normalized by the bottom 20% of A-B interactions, as previously described[104]. Long-range *cis* contacts were defined as interactions spanning greater than 10 megabytes (mb) and short-range *cis* contacts were defined as interactions 1-kb to 10-mb.

Insulation scores were computed for '.mcool' files at 10-kb resolution with a window size of 100 kb using the "cooltools insulation" module with default parameters. Only boundaries annotated as "is_boundary_100000 == "TRUE"" were included. Loop scores were computed using the "cooltools dots" module with default parameters at 25-kb resolution. Long-range loops were defined as those spanning greater than 500-kb, whereas short-range loops were defined as those spanning 500-kb or less.

To analyze the effect of loss of long-distance loops on enhancer function, we firstly filtered for 25-kb loops whose anchors span more than 500 kb in both the treated (mMSC TKO or Cal27/Detroit H3K36M-OE) and wildtype (PA) samples. Next, we filtered for long-distance loops found in the PA samples but not in the treated samples - these loops were considered those affected by the loss of H3K36me. Then, in this subset of loops, we identified annotated enhancers (from FANTOM and ENSEMBL) situated within the anchors of these loops. Finally, the log2 fold gene expression changes for protein-coding genes most proximal to these identified enhancers were compared between the PA and treated samples (TKO or H3K36M-OE for mMSC and Cal27/Detroit, respectively).

## Statistical analyses

The statistical details used in each analysis are provided in the figure legends wherever applicable. Two-tailed Student's *t*-tests were used to compare gene expression levels of specific genes, such as SUV39H1, where sample sizes were equal between groups. For all other statistical comparisons, where unequal sample sizes were involved, the Wilcoxon rank-sum test was used. Statistical significance was denoted as: *, $p < 0.05$; **, $p < 0.01$; ***, $p < 0.001$; ****, $p < 0.0001$. The BH (Benjamini-Hochberg) correction was used whenever post-hoc corrections for multiple comparisons were necessary. No statistical methods were used to pre-determine sample sizes. No data were excluded from the analyses. The experiments and outcome assessment were not performed in a blinded manner. Except for the mMSC NSD1/2-DKO and mMSC H3K36M-OE samples—where repeated measurements were performed on the same samples—all other measurements were obtained from distinct biological samples. Boxplots were generated using R. In the box plots, boxes span the lower (first quartile) and upper quartiles (third quartile), median is indicated with a center line and whiskers extend to a maximum of 1.5 times the interquartile range. For barplots, error bars display the standard deviation around the mean. Results for correlation analyses are reported for Spearman's rho (ρ) or Pearson's correlation coefficient (*R*) and the associated *p*-value where applicable.

## Reporting summary

Further information on research design is available in the Nature Portfolio Reporting Summary linked to this article.

## Data availability

Previously published and publicly available ChIP-seq data can be accessed via the National Center for Biotechnology Information Gene Expression Omnibus (NCBI-GEO): H3K9me3 (parental) under accession number GSE118785, H3K27me3 (parental, SETD2-KO, H3K36M, and NSD1-2-DKO samples) under accession number GSE160266, and H3K36me2 (parental, ASH1L-KO, NSD3-KO, NSD1-2-SETD2-TKO, NSD1-2-3-SETD2-QKO, and NSD1-2-3-SETD2-ASH1L-QuiKO samples) under accession number GSE243566. Newly generated ChIP-seq, ATAC-seq, CUT&RUN, WGBS, RNA-seq and Hi-C data can be accessed under accession number GSE274367. Large H3K36me2 enhancer peaks identified in the mMSC NSD1/2-SETD2-TKO samples are available in the GitHub repository: (https://github.com/padilr1/H3K36me_guardian_epigenome_integrity_Padilla.git), located in the 'data' folder. Source data are provided with this paper.

## Code availability

All code used to perform the analyses presented here can be accessed in the GitHub repository: (https://github.com/padilr1/H3K36me_guardian_epigenome_integrity_Padilla.git) and in Zenodo: (https://doi.org/10.5281/zenodo.17372542)[105], under the GNU General Public License V3.0. Source data are provided with this paper.

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

## Acknowledgements

We would like to thank all members of the J.M. lab for their input and feedback. We thank the McGill University Genome Centre for their expertise in sample quality verification and sequencing services. We thank Dr. Daniel Weinberg and the lab of Dr. Chao Lu for earlier work that generated many of the resources (C3H10T1/2 NSD1/2-DKO, NSD1/2-SETD2-TKO, and H3.3K36M-OE cell lines) that became the core of this research. We thank the group of Dr. Benjamin Garcia for their expertise in mass spectrometry sample processing. We also thank Yanqing Liu and the group of Dr. Scott Rothbart for generating and providing the pCDH-blast-HA-3XFlag-SUV39H1-WT construct. J.M. is supported by the Canadian Institutes of Health Research (CIHR) Grant CIHR PJT-183939 and the United States National Institutes of Health (NIH) Grant P01-CA196539. G.S. was supported by the Gershman Memorial and Kangles Fellowships through the McGill University Faculty of Medicine and Health Sciences. R.P. was supported by the Fonds de Recherche du Québec Santé.

## Author contributions

R.P. and G.S. contributed equally to this work. Study design, G.S., R.P., and J.M. Writing (original draft), G.S., R.P., and J.M. Laboratory experiments, G.S., C.H., and M.G. Bioinformatic and statistical analyses, R.P. Data processing, R.P. and E.B. Supervision, J.M. Funding acquisition, J.M. All author(s) read and approved the final manuscript.

## Competing interests

The authors declare no competing interests.

## Additional information

**Peer review information** : *Nature Communications* thanks the anonymous reviewer(s) for their contribution to the peer review of this work. A peer review file is available.

