## [Peer Review File · Nature Communications]

H3K36 Methylation as a Guardian of Epigenome Integrity

Corresponding Author: Dr Jacek Majewski

Version 0:

Reviewer comments:

Reviewer #1

(Remarks to the Author)

Padilla et al studied effect of combined knock-out (KO) of the H3K36 lysine methyltransferases (KMTases) NSD1-3, ASH1 and SETD2 on genome-wide distribution of H3K36me₂, H3K27me₁₋₃ and H3K9me₃. They found H3K36me₂ marked enhancers. Reduction of H3K36me₂ reduced expression of some proportion of genes that supports the role H3K36 methylation on enhancer regulation of promoters and gene expression. Within the genic region, authors studied enrichment of H3K27me in the various KO mutants and reported H3K36me₂ prohibit H3K27me_{2/3} accumulation. They also showed redistribution of H3K9me₃ from heterochromatin to genic regions. Finally using Hi-C, they showed significant changes in chromatin compartmentalization associated with the loss of H3K36me_{2/3}. The work is extensive and show interesting connection of H3K36me₂ with higher order chromatin organization. If the work can be more connected and mechanistically explored, it would be a significant finding in the field.

Major Comments:

1. The work currently composed of mostly three weakly connected parts: enhancer, H3K9me₃ on heterochromatin and Hi-C compartmentalization study. More work needs to be done to flow these parts into a complete story.
2. It is important to ensure the actual endogenous status of individual KMTases as their names imply (DKO, TKO, QKO and QuiKO). The authors should show immunoblots to demonstrate the level of knockdown of all individual KMTases (NSD1-3, SETD2 and ASH1L) in single KOs, DKO, TKO, QKO and QuiKO to demonstrate that each gene knockout did not ectopically change the levels of the other H3K36 KMTases and that the level of the unaffected KMTases are similar between different mutants.
3. H3K36me profiles of single KMTase knockouts should be included as controls in Fig1a. Profile of H3K27ac should be included in bottom panel of QKO results in Fig 1a.
4. In Figs 1b-c, the decrease in accessibility (20-26%), H3K27ac (30-40%) and increase in H3K27me₃ (20-54%) were partial and significant amount of these still retained on the enhancers in the KO. The authors need to obtain quantitative functional evidence that such residual level is insufficient to support a full activity of these enhancers and the amount of enhancer activity is reduced in QKO relative to TKO as they expected.
5. ChIP-seq of core and linker histones (H1, H2A, H2B, H3 and H4) should be performed to compare between TKO/QKO/QuiKO in Fig 1b-c and 2e-f to assess whether nucleosomal positioning and occupancy are affected by the KO.
6. Fig 1e showed approx. 3x more genes upregulated than downregulated in QKO contradictory to what is expected from compromised enhancer function and authors attributed to "secondary, downstream effects from global H3K36me₂ depletion and/or ablation of NSD3". It is unclear why this was listed as the possible reason. Considering that H3K36me₂ coordinates transcriptional elongation; its depletion is expected to downregulate transcription. It becomes even more confusing when the authors obtained a different number of differentially expressed genes by considering H3K36me₂ enrichment. What is the proportion of genes that overlap between these two lists? Are all the 97 downregulated genes in Fig 1e found within that 150 in Fig 1f? The authors should explain why there is such a big discrepancy between these results.
7. The authors referred to the function and regulation of the KMTases, but none of these proteins were studied in this work. Localization of the H3K36-KMTase proteins should be done in non- and KO cells, relative to the localization of PRC2 and SUV3-9 methyltransferase. Line 218 says "...demonstrating that NSD3 is recruited to these regions". However, there is no evidence to support such claim, and authors should provide ChIP-seq data of NSD3 in TKO versus QKO. For this matter, the authors should perform whole genome ChIP-seq of the KMTases (NSD1-3, SETD2, ASH1L) in their PA and KO cells to correlate with the progressive loss of H3K36me₂.
8. SETD2 is mostly a H3K36 trimethylase. KO of SETD2 is expected to be opposite to the other K36 dimethylases (NSD1-3, ASH1L) by increasing the levels of H3K36me₂ (instead of decreasing). If so, then TKO, QKO and QuiKO would contain higher H3K36me₂ at some loci to confound and mask the effect of the dimethylases KO. The authors should show

experimental evidence that this possibility is not the case, or at least do not significantly confound the effects of the dimethylases KO.

9. The authors should perform whole genome analyses of RNA polymerase II as an independent confirmation of transcriptional changes. Does the invasion of H3K27me3 and H3K9me3 into the gene body correlate with the exclusion of RNA polymerase II from those localities?

10. Why do enhancers retain H3K36me2 so much stronger over the other regions? Is this due to higher NSD3 localization at the enhancers? Do these enhancers contain sequence motifs to preferentially recruit NSD3? Alternatively, would NSD3 be bound preferentially to H3K27ac or H3K4me1 that are enriched at the enhancer sequences?

11. The authors should explain why their results on DNA methylation was inconsistent with previous findings on the recruitment of DNMT3A/B by H3K36me2/3. The authors should check whether localization of DNMT3A/B are affected in their knockout cells.

12. The authors should experimentally test whether the 3D organization disruption could abrogate long-range interaction in the KOs (presumably should be more severe in QuiKO > QKO > TKO) to cause downregulation of enhancer function.

13. Line 323-324: Comparing the difference between ASH1L single KO and parental is unable to conclusively show whether transcriptional downregulation of the genes was due to the loss of catalytic activity. To achieve this aim, authors should instead perform RNA-seq in QuiKO cells ectopically expressed with non-mutated and catalytic site mutant of ASHL1.

14. Line 333-334: "... Genome-wide, we found that chromatin accessibility increases in QuiKO cells...". This increase looked rather marginal (20-30%), and it is unsure whether such small change could amount up to any significant impact to increase accessibility by RNA Pol II and other transcription machinery to affect the observed transcription changes. None of such molecular consequences were shown making the data only correlational.

15. Is the increase in bulk level of H3K9me3 correlated with the increase in the expression level of histone H3 and/or H3K9 methyltransferases? The author should perform ChIP-seq to assess whether RNA polymerase II is excluded by the increase in H3K9me3 in regions not normally occupied by H3K9me3. They should also perform whole genome profiling of other heterochromatin binding proteins including HP1, HDAC, DNMT proteins as independent confirmation of heterochromatin invasion into those regions.

16. The observation on the loss of compartmentalization and long-range chromatin interaction is very interesting that should be defined in more mechanistic details. The authors should perform ChIP-seq analyses in QuiKO on compartments and chromatin loops-forming condensin and cohesin proteins. They should also do ChIP-seq of CTCF to ascertain the loss of boundary function underlie the observed compartment dissolution.

17. The Discussion section is mostly a repeat of the results section. The authors should instead put their results in the context of what is known in the field and comment on the novelty of their findings compared with previous publications. They should also discuss limitations of their studies. They can include a model figure to summarize the major findings of the work.

Minor comments:

(1) Catalogue numbers of the antibodies should be included into the supplementary table.

(2) Line 46, although histone PTM is an epigenetic regulation, it is by no means the only epigenetic mark. Suggest remove "epigenetic or" in this sentence.

(3) Line 50, what do "steric changes" refer to?

(4) Legend of Supplementary Figure 1a is incongruent and needs rewriting.

(5) Line 399, what does the authors mean by "with only a weak dependence on transcription..." Clarify from which data they discern such conclusion.

(6) The authors should replace the word "spreading" with "broadening" (or similar), because spreading has a mechanistic connotation for heterochromatin expansion that involves the binding of silencing PTM by adaptors or KTMase that host PTM recognition motif. The lowering of peakiness score is simply descriptive and must not be confused with the mechanistic spreading event unless the authors can support with experiment evidence.

(7) Title can be more specific.

(Remarks on code availability)

Reviewer #2

(Remarks to the Author)

In this manuscript, the authors combine both new and previously published datasets with multiple new multiomic experiments and new analyses to examine the interplay between H3K36 methylation and other epigenetic modifications and gene expression. In doing so, they add further nuance to emerging data from both in vitro and in vivo systems to underscore the importance of H3K36 methylation in gene regulation. In particular, they reveal new observations regarding large effects on enhancers as well as on constitutive heterochromatin and long-range gene interactions and compartments.

More specifically, the authors make several interesting new observations including the role of NSD3 at enhancers and the redistribution of H3K9me3 with loss of H3K36me2/3. That being said, there are some questions regarding their results that should be addressed:

1) A major limitation of the current study is that virtually all of the experiments are in vitro results done in a single mouse cell line (apart from repeating the H3K9 result in the HNSCC cells). It would be helpful to know how applicable the other key findings are in other cells and contexts of H3K36me reduction/elimination.

2) They observe a redistribution of H3K9me3. Do the authors see any changes in the expression of H3K9me3 histone methyltransferases? What about H3K4me1 methyltransferases (i.e. KMT2C, KMT2D) given the increases in H3K4me1 they

report?

3) In a similar fashion, while they report increases in H3K27me3, do they also observe a redistribution of this modification in their data with regions that also lose H3K27me3? This has been recently described by two manuscripts describing the effects of impaired H3K36 on cell fate (Ko, et al. Dev Cell. 2024; Hoetker, et al. Nat Cell Biol. 2023.), which should be cited here. Along these lines, the authors do not really discuss much regarding the classes or categories of gene expression that change with the different manipulations of H3K36me. This would be interesting to know given all of the associations of dysregulated H3K36 methylation leading to both cancer and altered cell fates. For example, RNA-seq from both the mouse mesenchymal stem cells and HNSCC cells could be compared to see if there are any common genes and pathways that are perturbed through the massive H3K9me3 redistribution.

4) Given the dramatic changes in these repressive chromatin marks, can they use their datasets to determine which one is more associated with the concurrent changes in gene expression that they observe?

5) Do the authors have any thoughts as to why they may not be seeing any change in DNA methylation levels despite their strong reduction of H3K36 methylation?

(Remarks on code availability)

Reviewer #3

(Remarks to the Author)

The manuscript titled "H3K36 Methylation - a Guardian of Epigenome Integrity" by Padilla et al. explores the role of histone H3K36 methylation (H3K36me) in maintaining the integrity of the epigenome, particularly in the context of enhancer activity, gene expression regulation, and the interaction between active and repressive chromatin states. The authors achieved a systematic analysis of H3K36me function by knocking out all five methyltransferases targeting H3K36, individually and in various combinations, allowing the scrutiny of the functions of H3K36me2 and H3K36me3 separately. Their key findings include: 1) establishing the roles of H3K36me2 in sustaining enhancer activity; 2) providing further details on the antagonistic relationship between H3K36me H3K27me; 3) redistribution of H3K9me3 from large heterochromatic domains to euchromatic regions upon H3K36me depletion; the roles of H3K36me in maintaining large scale chromatin compartmentalization. Overall, the experiments described were well designed and executed. Data quality was generally good, and most data analyses are clear and thorough. The data and findings presented here offers new insights into the functions of H3K36me and provide valuable new avenues and resources for the community.

Major concerns:

1. The authors speculate two possible mechanisms for redistribution of H3K9me upon loss of H3K36me without further testing either. This lack of further verification, validation, and mechanistic insights becomes a major apparent deficiency in supporting this novel discovery. ChIP-seq for the H3K9MTs SUV39H1/2 would significantly strengthen this study.
2. The lack of intragenic DNA methylation change in highly expressed genes upon loss of H3K36me is a surprising finding. Given the gain of H3K9me in euchromatin and its direct link to DNA methylation, one could also speculate that the gain of H3K9me compensated for the loss of H3K36me in regulating genic DNA methylation. This scenario can be tested by knocking down/out H3K9MTs in the H3K36MT mutant background.

Minor concerns:

- The labels in some of the figures, especially in Figure 7a, are very small and hard to read. Increase the font size and/or improve figure image quality.

(Remarks on code availability)

No actual code was provided.

Version 1:

Reviewer comments:

Reviewer #2

(Remarks to the Author)

Overall, the authors have made excellent progress in responding to the reviews and have adequately answered my questions. As suggested in my initial review, I believe it would be helpful to cite several important recent articles (listed below) that have looked at the functional effects of perturbed H3K36 methylation, and have shown a redistribution of heterochromatic modifications (namely H3K27me3) and how that leads to tissue-specific phenotypic changes. Given that the current manuscript has been carried out all in in vitro experiments, citing these studies would help support the larger importance of maintaining epigenetic homeostasis in vivo, and also offer an opportunity to suggest how their new findings may actually be playing a larger, yet still unexplored role in these other systems where H3K9 methylation was not assessed.

Pashos ARS, Meyer AR, Bussey-Sutton C, O'Connor ES, Coradin M, Coulombe M, Riemondy KA, Potlapelly S, Strahl BD, Hansson GC, Dempsey PJ, Brumbaugh J. H3K36 methylation regulates cell plasticity and regeneration in the intestinal

epithelium. Nat Cell Biol. 2025 Feb;27(2):202-217. doi: 10.1038/s41556-024-01580-y. Epub 2025 Jan 8. PMID: 39779942.

Ko EK, Anderson A, D'souza C, Zou J, Huang S, Cho S, Alawi F, Prouty S, Lee V, Yoon S, Krick K, Horiuchi Y, Ge K, Seykora JT, Capell BC. Disruption of H3K36 methylation provokes cellular plasticity to drive aberrant glandular formation and squamous carcinogenesis. Dev Cell. 2024 Jan 22;59(2):187-198.e7. doi: 10.1016/j.devcel.2023.12.007. Epub 2024 Jan 9. PMID: 38198888; PMCID: PMC10872381.

Hoetker MS, Yagi M, Di Stefano B, Langerman J, Cristea S, Wong LP, Huebner AJ, Charlton J, Deng W, Haggerty C, Sadreyev RI, Meissner A, Michor F, Plath K, Hochedlinger K. H3K36 methylation maintains cell identity by regulating opposing lineage programmes. Nat Cell Biol. 2023 Aug;25(8):1121-1134. doi: 10.1038/s41556-023-01191-z. Epub 2023 Jul 17. PMID: 37460697; PMCID: PMC10896483.

(Remarks on code availability)

Reviewer #3

(Remarks to the Author)

In this revised manuscript, the authors have carried out additional experiments addressing some key questions raised by the reviewers. These new results, especially the SUV39H1 ChIP-seq experiment offered new insights on H3K9me3 redistribution in the H3K36 methyltransferase mutants. The overall quality and significance of the manuscript is much improved. I recommend this manuscript be accepted for publication.

(Remarks on code availability)

The codes are available with sufficient documentations. No concerns.

Reviewer #4

(Remarks to the Author)

I was asked to review the rebuttal and revised manuscript in place of reviewer one who was unable to re-review this manuscript. As such, I had not previously seen or assessed the original draft. In this position I assessed the quality of the revised manuscript and the author's responses to the reviewers comments (mainly focussing on reviewer one). Having done so I believe the revised manuscript is an interesting and robust assessment of the regulation and regulatory capacity of H3K36me. It represents a large amount of work that is impactful for the field of epigenetics and gene regulation, and explores important topics such as chromatin cross-talk and the impact of altered modification state on higher order chromosomal organisation.

With regard to reviewer comments, the authors did directly address several key concerns, but mainly elected to respectfully disagree with many of the points made. Working through these points, I believe that the additions to the paper in response to all reviewers has strengthened their findings. Where comments were not addressed, I believe that the justification that these were not addressed are sound. Particularly as many questions related to findings that have been previously published by the authors, additional redundant assays to those already performed or simply (costly) experiments that are beyond the scope of this already substantial study.

The results are impactful, critically appraised, robustly performed, interesting and suitable for publication in Nature communications in their current form. I enjoyed reviewing this study.

(Remarks on code availability)

Friday, June 6th, 2025

Dear Reviewers,

We sincerely thank you for your careful and constructive evaluation of our manuscript, "**H3K36 methylation – a Guardian of Epigenome Integrity.**" Your insightful comments and suggestions have significantly strengthened the quality, depth, and clarity of our work.

In response to your feedback, we have undertaken a substantial number of new experiments and analyses to address the key questions raised and expand the mechanistic scope of our findings. These efforts include, but are not limited to, the following major additions:

- **Mechanistic exploration of SUV39H1-mediated H3K9me3 redistribution**, including both overexpression and knockdown of SUV39H1 in H3K36me-deficient cells, paired with ChIP-seq and RNA-seq.
- **ChIP-seq for HP1** to confirm the redistribution of heterochromatin-associated proteins.
- **Expanded cross-species validation** of our most novel findings using two additional human HNSCC cell lines (Cal27 and Detroit562) overexpressing the H3K36M oncohistone. These analyses revealed conserved chromatin alterations in heterochromatin structure, transcriptional output, and 3D genome architecture.
- The integration of **new multi-omic datasets**, including ChIP-seq, CUT&RUN, ATAC-seq, RNA-seq, and Hi-C.

Together, these additions provide a more comprehensive and mechanistically grounded view of how H3K36 methylation protects genome function by maintaining enhancer activity, restricting repressive chromatin, and preserving higher-order nuclear structure. We have revised the manuscript accordingly to reflect these new data, clarified key interpretations, and expanded the discussion of implications, limitations, and future directions.

We greatly appreciate the time and expertise you devoted to reviewing our work. In the following pages, we provide detailed, point-by-point responses to each of your comments, indicating how each concern has been addressed in the revised manuscript.

Sincerely,

Padilla and Shipman et al.

Table of Contents

- I. Reviewer #1 comments and responses (pp. 3-35)
- II. Reviewer #2 comments and responses (pp. 36-48)
- III. Reviewer #3 comments and responses (pp. 49-52)

Reviewer #1

Padilla et al studied effect of combined knock-out (KO) of the H3K36 lysine methyltransferases (KMTases) NSD1-3, ASHL1 and SETD2 on genome-wide distribution of H3K36me2, H3K27me1-3 and H3K9me3. They found H3K36me2 marked enhancers. Reduction of H3K36me2 reduced expression of some proportion of genes that supports the role H3K36 methylation on enhancer regulation of promoters and gene expression. Within the genic region, authors studied enrichment of H3K27me in the various KO mutants and reported H3K36me2 prohibit H3K27me2/3 accumulation. They also showed redistribution of H3K9me3 from heterochromatin to genic regions. Finally using Hi-C, they showed significant changes in chromatin compartmentalization associated with the loss of H3K36me2/3. The work is extensive and show interesting connection of H3K36me2 with higher order chromatin organization. If the work can be more connected and mechanistically explored, it would be a significant finding in the field.

Major Comments:

1. The work currently composed of mostly three weakly connected parts: enhancer, H3K9me3 on heterochromatin and Hi-C compartmentalization study. More work needs to be done to flow these parts into a complete story.

We thank the reviewer for their feedback and the opportunity to clarify the conceptual framework of our study. The central objective of our work is to understand the **downstream consequences of H3K36me depletion**, a hallmark of various developmental disorders and cancers. H3K36 methylation is a key mark of euchromatin that influences transcription, antagonizes the spread of repressive chromatin, and—as

we demonstrate in our current manuscript—contributes to higher-order genome architecture. Our study builds upon this framework by systematically examining its loss at three interrelated levels:

1. **Local chromatin regulation at enhancers and promoters**
2. **The effect on the epigenome—including redistribution of repressive histone modifications, particularly H3K9me3—and transcription**
3. **Global changes in 3D genome organization**

These analyses are conceptually and mechanistically linked by their common dependence on the loss of H3K36me, rather than being independent lines of inquiry.

While we do not claim that all observed downstream effects are directly causally related to one another, they are coordinated consequences of a shared upstream perturbation, and collectively reflect the multi-layered role of H3K36me in genome regulation. As the reviewer notes, H3K9me3-marked heterochromatin and compartmentalization are related phenomena—a connection that is well supported in the literature (e.g., Allshire & Madhani, *Nat Rev Mol Cell Biol*, 2018; Feng et al., *Genome Biol*, 2020; Bian et al., *PNAS*, 2020; Nichols & Cortes, *Cell Rep*, 2021; Clyde, *Nat Rev Genet*, 2021; Spracklin et al., *Nat Struct Mol Biol*, 2023; etc.). Our findings align with and build upon these studies by showing that H3K36me depletion leads to ectopic distribution of H3K9me3 and SUV39H1 in euchromatic regions, which correlates with erosion of chromatin compartmentalization—a novel mechanistic link that expands the functional landscape of H3K36 methylation.

The foundational aspects of this work—including the generation and validation of the individual and multiple knockout mouse mesenchymal stem cell lines and their impact on H3K36 methylation—have been published previously by our group and others (Lu et al., *Science*, 2016; Weinberg et al., *Nature*, 2019; Rajagopalan et al., *PNAS*, 2021; Shipman et al., *Genome Biol*, 2024) and are appropriately cited. This allows the current manuscript to focus on newly uncovered mechanistic relationships without reiterating prior findings. We respectfully disagree that the manuscript presents a disjointed narrative. Reviewers #2 and #3 independently noted the coherence and significance of our study design, and we have further clarified the manuscript structure in the revised text to help guide the reader through the logical progression from local to global consequences of H3K36me loss. In addition, in the revised version of our manuscript we have reproduced our most novel findings in two human cell lines and provided additional mechanistic insights. We hope this better articulates the unifying theme and scientific value of the work.

2. It is important to ensure the actual endogenous status of individual KMTases as their names imply (DKO, TKO, QKO and QuiKO). The authors should show immunoblots to demonstrate the level of knockdown of all individual KMTases (NSD1-3, SETD2 and ASH1L) in single KOs, DKO, TKO, QKO and QuiKO to demonstrate that each gene knockout did not ectopically change the levels of the other H3K36 KMTases and that the level of the unaffected KMTases are similar between different mutants.

We appreciate the reviewer's concern regarding the specificity and validation of the knockout (KO) cell lines used in this study. As outlined in our response to major

comment #1, **all mouse mesenchymal stem cell (mMSC) lines employed here—including single, double (DKO), triple (TKO), quadruple (QKO), and quintuple (QuiKO) knockouts—were generated and thoroughly validated in previously published studies.** No new KO cell lines were generated for the present work. In particular, we refer the reviewer to our recent publication, *“Systematic perturbations of SETD2, NSD1, NSD2, NSD3, and ASH1L reveal their distinct contributions to H3K36 methylation”* (Genome Biology, 2024), in which we directly addressed the potential for compensatory expression changes among the H3K36 KMTases. As shown in Supplementary Figure S6 (also included below for convenience), expression profiling of all five KMTases across the KO lines demonstrated that the majority of knockouts do not induce substantial changes in the expression levels of the remaining KMTases. The one notable exception is a ~2-fold increase in NSD2 expression observed in the NSD1-KO line. However, this increase appears to be functionally insignificant, as NSD1-KO cells exhibit the most profound reduction in intergenic H3K36me2 among all individual KO lines—this suggests that compensatory effects are relatively minor. Taken together, these data support the specificity of each KO and indicate that the loss of one KMTase does not broadly perturb the expression of the others in a manner that would confound interpretation of the observed epigenomic and transcriptomic consequences.

3. H3K36me profiles of single KMTase knockouts should be included as controls in Fig1a. Profile of H3K27ac should be included in bottom panel of QKO results in Fig 1a.

We thank the reviewer for these suggestions. In response, we have now **included both H3K27ac and ATAC-seq profiles for the QKO sample in Fig. 1a** of the revised manuscript to provide a more comprehensive view of regulatory element activity and chromatin accessibility in the context of H3K36me loss. Regarding the inclusion of H3K36me profiles from individual KMTase knockouts: as previously noted, a detailed, genome-wide comparative analysis of H3K36 methylation patterns in all individual KO lines, as well as their combined effects in multiple-KO contexts, has already been conducted and published in our recent study (*Genome Biology*, 2024). We specifically refer the reviewer to **Figure 4** of that study, which addresses the exact comparison

requested. As our current manuscript builds directly upon this previous work, the emphasis here is not on characterizing differences among the individual KMTase knockouts per se, but rather on investigating the downstream consequences of global H3K36me depletion, particularly in the QKO and QuiKO models. These include changes to other epigenetic marks, chromatin organization, and transcriptional programs. We have cited the previous publication throughout the current manuscript to clearly distinguish between previous work and the new mechanistic insights presented in this study.

4. In Figs 1b-c, the decrease in accessibility (20-26%), H3K27ac (30-40%) and increase in H3K27me3 (20-54%) were partial and significant amount of these still retained on the enhancers in the KO. The authors need to obtain quantitative functional evidence that such residual level is insufficient to support a full activity of these enhancers and the amount of enhancer activity is reduced in QKO relative to TKO as they expected.

We respectfully disagree that additional functional assays are necessary to support the conclusion that enhancer activity is reduced. The observed reductions in H3K4me1 and H3K27ac—hallmarks of active enhancers—are widely accepted indicators of diminished enhancer function. H3K4me1 marks enhancers that are poised for activation, while H3K27ac distinguishes active enhancers by promoting an open chromatin environment that facilitates transcription factor binding. Conversely, increased deposition of H3K27me3 is associated with reduced enhancer activity through promotion of chromatin compaction and transcriptional repression (Calo et al., *Mol. Cell*, 2013; Creighton et al., *PNAS*, 2010; Heintzman et al., *Nature*, 2009). Our conclusions in Figures 1b and 1c are

supported by multiple independent and complementary assays: **CUT&RUN**, which reveals a decrease in H3K4me1 and a reciprocal increase in H3K27me3; **ChIP-seq**, which demonstrates a reduction in H3K27ac levels; and **ATAC-seq**, which shows a decrease in chromatin accessibility at enhancer regions. In addition, **RNA-seq** analysis further supports a functional consequence of these epigenomic changes, showing that genes associated with the affected enhancers are downregulated. Specifically, among genes proximal to strong enhancers described in Fig. 1c, 154 are downregulated while only 88 are upregulated (provided again below for convenience), reinforcing the conclusion that enhancer activity is diminished. These findings are grounded in validated genomic annotations (see Methods), and are consistent with a broad body of literature demonstrating that **H3K36 methylation significantly influences enhancer activity** (Yuan et al., *J. Biol. Chem.*, 2011; Streubel et al., *Mol. Cell*, 2018; Yuan et al., *Cancer Discov.*, 2020; Farhangdoost et al., *Cell Rep.*, 2021; Hoetker et al., *Nat. Cell Biol.*, 2023; Barral and Déjardin, *Nucleus*, 2023; etc.). Our current study extends these insights by examining this influence in a localized and mechanistically integrated context.

5. ChIP-seq of core and linker histones (H1, H2A, H2B, H3 and H4) should be performed to compare between TKO/QKO/QuiKO in Fig 1b-c and 2e-f to assess whether nucleosomal positioning and occupancy are affected by the KO.

We agree that quantifying changes to nucleosome dynamics in the absence of H3K36me is an intriguing direction for future investigation. However, we respectfully maintain that this request is not necessary for supporting the current conclusions drawn from Figures 1b–c and 2e–f. These figures aim to demonstrate changes in enhancer and promoter activity through alterations in chromatin accessibility and histone modification states—mechanisms that are already well-characterized by the assays we employed, including ChIP-seq, CUT&RUN, ATAC-seq, RNA-seq, and Whole Genome Bisulfite Sequencing. The reduction in H3K27ac and H3K4me₁, combined with the gain of H3K27me₃ and reduced ATAC-seq signal, robustly indicate decreased regulatory

element activity and chromatin accessibility. These findings are consistent with canonical interpretations of enhancer and promoter inactivation. As cited in our response to Major comment #4, H3K27me3 deposition is associated with facultative heterochromatin and chromatin compaction, while the loss of H3K27ac and H3K4me1 reflects repression of enhancer activity. ATAC-seq directly measures chromatin accessibility, providing a functional readout of nucleosome positioning without the need for core histone ChIP-seq.

Moreover, the assumption that core histone occupancy data would provide substantially more insight in this context is not well supported. ChIP-seq for core histones such as H3 or H2B generally provides a relatively uniform signal across the genome and reflects bulk nucleosome presence and lacks the resolution or specificity to distinguish between active and inactive regulatory states, especially when compared to the epigenetic and transcriptional assays already included (Rando and Chang, *Annu. Rev. Biochem.*, 2009; Henikoff and Shilatifard, *Trends Genet.*, 2011, etc.). Additionally, the enrichment or depletion of histone marks already reflects changes to nucleosomal context and chromatin state. Overall, we believe our multimodal datasets provide sufficient evidence to support our conclusions regarding chromatin and transcriptional changes upon H3K36me depletion, and we consider the additional experiments requested by the reviewer beyond the scope and objectives of this study.

6. Fig 1e showed approx. 3x more genes upregulated than downregulated in QKO contradictory to what is expected from compromised enhancer function and authors attributed to “secondary, downstream effects from global H3K36me2 depletion and/or

ablation of NSD3". It is unclear why this was listed as the possible reason. Considering that H3K36me2 coordinates transcriptional elongation; its depletion is expected to downregulate transcription. It becomes even more confusing when the authors obtained a different number of differentially expressed genes by considering H3K36me2 enrichment. What is the proportion of genes that overlap between these two lists? Are all the 97 downregulated genes in Fig 1e found within that 150 in Fig 1f? The authors should explain why there is such a big discrepancy between these results.

To clarify, although there is a positive correlation between H3K36me2 and gene expression (see Additional File 1: Fig. S5b in Shipman et al., *Genome Biology*, 2024), global changes in gene expression do not necessarily mirror changes in H3K36me levels. The ~3-fold excess of upregulated versus downregulated genes in the QKO versus TKO comparison (Fig. 1e) represents a **global transcriptomic response**, not one solely attributable to compromised enhancer function. Such global effects are also observed in our H3K36M-overexpressing HNSCC models (Cal27 and Detroit562), where similar trends in transcriptional upregulation occur despite near-complete loss of H3K36 methylation (see volcano plots below). It is currently unclear what is driving the genome-wide transcriptional activation. Instead, we emphasize the importance of focusing on functionally relevant gene subsets—particularly those whose enhancers are directly impacted by H3K36me2 loss—to discern specific regulatory mechanisms rather than genome-wide trends. In Fig. 1f, we identified predicted H3K36me2-dependent genes based on proximity to large H3K36me2 peaks overlapping annotated enhancers, and compared them to randomized control genes with similar baseline expression. Within this focused subset, the trend is reversed: 150 genes are downregulated versus

69 upregulated, directly linking the loss of NSD3-deposited H3K36me2 at enhancers to gene repression. The proportion of genes that overlap between the 97 downregulated genes in Fig.1e and the 150 downregulated genes in Fig.1f is only 2%. However, to generate a control set of genes with similar gene expression, we only selected genes between 9 to 12 FPKM for Fig. 1f. Within the 97 downregulated genes in Fig. 1e, only 5 genes meet the criteria of being within 9 to 12 FPKM, with 3 of these genes found in the downregulated genes in Fig. 1f. In the same section of the manuscript, we also compared gene expression changes in NSD3-KO cells relative to parental controls and observed a similar genome-wide trend toward transcriptional upregulation (Supplementary Fig. 1d). However, when we focused specifically on the same set of predicted enhancer-dependent NSD3 target genes identified in TKO cells, we again observed a tendency toward downregulation (Supplementary Fig. 1e). This parallel result supports our conclusion that loss of NSD3-deposited H3K36me2 at enhancers directly impacts the expression of specific target genes, even in the context of partial compensation by other K36 methyltransferases. We also acknowledge in the manuscript that it is important to note that NSD3 may have non-catalytic roles that could influence gene expression independently of H3K36 methylation. For example, NSD3 has been shown to interact with BRD4, and serve as a scaffolding protein in transcriptional complexes (Shen et al., *Molecular Cell*, 2015; etc.). This might partially explain the observation that both NSD3-KO (compared to parental) and QKO (compared to TKO) lead to more upregulation than downregulation, even though H3K36me2 levels are reduced.

Finally, we also wish to address the reviewer’s statement that H3K36me2 “coordinates transcriptional elongation.” To our knowledge, there is no direct evidence supporting a role for H3K36me2 itself in coordinating transcriptional elongation. Most available evidence attributes transcription elongation functions to **H3K36me3**, deposited by SETD2—which tethers to the elongating Pol II complex—and associated with recruitment of factors for splicing and DNA repair fidelity (Molenaar and Leeuwen, *Cell. and Mol. Life Sci.*, 2022; etc.). In contrast, H3K36me2 has primarily been linked to intergenic chromatin organization, enhancer regulation, and DNA methylation templating (Weinberg et al., *Nature*, 2019; etc). We would appreciate a specific citation if the reviewer is referring to a study suggesting a role for H3K36me2 in elongation.

Cal27

Detroit

7. The authors referred to the function and regulation of the KMTases, but none of these proteins were studied in this work. Localization of the H3K36-KMTase proteins should be done in non- and KO cells, relative to the localization of PRC2 and SUV3-9 methyltransferase. Line 218 says "...demonstrating that NSD3 is recruited to these regions". However, there is no evidence to support such claim, and authors should provide ChIP-seq data of NSD3 in TKO versus QKO. For this matter, the authors should perform whole genome ChIP-seq of the KMTases (NSD1-3, SETD2, ASH1L) in their PA and KO cells to correlate with the progressive loss of H3K36me2.

As we have noted in response to earlier comments, we refer the reviewer to our previously published manuscript, which contains most of the foundational and validation work for these cell lines (Genome Biology, 2024). While we did not perform ChIP-seq on the KMTases themselves, our experimental framework is specifically designed to

dissect their enzymatic contributions. The TKO cells lack NSD1, NSD2, and SETD2, and retain only a small set of H3K36me2 peaks. When NSD3 is ablated to generate the QKO line, most of the remaining peaks are entirely lost. This functional epistasis is strong evidence that NSD3 is solely responsible for depositing the residual H3K36me2 in the TKO background. To further confirm this, we performed NSD3 overexpression in QKO cells and observed restoration of H3K36me2 at active promoters and enhancers, approaching the levels seen in TKO cells (see heatmaps, below). These functional gain- and loss-of-function experiments establish a direct role for NSD3 in H3K36me2 deposition at these regulatory elements. Thus, our conclusion that “NSD3 is recruited to these regions” reflects clear biochemical evidence derived from endogenous and ectopic NSD3 perturbation experiments. We do not present the overexpression experiment in our current manuscript as the individual and combined contributions of the K36 KMTases were rigorously evaluated across several previous studies (see citations in response to previous comments). Similarly, our previous study showed that ASH1L knockout in the QKO background—where all other K36 KMTases are absent—abolishes all remaining H3K36me2 peaks, indicating that ASH1L is responsible for depositing the final ~119 H3K36me2 peaks in the genome. These peaks localize to developmentally regulated genes and are likely targeted via PBX2, as demonstrated in that work. We reiterate that extensive functional and epigenomic validation of each KMTase has been performed and published, and the present study builds on that established foundation to address the regulatory consequences of H3K36me2 loss at cis-regulatory elements.

8. SETD2 is mostly a H3K36 trimethylase. KO of SETD2 is expected to be opposite to the other K36 dimethylases (NSD1-3, ASH1L) by increasing the levels of H3K36me2 (instead of decreasing). If so, then TKO, QKO and QuiKO would contain higher H3K36me2 at some loci to confound and mask the effect of the dimethylases KO. The

authors should **show experimental evidence** that this possibility is not the case, or at least do not significantly confound the effects of the dimethylases KO.

We appreciate the reviewer's question but note that this concern has also been addressed in detail in our prior study (*Genome Biology*, 2024). As shown there, the individual knockout of SETD2 does lead to an increase in H3K36me2 levels within gene bodies—where SETD2 normally deposits H3K36me3. However, this effect is dependent on the presence of functional dimethylases. In the QKO and QuiKO cells, where NSD1, NSD2, and NSD3 are already ablated, global H3K36me2 is nearly eliminated: only 0.03% and 0.009% of peptides, respectively, retain this mark (as shown by both mass spectrometry and Western blot in our previous publication, and provided again below for convenience). Moreover, even with SETD2 knockout in the TKO cells, global H3K36me2 levels are significantly reduced compared to DKO (from 8.97% to 4.60% of H3K36me2-modified peptides), demonstrating that SETD2-KO does not obscure or compensate for the loss of the dimethylases. Lastly, as also discussed in our prior work, SETD2 may contribute modestly to H3K36me2 deposition at some loci, but this contribution is minor and does not alter our interpretation of NSD1/2/3 function. Collectively, these data rule out the possibility that SETD2-KO confounds or masks the effects of NSD-family KMTase ablation.

9. The authors should perform **whole genome analyses of RNA polymerase II as an independent confirmation of transcriptional changes**. Does the invasion of H3K27me3 and H3k9me3 into the gene body correlate with the exclusion of RNA polymerase II from those localities?

While determining the exact mathematical relationships between the levels of repressive modifications, RNA polymerase II occupancy, and nascent transcript levels is a worthy future goal, it is beyond the scope of our current study. Although we have not directly profiled RNA polymerase II occupancy, the well-established mutual exclusivity

between these repressive marks and Pol II (Barski et al., Cell, 2007; etc.) and the robust correlation between transcriptional output—as measured by RNA-seq—and Pol II activity (Honkela et al., PNAS, 2015; Pretis et al., Genome Res., 2017; Zhu et al., Nat. Plants, 2018; Kaye et al. Nat. Comm., 2024) together support the inference that accumulation of H3K27me3 and H3K9me3 in gene bodies is accompanied by reduced Pol II occupancy. Furthermore, the observed negative correlation between these histone marks and RNA-seq signal in our datasets is consistent with such exclusion (see statistics for a multiple linear regression model below). Thus, we believe our RNA-seq-based conclusions sufficiently capture the underlying transcriptional repression associated with these chromatin modifications.

RNA-Seq log2 fold changes (TKO/PA)	Coefficient	Std. error	t-value	p-value
Δ H3K27me3	-1.034	0.056	-18.435	0.000
Δ H3K9me3	-0.236	0.107	-2.194	0.0283
Δ H3K27me3: Δ H3K9me3	-2.408	0.214	-11.216	0.000

DF = 6425

Residual standard error = 2.274

P-value = 0.000

R-squared = 0.0746

Adjusted R-squared = 0.0742

10. Why do enhancers retain H3K36me2 so much stronger over the other regions? Is this due to higher NSD3 localization at the enhancers? **Do these enhancers contain sequence motifs to preferentially recruit NSD3?** Alternatively, would NSD3 be bound preferentially to **H3K27ac or H3K4me1 that are enriched at the enhancer sequences?**

In our recently published paper (Shipman et al., 2024), we have shown that NSD3 is preferentially recruited to active enhancers and promoters (also see heatmaps in response to major comment #7), although this is only uncovered after knocking out

NSD1 and NSD2. After performing a HOMER motif analysis for open chromatin regions (as determined by ATAC-Seq peaks) within NSD3-deposited H3K36me2 regions, we found 50 significantly enriched TF binding sites. However, due to the large size of these remaining H3K36me2 peaks, we cannot confirm whether these are indeed NSD3 targets or simply enriched TF binding sites found at cis-regulatory elements. Below are the top 10 significantly enriched TF binding sites at the remaining H3K36me2 peaks found in TKO but absent in QKO. Those bZIP motifs are the most common general transcription activator sequences that we find in many analyses and many cell types, suggesting that NSD3 operates in a general open chromatin context. To avoid unnecessary speculation and for brevity, we prefer not to include this relatively uninformative analysis in the manuscript. Regarding chromatin context, we cannot ascertain whether NSD3 recruitment is driven by co-occurrence with H3K27ac or H3K4me1, as these marks are commonly present together at active enhancers, and experimentally resolving this lies beyond the scope of the present study. Overall, our data suggest that NSD3 preferentially functions in transcriptionally permissive regions, but further dissection of its recruitment determinants is beyond the scope of this study.

Motif	Name	P-value	log P-value	q-value (Benjamini)	# Target Sequences with Motif	% of Targets Sequences with Motif	# Background Sequences with Motif	% of Background Sequences with Motif
	BATF(bZIP)/Th17-BATF-ChIP-Seq(GSE39756)/Homer	1e-5	-1.185e+01	0.0034	875.0	48.02%	12761.4	42.95%
	AP-1(bZIP)/ThioMac-PU.1-ChIP-Seq(GSE21512)/Homer	1e-5	-1.175e+01	0.0034	912.0	50.05%	13366.8	44.98%
	Atf3(bZIP)/GBM-ATF3-ChIP-Seq(GSE33912)/Homer	1e-4	-1.121e+01	0.0034	883.0	48.46%	12939.1	43.54%
	JunB(bZIP)/DendriticCells-Junb-ChIP-Seq(GSE36099)/Homer	1e-4	-1.024e+01	0.0042	811.0	44.51%	11857.5	39.91%
	Jun-AP1(bZIP)/K562-cJun-ChIP-Seq(GSE31477)/Homer	1e-4	-9.488e+00	0.0072	546.0	29.97%	7723.2	25.99%
	Fra2(bZIP)/Striatum-Fra2-ChIP-Seq(GSE43429)/Homer	1e-3	-8.599e+00	0.0145	760.0	41.71%	11178.4	37.62%
	Fra1(bZIP)/BT549-Fra1-ChIP-Seq(GSE46166)/Homer	1e-3	-8.542e+00	0.0145	809.0	44.40%	11969.7	40.28%
	Fos(bZIP)/TSC-Fos-ChIP-Seq(GSE110950)/Homer	1e-3	-8.291e+00	0.0148	832.0	45.66%	12363.0	41.61%
	Fos12(bZIP)/3T3L1-Fos12-ChIP-Seq(GSE56872)/Homer	1e-3	-8.127e+00	0.0155	643.0	35.29%	9357.1	31.49%
	ETV4(ETS)/HepG2-ETV4-ChIP-Seq(ENCODE)/Homer	1e-3	-7.592e+00	0.0238	810.0	44.46%	12073.3	40.63%

11. The authors should explain why their results on DNA methylation was inconsistent with previous findings on the recruitment of DNMT3A/B by H3K36me2/3. The authors should check whether localization of DNMT3A/B are affected in their knockout cells.

While prior studies have shown that DNMT3A/B recruitment can be guided by H3K36me2/3 (Weinberg et al., 2019), our findings in mMSCs reveal a more nuanced context-dependent relationship. When examining intergenic regions (IGRs) within cluster A regions in our TKO samples, we observed a 15% decrease in DNA methylation, which coincided with the spread of H3K9me3 and H3K27me3 (see below).

In lowly expressed genes, DNA methylation is reduced approximately 20% in TKO and 15% in QKO and QuiKO cells (Fig. 4a; middle box). Highly expressed genes, on the other hand, retain high methylation levels despite complete loss of H3K36 methylation (Fig. 4a; rightmost box). To investigate this further, we have now created DNMT3A/B double-knockout mouse mesenchymal stem cells and found they phenocopied the methylation loss observed in our multiple-KO models at lowly expressed genes, but had minimal impact at highly expressed loci (see below). This suggests that DNA methylation maintenance at actively transcribed regions is largely independent of DNMT3A/B and H3K36me2/3, likely reflecting DNMT1-mediated maintenance mechanisms. We are currently investigating the contributions of de novo versus

maintenance methyltransferases and their respective interactions with histone modifications, but this project is beyond the scope of the current manuscript.

12. The authors should **experimentally test** whether the 3D organization disruption could abrogate long-range interaction in the KOs (presumably should be more severe in QuiKO > QKO > TKO) to cause downregulation of enhancer function.

We appreciate the reviewer's interest in the connection between chromatin architecture and enhancer function. While the requested experiments are beyond the current study's aims, our Hi-C data demonstrate that long-range interactions (loops with > 500 kb distance from anchor to anchor) are significantly disrupted in both our mMSC and HNSCC models following H3K36me depletion. As shown in the barplots below, these disruptions are more pronounced in QuiKO cells than in TKO cells. To analytically assess whether disruptions to chromatin loops affect enhancers and their associated target genes, we identified annotated enhancers located at the anchors of the affected loops. We then determined their corresponding target genes and quantified changes in gene expression. Notably, we found that genes associated with these disrupted loops exhibit preferential downregulation (TKO: 163 downregulated vs 123 upregulated; QuiKO: 173 downregulated vs 109 upregulated). These findings suggest that the loss of long-range chromatin interactions is associated with reduced enhancer function,

consistent with the observed downregulation of their target genes. These results have been incorporated into our revised manuscript (Supplementary Figure 8k).

Genes associated with disrupted chromatin loops

13. Line 323-324: Comparing the difference between ASH1L single KO and parental is unable to conclusively show whether transcriptional downregulation of the genes was due to the loss of catalytic activity. To achieve this aim, authors should instead perform RNAseq in QuiKO cells ectopically expressed with non-mutated and catalytic site mutant of ASHL1.

We thank the reviewer for raising this point regarding the potential distinction between catalytic and non-catalytic functions of ASH1L. While we agree that ectopic expression of wild-type versus catalytically inactive ASH1L in QuiKO cells would, in theory, allow for a direct assessment, this experiment is not currently feasible due to the enormous size of the ASH1L protein (>300 kDa), which poses significant technical barriers to overexpression. Neither we, nor any of our collaborators, have been able to express exogenous ASH1L in any cell type. That said, we respectfully disagree with the reviewer's assessment that our existing data are insufficient to support a catalytic role for ASH1L. As described in the manuscript, we focused our analyses on a specific set of 84 genes associated with the 119 residual H3K36me2 peaks deposited by ASH1L in QKO cells. When ASH1L is ablated (i.e., in QuiKO cell), these peaks—and their associated H3K36me2 signal—are lost, and a majority of the associated genes are significantly downregulated. To further interrogate whether this transcriptional downregulation could be attributed to the absence of ASH1L protein per se (rather than its catalytic activity), we examined gene expression in parental versus ASH1L single-KO cells. Importantly, in these cells, H3K36me2 is retained at the promoters of these target genes, likely due to compensatory activity from other K36 methyltransferases. Under these conditions, we observe no significant change in gene expression. This

comparative analysis provides support that the catalytic product (H3K36me2), rather than ASH1L protein alone, plays the predominant role in maintaining expression at these loci. Finally, we note that our overall experimental design mirrors the logic employed in our NSD3 analyses, in which we considered only genes associated with H3K36me2-marked enhancers, allowing us to more specifically attribute expression changes to H3K36me2 depletion.

14. Line 333-334: "... Genome-wide, we found that chromatin accessibility increases in QuiKO cells...". This increase looked rather marginal (20-30%), and it is unsure whether such small change could amount up to any significant impact to increase accessibility by RNA Pol II and other transcription machinery to affect the observed transcription changes. None of such molecular consequences were shown making the data only correlational.

In Section 2, our intention was not to imply that the modest genome-wide increase in chromatin accessibility observed in QuiKO cells directly facilitates transcriptional upregulation. Rather, we aimed to emphasize a contrasting and more functionally relevant observation: that cis-regulatory elements previously marked with residual H3K36me2 in QKO cells exhibit a significantly attenuated gain in accessibility relative to the genome-wide trend upon ASH1L loss in QuiKO cells. This disproportionate reduction in accessibility at these loci likely reflects localized functional consequences of H3K36me2 loss. Supporting this interpretation, we observed that these same regions concurrently lose H3K27ac and gain H3K27me3 (Fig. 2e, f), consistent with a shift toward a repressive chromatin state. Furthermore, genes associated with these residual H3K36me2-marked regions—which lose this mark in QuiKO—are significantly

downregulated (Fig. 2c), reinforcing a mechanistic link between the local loss of H3K36me2 and transcriptional repression.

15. Is the increase in bulk level of H3K9me3 correlated with the increase in the expression level of histone H3 and/or H3K9 methyltransferases? The author should perform ChIP-seq to assess whether RNA polymerase II is excluded by the increase in H3K9me3 in regions not normally occupied by H3K9me3. They should also perform whole genome profiling of other heterochromatin binding proteins including **HP1**, HDAC, DNMT proteins as independent confirmation of heterochromatin invasion into those regions.

In response to the first point: we examined the expression levels of major H3K9 methyltransferases (SUV39H1, SUV39H2, SETDB1) and heterochromatin protein 1 (HP1; CBX5 in mouse) and found no significant changes in transcript levels between parental and TKO cells (see barplots below). This suggests that the changes in H3K9me3 distribution is not driven by increased expression of these enzymes. This has been included in the revised version of our manuscript.

In response to the suggestion to profile additional heterochromatin-binding proteins: we performed ChIP-seq for both HP1 and FLAG-tagged SUV39H1 in the context of H3K36me-deficient cells. These experiments revealed that SUV39H1 redistributes from typically inactive (cluster B) regions to active (cluster A) chromatin following H3K36me depletion (Fig. 5e, f). Similarly, HP1 binding increased at active gene bodies and transposable elements in cluster A (Fig. 6a, b; Supplementary Fig. 6a), mirroring the redistribution of H3K9me3. These results have been included in the

revised manuscript and provide direct evidence that heterochromatin machinery invades previously active chromatin domains in response to H3K36me loss.

Regarding the recommendation to perform ChIP-seq for RNA polymerase II, HDACs, and DNMTs: we respectfully disagree that these experiments are necessary to support our conclusions. First, the exclusion of RNA Pol II from repressive chromatin marked by H3K9me3 is a well-established phenomenon (see citations from previous comments), and our RNA-seq data already show significant downregulation of genes within regions where H3K9me3 and HP1 are enriched. Thus, the functional consequence—transcriptional repression—is already evident without needing Pol II occupancy data. Second, the redistribution of SUV39H1 and HP1, along with increased H3K9me3 and transcriptional repression, sufficiently demonstrate the expansion of heterochromatin into previously active regions.

16. The observation on the loss of compartmentalization and long-range chromatin interaction is very interesting that should be defined in more mechanistic details. The authors should perform ChIP-seq analyses in QuiKO on compartments and chromatin

loops-forming condensin and cohesin proteins. **They should also do ChIP-seq of CTCF to ascertain the loss of boundary function underlie the observed compartment dissolution.**

We thank the reviewer for their interest in the mechanistic basis underlying the observed disruption of 3D genome organization in H3K36me-deficient cells. In response to the suggestion to assess boundary and loop-forming factors, we performed ChIP-seq for CTCF in both mMSCs (TKO and QuiKO) and Cal27 cells overexpressing H3K36M. In all cases, we observed a genome-wide increase in CTCF binding, including at both cluster A and B regions as well as at compartment boundaries (see below). This surprising—yet consistent—increase in CTCF occupancy suggests that the loss of compartmentalization and long-range interactions is not due to diminished CTCF binding or boundary element collapse. While this increased CTCF binding is highly intriguing, it does not explain the dissolution of compartment boundaries, and we don't feel that it would contribute additional clarity to the present observations on compartmentalization. We are currently investigating the underlying causes, but resolving this question will require extensive additional experiments beyond the scope of this manuscript. Furthermore, we agree that further mechanistic insights into the role of loop extrusion factors such as cohesin and condensin would be valuable in future studies, however, dissecting their redistribution or potential alterations is not necessary for the results presented and beyond the current scope.

17. The Discussion section is mostly a repeat of the results section. The authors should instead put their results in the context of what is known in the field and comment on the novelty of their findings compared with previous publications. **They should also discuss limitations of their studies.** They can include a model figure to summarize the major findings of the work.

We thank the reviewer for their feedback. We have revised our discussion to include study limitations, and a model to summarize the major findings (Fig. 9).

Minor comments:

(1) Catalogue numbers of the antibodies should be included into the supplementary table.

The catalogue numbers for each antibody was included in the provided supplementary table. A dedicated column detailing the catalogue numbers was made in the revised manuscript for clarity (Supplementary Table #2; Additional Files).

(2) Line 46, although histone PTM is an epigenetic regulation, it is by no means the only epigenetic mark. Suggest remove “epigenetic or” in this sentence.

We have removed “epigenetic or” as suggested in line 46.

(3) Line 50, what do “steric changes” refer to?

“Steric changes” refers to hindering or facilitating reactions via the physical or chemical presence of the PTM. In the revised manuscript, this has been reworded in the associated text for clarity.

(4) Legend of Supplementary Figure 1a is incongruent and needs rewriting.

We have revised the legend for Supplementary Figure 1a to include more comprehensive details that precisely match the heatmap visualization. Specifically, the legend has been rewritten to: “Correlation matrix heatmaps of genome-wide H3K27ac, H3K4me1, H3K27me3 and H3K4me3 signals, binned in 1 kb windows, demonstrate strong reproducibility between replicates, with samples from each cell line clustering together. Pearson’s correlation coefficients were used.”

(5) Line 399, what does the authors mean by "with only a weak dependence on transcription..." Clarify from which data they discern such conclusion.

What we intended to convey is that the redistribution of H3K27me2/3 into gene bodies in H3K36me-deficient cells occurs **largely independently of transcriptional activity**. This conclusion is based on the data shown in **Figure 3a**, where genes are stratified by their transcriptional activity (high, medium, low). In parental cells, the exclusion of H3K27me3 from gene bodies is more pronounced in highly expressed genes, as expected due to the mutually exclusive relationship between transcription and PRC2-mediated repression. However, in the TKO and higher-order knockout conditions (QKO, QuiKO), H3K27me2 and H3K27me3 accumulate within gene bodies across all expression levels—not just in lowly expressed genes.

This pattern indicates that the depletion of H3K36me permits H3K27me2/3 to spread into gene bodies irrespective of transcriptional status. If the effect were strictly transcription-dependent, one would expect invasion primarily in silent or weakly transcribed genes, not in those with active transcription. Thus, the redistribution of H3K27 methylation into gene bodies shows **only a weak dependence on transcription**, meaning the effect is not strongly modulated by transcriptional activity and occurs broadly when H3K36 methylation is lost. We have clarified this language in the revised manuscript (now line 428).

(6) The authors should replace the word “spreading” with “broadening” (or similar), because spreading has a mechanistic connotation for heterochromatin expansion that involves the binding of silencing PTM by adaptors or KTMase that host PTM recognition

motif. The lowering of peakiness score is simply descriptive and must not be confused with the mechanistic spreading event unless the authors can support with experiment evidence.

We have replaced “spreading” with “broadening”.

(7) Title can be more specific.

We agree that the title is broad, but we would like it to remain intriguing and to invite the reader to read the entire manuscript to learn how various components of H3K36 methylation contribute to the integrity of the epigenome.

Reviewer #2

In this manuscript, the authors combine both new and previously published datasets with multiple new multiomic experiments and new analyses to examine the interplay between H3K36 methylation and other epigenetic modifications and gene expression. In doing so, they add further nuance to emerging data from both in vitro and in vivo systems to underscore the importance of H3K36 methylation in gene regulation. In particular, they reveal new observations regarding large effects on enhancers as well as on constitutive heterochromatin and long-range gene interactions and compartments. More specifically, the authors make several interesting new observations including the role of NSD3 at enhancers and the redistribution of H3K9me3 with loss of H3K36me2/3. That being said, there are some questions regarding their results that should be addressed:

1) A major limitation of the current study is that virtually all of the experiments are in vitro results done in a single mouse cell line (apart from repeating the H3K9 result in the HNSCC cells). **It would be helpful to now how applicable the other key findings are in other cells and contexts of H3K36me reduction/elimination.**

We agree with the reviewer that generalizability beyond a single cell type is essential for broader interpretation of our findings. In this revised version of the manuscript, we have substantially expanded the scope of our analyses by validating key findings in **two additional human HNSCC cell lines (Cal27 and Detroit562)**, which do not harbor endogenous mutations affecting H3K36 methylation. In these lines, we overexpressed the **H3K36M** oncohistone, which reduces both H3K36me2 and H3K36me3 by more than threefold. Importantly, this mutation is clinically relevant and has been implicated in

several cancer types. Thus, our use of the H3K36M model provides valuable mechanistic insight into the epigenetic dysregulation driven by this cancer-associated mutation, thereby broadening the biological and translational impact of our findings. To comprehensively assess the effects of H3K36me depletion, we performed a full panel of genome-wide assays in Cal27 and Detroit562 cells, including: **Mass Spectrometry**, **ChIP-seq** (H3K9me3, and H3K36me2/3), **CUT&RUN** (H3K9me3), **ATAC-seq**, **RNA-seq**, and **Hi-C**. Across both human cell lines, these assays independently recapitulated our key findings in mouse mesenchymal stem cells (mMSCs), including:

- **Loss of broad H3K9me3 domains and invasion of H3K27me3** into these regions (Figs. 5b-d and Supplementary Figs. 5b-e).
- **Gain of chromatin accessibility and active chromatin marks** at previously silenced H3K9me3-marked loci (cluster B), as shown by ATAC-seq and RNA-seq (Figs. 6f,g,i and Supplementary Figs. 6e-i,k).
- **Gene and TE expression changes** at previously silenced H3K9me3-marked loci (cluster B), as shown by RNA-seq (Figs.7a,c,d and Supplementary Figs. 7b,c)
- **Decompartmentalization of nuclear architecture**, with weakening of B-B interactions, A-to-B compartment shifts, and widespread **loss of long-range chromatin loops**, as revealed by Hi-C (Figs. 8b-e, and Supplementary Figs. 8b-k).
- **Disruption of nuclear H3K9me3 foci**, with reduced numbers and more internal localization of these domains (Fig. 8g,h and Supplementary Fig. 8m), supporting architectural destabilization.

We did not replicate the enhancer-specific analyses (Sections 1–2) or PRC2 antagonism studies (Sections 3–4) in Cal27 or Detroit562 cells due to experimental and interpretive constraints inherent to the H3K36M mutation:

1. **Sections 1–2 (Enhancer regulation):** The effects studied in these sections rely on localized removal of H3K36me2 in TKO (NSD3-specific) and QKO (ASH1L-specific) cells. In contrast, H3K36M-OE causes widespread loss across megabase-scale domains, making it infeasible to resolve the subtle distinctions in enhancer regulation at a resolution comparable to our CRISPR-based deletion models.
2. **Section 3 (H3K36me2/3 antagonism of PRC2):** The H3K36M model depletes both H3K36me2 and H3K36me3, which precludes distinguishing their relative contributions, as we were able to do using SETD2-KO, NSD1/2-DKO, and NSD1/2-SETD2-TKO samples in mMSC lines.
3. **Section 4 (Progressive chromatin changes in gene bodies):** The H3K36M-OE system does not allow for titrated depletion of H3K36 methylation. Thus, it is incompatible with our analysis of stepwise chromatin invasion and transcriptional effects across multiple engineered genotypes (mMSC cell lines).

Together, these new human cell line data confirm the reproducibility of our most salient findings and highlight the conserved role of H3K36 methylation in maintaining heterochromatin architecture, 3D genome organization, and epigenetic insulation across distinct biological systems.

2) They observe a redistribution of H3K9me3. Do the authors see any changes in the

expression of H3K9me3 histone methyltransferases? What about H3K4me1 methyltransferases (i.e. KMT2C, KMT2D) given the increases in H3K4me1 they report?

We thank the reviewer for their inquiry. In response, we performed differential expression analysis of the key H3K9 and H3K4 methyltransferases to evaluate whether transcriptional changes in these enzymes might contribute to the observed redistribution of H3K9me3 and H3K4me1, respectively. For H3K9me3, we analyzed the expression of SUV39H1, SUV39H2, SETDB1, and HP1 (CBX5 in mouse) in parental and TKO mMSCs, and also in Cal27 and Detroit562 HNSCC cells. None of these showed significant changes in expression, suggesting that the redistribution of H3K9me3 is not driven by altered expression of its core methyltransferases (see barplots below). For H3K4me1, which we found to increase globally, we examined the expression of KMT2C and KMT2D, the primary H3K4me1 methyltransferases. Interestingly, KMT2C expression was modestly reduced, whereas KMT2D—which is ~3-fold more highly expressed than KMT2C in mMSCs—was significantly upregulated in TKO cells (see barplots below). This trend was also observed across our other multiple-KO mMSC models. We have now incorporated these findings into the revised manuscript, including the relevant expression barplots and discussion (H3K9 methyltransferases in **Supplementary Fig. 5f** and H3K4 methyltransferases in **Supplementary Fig. 4d**).

3) In a similar fashion, while they report increases in H3K27me3, do they also observe a redistribution of this modification in their data with regions that also lose H3K27me3?

This has been recently described by two manuscripts describing the effects of impaired H3K36 on cell fate (Ko, et al. *Dev Cell*. 2024; Hoetker, et al. *Nat Cell Biol*. 2023.),

which should be cited here. Along these lines, the authors do not really discuss much regarding the classes or categories of gene expression that change with the different manipulations of H3K36me. This would be interesting to know given all of the associations of dysregulated H3K36 methylation leading to both cancer and altered cell fates. For example, RNA-seq from both the mouse mesenchymal stem cells and HNSCC cells could be compared to see if there are any common genes and pathways that are perturbed through the massive H3K9me3 redistribution.

We thank the reviewer for their suggestions. In contrast to our observations of H3K9me3, where we detect both gains and losses across distinct genomic regions, H3K27me3 shows a global gain with little evidence of regional loss in our TKO mMSCs. As shown in the included scatterplot (please see below), where H3K27me3 ChIP-seq signal was binned into 10 kb windows, there is an increase of H3K27me3 in both genic and intergenic regions in mMSC TKO compared to parental cells (coloured bins to the left of the diagonal). This increase indicates a spreading rather than a redistribution of H3K27me3. In contrast, the H3K9me3 scatterplot (Fig. 5b) shows data points to the right of the diagonal (cluster B), reflecting regional H3K9me3 loss. Bins on the diagonal indicate similar levels of signal between the two conditions.

Among the genes that lost H3K9me3 in cluster B regions in mMSC TKO and HNSCC H3K36M-OE cells (Fig. 7a), 132 were common to both cell lines. Approximately 35% (46/132) of these genes are olfactory receptor (OR) genes (see results below). This is consistent with previous findings in which Magklara et al. (2011) found that OR genes are heavily marked by H3K9me3 in mice. Another notable group of genes we've found to be significantly enriched in these regions of H3K9me3 loss in both HNSCC cells and mMSCs are gamma-aminobutyric acid receptor (GABA) receptor genes. There is some indication in previous studies that H3K9me3 may directly repress GABA receptor expression (Ionescu-Tucker et al., 2021). Furthermore, GABA receptor expression

modulates brain-derived neurotrophic factor (BDNF), which has been shown to be inhibited by H3K9me3 in mice (Snigdha et al., 2016). Thus, in mMSCs and HSNCC, H3K9me3 may act on the upstream regulators of BDNF instead, which are the GABA receptors. These findings have been included in our revised manuscript (Fig. 7c, d).

Furthermore, we examined genes gaining H3K9me3 in cluster A regions that were common between mMSC and Cal27. We identified 1,237 genes with elevated H3K9me3 at their gene bodies. These genes were enriched for mitochondrial and metabolic gene ontology terms, such as the tricarboxylic acid cycle (see below).

Genes gaining H3K9me3 in cluster A

Gene ontology (GO)

4) Given the dramatic changes in these repressive chromatin marks, can they use their datasets to determine which one is more associated with the concurrent changes in gene expression that they observe?

Yes—using partial correlation analyses, we assessed how changes in H3K27me3 and H3K9me3 relate to gene expression. In cluster A regions, where most transcriptional

changes occur, H3K27me3 changes were more strongly associated with gene expression than H3K9me3, even though both marks are positively correlated (left plot below). In cluster B regions, which show H3K9me3 loss, gene upregulation correlated most with H3K9me3 depletion. While H3K27me3 increased in these regions, it did not offset the transcriptional activation and was not significantly linked to expression changes (right plot below). Overall, H3K27me3 is the main epigenetic correlate of expression changes genome-wide, however, the loss of H3K9me3 plays a larger role within cluster B regions.

5) Do the authors have any thoughts as to **why they may not be seeing any change in DNA methylation levels despite their strong reduction of H3K36 methylation?**

We did observe modest but consistent reductions in DNA methylation following loss of H3K36 methylation. Specifically, in TKO cells, there was an ~20% decrease in DNAm_e at transcribed gene bodies (Fig. 4a, middle box). Additionally, intergenic regions within cluster A domains—where H3K27me₃ and H3K9me₃ spread—showed a ~15% reduction in DNAm_e (see plot below). Interestingly, in our multiple-KO cells, DNAm_e was predominantly reduced at lowly expressed genes, while highly expressed genes retained high methylation levels, despite near-complete H3K36me depletion (Fig. 4a). To explore this further, we generated DNMT3A/B double-knockout (DKO) mMSCs, as DNMT3A and DNMT3B are known to interact with H3K36me_{2/3} (Weinberg et al., 2019). The DNMT3A/B-DKO cells showed a similar pattern to the multiple-KO cells: loss of DNA methylation at lowly expressed genes but no reduction at highly expressed genes.

These findings suggest that DNA methylation at these loci is maintained independently of H3K36 methylation, likely via DNMT1-mediated maintenance methylation.

Reviewer # 3

The manuscript titled "H3K36 Methylation - a Guardian of Epigenome Integrity" by Padilla et al. explores the role of histone H3K36 methylation (H3K36me) in maintaining the integrity of the epigenome, particularly in the context of enhancer activity, gene expression regulation, and the interaction between active and repressive chromatin states. The authors achieved a systematic analysis of H3K36me function by knocking out all five methyltransferases targeting H3K36, individually and in various combinations, allowing the scrutiny of the functions of H3K36me₂ and H3K36me₃ separately. Their key findings include: 1) establishing the roles of H3K36me₂ in sustaining enhancer activity; 2) providing further details on the antagonistic relationship between H3K36me H3K27me; 3) redistribution of H3K9me₃ from large heterochromatic domains to euchromatic regions upon H3K36me depletion; the roles of H3K36me in maintaining large scale chromatin compartmentalization. Overall, the experiments described were well designed and executed. Data quality was generally good, and most data analyses are clear and thorough. The data and findings presented here offers new insights into the functions of H3K36me and provide valuable new avenues and resources for the community.

Major concerns:

1. The authors speculate two possible mechanisms for redistribution of H3K9me upon loss of H3K36me without further testing either. This lack of **further verification, validation, and mechanistic insights** becomes a major apparent deficiency in supporting this novel discovery. **ChIP-seq for the H3K9MTs SUV39H1/2 would significantly strengthen this study.**

We appreciate the reviewer's suggestion and agree that understanding the mechanistic basis for H3K9me3 redistribution is critical to reinforcing our findings. Based on our RNA-Seq data from parental and multiple-KO mMSCs, we focused on SUV39H1 as the most relevant H3K9 methyltransferase for further investigation. SUV39H2 is minimally expressed in somatic cells and is primarily testis-enriched (supported by our own RNA-seq data, which indicates SUV39H2 is lowly expressed—Supplementary Fig. 5f), while SETDB1 predominantly targets repetitive elements and deposits focal H3K9me3 peaks. In contrast, SUV39H1 displays robust expression in our mMSC models and is known to form broad H3K9me3 domains—making it a compelling candidate for mediating the observed redistribution (Supplementary Fig. 5f in our revised manuscript). We initially attempted ChIP-Seq using multiple antibodies targeting endogenous SUV39H1; however, none yielded reliable or reproducible signal. As a solution, we overexpressed FLAG-tagged SUV39H1 and performed ChIP-seq using an anti-FLAG antibody in both parental and TKO mMSCs. Strikingly, SUV39H1 redistributes from heterochromatic cluster B regions to euchromatic cluster A regions upon H3K36me depletion in the TKO cell line, closely mirroring the redistribution pattern of H3K9me3 (Fig. 5e-h in our revised manuscript). Additionally, consistent with the gain of H3K9me3 within actively transcribed genes, we observed increased SUV39H1 occupancy in gene bodies in TKO cells (Fig. 5g). To further support this mechanism, we performed ChIP-Seq for endogenous HP1, a canonical reader of H3K9me3 (Figs. 6a-c; Supplementary Figs. a,b). We found that HP1 similarly redistributes into the same euchromatic regions where SUV39H1 and H3K9me3 accumulate following H3K36me loss. These results reinforce a model in which SUV39H1 mediates ectopic H3K9me3 deposition in the absence of

H3K36 methylation and that this redistribution is accompanied by HP1 binding in these regions. Finally, as noted in our response to Reviewer #2, endogenous SUV39H1 expression remains largely unchanged in TKO mMSCs (Supplementary Fig. 5f), further supporting the notion that its redistribution, rather than altered expression, underlies the ectopic H3K9me3 deposition. Together, these new ChIP-Seq data for SUV39H1 and HP1 provide mechanistic insight into the chromatin remodeling observed upon H3K36me loss and strengthen the evidence that SUV39H1 is the primary methyltransferase responsible for the H3K9me3 redistribution we describe.

2. The lack of intragenic DNA methylation change in highly expressed genes upon loss of H3K36me is a surprising finding. Given the gain of H3K9me in euchromatin and its direct link to DNA methylation, one could also speculate that the gain of H3k9me compensated for the loss of H3K36me in regulating genic DNA methylation. This scenario can be tested by knocking down/out H3K9MTs in the H3K36MT mutant background.

We agree that the preservation of intragenic DNA methylation in highly expressed genes despite the loss of H3K36me is unexpected and appreciate the reviewer's suggestion regarding potential compensatory mechanisms. As shown in Fig. 4a, our multiple-KO mMSCs exhibit a ~20% loss of DNA methylation at transcribed genes overall. However, stratification by gene expression revealed that this reduction is restricted to lowly expressed genes, while highly expressed genes retain high methylation levels despite complete depletion of H3K36me_{2/3}. Given the known recruitment of DNMT3A/B to gene bodies via H3K36me_{2/3} (Weinberg et al., 2019), we

generated DNMT3A/B double knockout (DKO) mMSCs to determine whether these enzymes deposit the residual methylation observed in the TKO mMSCs. The DNMT3A/B-DKO phenocopied the methylation loss seen in TKO cells at lowly expressed genes, but—similar to TKO cells—had no effect on methylation at highly expressed genes (see aggregate plots below). This suggests that residual methylation at highly expressed genes is maintained independently of both H3K36me and DNMT3A/B—potentially via DNMT1 maintenance mechanisms.

Minor concerns:

- The labels in some of the figures, especially in Figure 7a, are very small and hard to read. Increase the font size and/or improve figure image quality.

The font size has been increased and figure image quality has been improved, including in Figure 7a.

Reviewer #3 (Remarks on code availability):

Compiled code has been included with revisions. Please access here:

https://github.com/padilr1/H3K36me_guardian_epigenome_integrity_Padilla

Response to reviewers' comments

Reviewer #2 (Remarks to the Author):

Overall, the authors have made excellent progress in responding to the reviews and have adequately answered my questions. As suggested in my initial review, I believe it would be helpful to cite several important recent articles (listed below) that have looked at the functional effects of perturbed H3K36 methylation, and have shown a redistribution of heterochromatic modifications (namely H3K27me3) and how that leads to tissue-specific phenotypic changes. Given that the current manuscript has been carried out all in in vitro experiments, citing these studies would help support the larger importance of maintaining epigenetic homeostasis in vivo, and also offer an opportunity to suggest how their new findings may actually be playing a larger, yet still unexplored role in these other systems where H3K9 methylation was not assessed.

Pashos ARS, Meyer AR, Bussey-Sutton C, O'Connor ES, Coradin M, Coulombe M, Riemondy KA, Potlapelly S, Strahl BD, Hansson GC, Dempsey PJ, Brumbaugh J. H3K36 methylation regulates cell plasticity and regeneration in the intestinal epithelium. *Nat Cell Biol.* 2025 Feb;27(2):202-217. doi: 10.1038/s41556-024-01580-y. Epub 2025 Jan 8. PMID: 39779942.

Ko EK, Anderson A, D'souza C, Zou J, Huang S, Cho S, Alawi F, Prouty S, Lee V, Yoon S, Krick K, Horiuchi Y, Ge K, Seykora JT, Capell BC. Disruption of H3K36 methylation provokes cellular plasticity to drive aberrant glandular formation and squamous carcinogenesis. *Dev Cell.* 2024 Jan 22;59(2):187-198.e7. doi: 10.1016/j.devcel.2023.12.007. Epub 2024 Jan 9. PMID: 38198888; PMCID: PMC10872381.

Hoetker MS, Yagi M, Di Stefano B, Langerman J, Cristea S, Wong LP, Huebner AJ, Charlton J, Deng W, Haggerty C, Sadreyev RI, Meissner A, Michor F, Plath K, Hochedlinger K. H3K36 methylation maintains cell identity by regulating opposing lineage programmes. *Nat Cell Biol.* 2023 Aug;25(8):1121-1134. doi: 10.1038/s41556-023-01191-z. Epub 2023 Jul 17. PMID: 37460697; PMCID: PMC10896483.

Response:

We thank the reviewer for their helpful feedback and suggestions throughout the peer-review process. We have incorporated the recommended citations to the manuscript (references #19-21 in the revised manuscript).

Reviewer #3 (Remarks to the Author):

In this revised manuscript, the authors have carried out additional experiments addressing some key questions raised by the reviewers. These new results, especially the SUV39H1 ChIP-seq experiment offered new insights on H3K9me3 redistribution in

the H3K36 methyltransferase mutants. The overall quality and significance of the manuscript is much improved. I recommend this manuscript be accepted for publication.

Reviewer #3 (Remarks on code availability):

The codes are available with sufficient documentations. No concerns.

Response:

We thank the reviewer for their recommendation for publication, and for their insightful suggestions and feedback in strengthening the quality and impact of our manuscript.

Reviewer #4 (Remarks to the Author):

I was asked to review the rebuttal and revised manuscript in place of reviewer one who was unable to re-review this manuscript. As such, I had not previously seen or assessed the original draft. In this position I assessed the quality of the revised manuscript and the author's responses to the reviewers comments (mainly focussing on reviewer one). Having done so I believe the revised manuscript is an interesting and robust assessment of the regulation and regulatory capacity of H3K36me. It represents a large amount of work that is impactful for the field of epigenetics and gene regulation, and explores important topics such as chromatin cross-talk and the impact of altered modification state on higher order chromosomal organisation.

With regard to reviewer comments, the authors did directly address several key concerns, but mainly elected to respectfully disagree with many of the points made. Working through these points, I believe that the additions to the paper in response to all reviewers has strengthened their findings. Where comments were not addressed, I believe that the justification that these were not addressed are sound. Particularly as many questions related to findings that have been previously published by the authors, additional redundant assays to those already performed or simply (costly) experiments that are beyond the scope of this already substantial study.

The results are impactful, critically appraised, robustly performed, interesting and suitable for publication in Nature communications in their current form. I enjoyed reviewing this study.

Response:

We thank the reviewer for their thoughtful evaluation and positive assessment of our revised manuscript. We appreciate their recognition of the scope, rigor, and impact of our study, as well as their acknowledgment that our responses and justifications appropriately addressed the prior reviewers' comments. We are grateful for their supportive recommendation for publication.